# ADAPTIVE REINFORCEMENT LEARNING FOR UNOBSERVABLE RANDOM DELAYS

## ABSTRACT

In standard Reinforcement Learning (RL) settings, the interaction between the agent and the environment is typically modeled as a Markov Decision Process (MDP), which assumes that the agent observes the system state instantaneously, selects an action without delay, and executes it immediately. In real-world dynamic environments, such as cyber-physical systems, this assumption often breaks down due to delays in the interaction between the agent and the system. These delays can vary stochastically over time and are typically *unobservable* when deciding on an action. Existing methods deal with this uncertainty conservatively by assuming a known fixed upper bound on the delay, even if the delay is often much lower. In this work, we introduce the *interaction layer*, a general framework that enables agents to adaptively handle unobservable and time-varying delays. Specifically, the agent generates a matrix of possible future actions, anticipating a horizon of potential delays, to handle both unpredictable delays and lost action packets sent over networks. Building on this framework, we develop a model-based algorithm, *Actor-Critic with Delay Adaptation (ACDA)*, which dynamically adjusts to delay patterns. Our method significantly outperforms state-of-the-art approaches across a wide range of locomotion benchmark environments.

## 1 INTRODUCTION

State-of-the-art reinforcement learning (RL) algorithms, such as Proximal Policy Optimization (PPO) (Schulman et al., 2017) and Soft Actor-Critic (SAC) (Haarnoja et al., 2018), are typically built on the assumption that the environment can be modeled as a Markov Decision Process (MDP). This framework implicitly assumes that the agent observes the current state instantaneously, selects an action without delay, and executes it immediately.

This assumption often breaks down in real-world systems due to *interaction* delays that arise from various sources: the time taken to collect and transmit observations, the computation time needed for the agent to select an action, and the transmission and actuation delay when executing that action in the environment (as illustrated in Figure 1). Delays pose no issue if the state of the environment is not evolving between its observation and the execution of the selected action. But in continuously evolving systems–such as robots operating in the physical world–the environment's state may have changed by the time the action is executed (Brooks & Leondes, 1972). Delays have been recognized as a key concern when applying RL to cyber-physical systems (Tan et al., 2018). Outside the scope of RL, delays have also been studied in classic control (Ray, 1988; Luck & Ray, 1990).

These interaction delays can be implicitly modeled by altering the transition dynamics of the MDP to form a partially observable Markov decision process (POMDP), in which the agent only receives outdated sensor observations. While this approach is practical and straightforward, it limits the agent's access to information about the environment's evolution during the delay period.

Another common approach to handling delays in RL is to enforce that actions are executed after a fixed delay (Katsikopoulos & Engelbrecht, 2003; Walsh et al., 2008). This is typically implemented by introducing an action buffer between the agent and the environment, ensuring that all actions are executed after a predefined delay. However, this method requires prior knowledge of the maximum possible delay and enforces that all actions incur this worst-case delay—even when most interactions in practice experience minimal or no delay. The advantage of this fixed-delay approach is that it provides the agent with perfect information about when its actions will take effect, simplifying

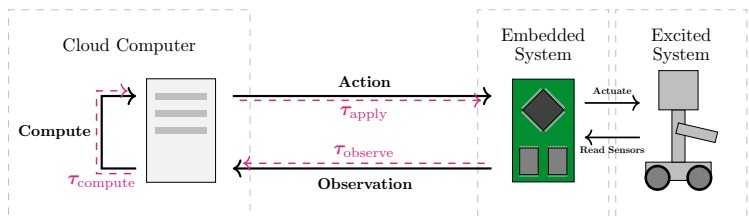

Figure 1: Illustration of a setup affected by interaction delays. Any delay between the embedded system and the excited system is considered negligible or otherwise accounted for (see Appendix F.1 for a detailed discussion). The factors contributing to interaction delay are $\tau_{\text{observe}}$ ($\tau_{\text{o}}$), $\tau_{\text{compute}}$ ($\tau_{\text{c}}$), and $\tau_{\text{apply}}$ ($\tau_{\text{a}}$). See Section 3.1 for more details about these factors.

decision-making. However, it is overly conservative and fails to adapt and account for variability in delay. Note that state-of-the-art algorithms for delayed MDPs, such as BPQL (Kim et al., 2023) and VDPO (Wu et al., 2024), rely on this fixed-delay paradigm.

Moving beyond this fixed-delay framework is challenging, especially because in real-world systems, delays are often unobservable. The agent does not know, at decision time, how long it will take for an action to be executed. One existing approach that attempts to address varying delays is DCAC (Bouteiller et al., 2021), but it does not offer any guarantees for when a generated action will be applied to the environment.

In this paper, we make the following contributions:

(i) We introduce a novel framework, the *interaction layer*, which allows agents to adapt to randomly varying delays—even when these delays are unobservable. In this setup, the agent generates a matrix of candidate actions ahead of time, each row in the matrix intended for a possible future arrival time (without knowing for certain which row will be selected). Specifically, the design handles both (a) that the future actions can have varying delays, and (b) that action packets sent over a network can be lost or arrive in incorrect order. The actual action is selected at the interaction layer once the delay is revealed. Similar to DCAC, we also report back the revealed delays in hindsight. This approach enables informed decision-making under uncertainty and robust behavior in the presence of stochastic, unobservable delays (Section 3).

(ii) We develop a new model-based reinforcement learning algorithm, *Actor-Critic with Delay Adaptation (ACDA)*, which leverages the interaction layer to adapt dynamically to varying delays. The algorithm provides two key concepts: (a) instead of using states as input to the policy, it uses a distribution of states as an embedding that enables the generation of more accurate time series of actions, and (b) an efficient heuristic to determine which of the previously generated actions are executed. These actions are needed to compute the state distributions. The approach is particularly efficient when delays are temporally correlated, something often seen in scenarios when communicating over transmission channels (Section 4).

(iii) We evaluate ACDA on a suite of MuJoCo locomotion tasks from the Gymnasium library (Towers et al., 2024), using randomly sampled delay processes designed to mimic real-world latency sources. Our results show that ACDA, equipped with the interaction layer, consistently outperforms state-of-the-art algorithms designed for fixed delays and for unobservable random delays. It achieves higher average returns across all benchmarks except one, where its performance remains within the standard deviation of the best constant-delay method (Section 5).

## 2 RELATED WORK

To our knowledge, there is no previous work that allows agents to make informed and controlled decisions under random unobservable delays in RL. Much of the existing work on how to handle delays in RL acts as if delays are constant equal to $h$, in which case, the problem can be modeled as an MDP with augmented state $(s_t, a_t, a_{t+1}, \ldots, a_{t+h-1})$ consisting of the last observed state and memorized actions to be applied in the future Katsikopoulos & Engelbrecht (2003). Even if the true delay is not constant, a construction used in previous work is to enforce constant interaction delay through *action buffering*, under the assumption that the maximum delay does not exceed $h$ time-steps.

Through action buffering and state augmentation, one may, in principle, use existing RL techniques to deal with constant delays. However, it is hard to directly learn policies on augmented states in practice, which has prompted the development of algorithms that exploit the delayed dynamics. The real-time actor-critic by Ramstedt & Pal (2019) optimizes for a constant delay of one time step. Belief projection-based Q-learning (BPQL) by Kim et al. (2023) explicitly uses the delayed dynamics under constant delay to simplify the critic learning. BPQL achieves good performance during evaluation over longer delays, despite a simple structure of the learned functions. Our algorithm in Section 4.3 uses the same critic simplification, but applied to the randomly delayed setting.

Another approach explored for constant-delay RL is to have a delayed agent trying to imitate an undelayed expert, used in algorithms such as DIDA (Liotet et al., 2022) and VDPO (Wu et al., 2024). These assume access to the undelayed MDP, which in the real world can be applied in sim-to-real scenarios, but not when training directly on the real physical system.

DCAC by Bouteiller et al. (2021) is a framework that allows agents to make decisions under unobservable delays, but without any control over when an action is going to be applied to the environment. Like our approach, delays are available in hindsight, which DCAC uses for future decision making and value accreditation. Other approaches for random delays typically assume observability (Valensi et al., 2024; Wu et al., 2025), which is not applicable in our problem setting.

Model-based approaches have also been explored for delayed RL, as a way to plan into future horizons (Chen et al., 2021) or to estimate future states as policy inputs (Walsh et al., 2008; Firoiu et al., 2018). A commonly used dynamics model architecture is the recurrent state space model (RSSM) (Hafner et al., 2019) that combines a recurrent latent state with stochastically sampled states to transition in latent space. RSSM was designed for planning algorithms, but can also be used for state prediction. The model used by Firoiu et al. (2018) is similar to RSSM, but uses deterministic output of states from the latent representation. Another approach using RSSM is Dreamer (Hafner et al., 2020) that learns a latent state representation for the agent to make decisions in, originally in an undelayed setting but extended to the delayed setting by Karamzade et al. (2024). Wang et al. (2024) have explored further variations in model structures that can be used for delayed RL.

Our approach also learns a model to make decisions in latent space, but does not follow the RSSM structure. Instead, our model (introduced in Section 4.2) follows a simpler structure that learns a latent representation describing actual state distributions rather than uncertainty about an assumed existing true state. By the definitions of Moerland et al. (2023), our model is classified as a multi-step prediction model with state abstraction, even though we are only estimating distributions.

## 3    THE INTERACTION LAYER

In this section, we explain how random and unpredictable delays may affect the interaction between the agent and the system. To handle these delays, we introduce a new framework, called the *interaction layer* (Section 3.2), and model the way the agent and the system interact by a Partially Observable MDP (Section 3.3). The notation used for the interaction layer is explained as it appears in the text. See Appendix C for a more compact, formal description of the interaction layer.

### 3.1    DELAYED MARKOV DECISION PROCESSES

We consider a controlled dynamical system modeled as an MDP $\mathcal{M} = (S, A, p, r, \mu)$, where $S$ and $A$ are the state and action spaces, respectively, where $p(s'|s, a)_{(s',s,a) \in S \times S \times A}$ represents the system dynamics in the absence of delays, $r$ is the reward function, and $\mu$ is the distribution of the initial state.

As in usual MDPs, we assume that at the beginning of each step $t$, the state of the system is sampled, but this information does not reach the agent immediately, but after an observation delay, $\tau_{\mathrm{o}}$. After the agent receives the information, an additional computational delay, $\tau_{\mathrm{c}}$, occurs due to processing the information and deciding on an appropriate action. The action created by the agent is then communicated to the system, with an additional final delay $\tau_{\mathrm{a}}$ before this action can be applied to the system. The delays[1] $(\tau_{\mathrm{o}}, \tau_{\mathrm{c}}, \tau_{\mathrm{a}})$ are random variables that may differ across steps and can be

---

[1] We assume that The total delay $\tau_{\mathrm{o}} + \tau_{\mathrm{c}} + \tau_{\mathrm{a}}$ is measured in number of steps. In Appendix G.1, we present a similar model where delays can take any real positive value.

correlated. While it is possible to consider frameworks where $\tau_\text{o}$ and $\tau_\text{c}$ are observable, the action delay $\tau_\text{a}$ is inherently unobservable, as this delay may be caused by events taking place after the action has been generated. Therefore, to simplify the problem in our framework, we consider the sum of these three delays as a single delay $d_t$, which is unobservable. This single delay represents the full round-trip delay of observing, computing an action, and applying the action to the system. This is further explained in Section 3.3.

## 3.2 HANDLING DELAYS VIA THE INTERACTION LAYER

The unpredictable delays pose significant challenges from the agent's perspective. First, the agent cannot respond immediately to the newly observed system state at each step. Second, the agent cannot determine when the selected action will be applied to the system. To address these issues, we introduce the *interaction layer*, consisting of an *observer* and an *action buffer*, as illustrated in Figure 2. The interaction layer is a direct part of the system that performs sensing and actuation, whereas the agent can be far away, communicating over a network. Within the interaction layer, the observer is responsible for sampling the system's state and sending relevant information to the agent. The agent generates a matrix of possible actions. These are sent back to the interaction layer and stored in the action buffer. Depending on when the actions arrive in the action buffer, it selects a row of actions, which are then executed in the following steps if no further decision is received. The rest of this section gives technical details of the interaction layer, whereas Section 4 details the policy for generating actions at the agent.

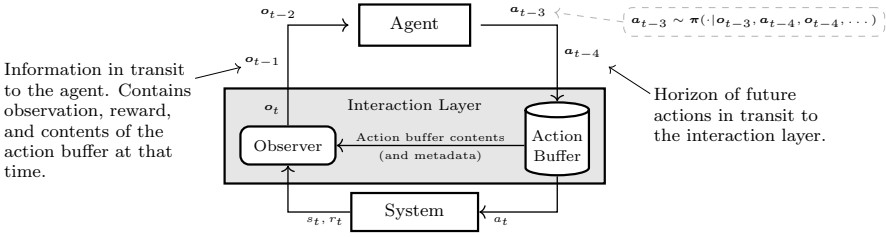

Figure 2: Illustration of the interaction layer and how the agent interacts with it from a global perspective. As the observation is received from the dynamical system, the next action is immediately applied from the action buffer. Packets in transit with random delay imply partial observability.

**Action packet.** After that, the agent receives an observation packet $\boldsymbol{o}_t$ (generated at step $t$ by the interaction layer, described further below), the agent generates and sends an action packet $\boldsymbol{a}_t$. The packet includes a time stamp $t$, and a matrix of actions, as follows:

$$
\boldsymbol{a}_t = \left( t, \begin{bmatrix} a_1^{t+1} & a_2^{t+1} & a_3^{t+1} & \ldots & a_h^{t+1} \\ a_1^{t+2} & a_2^{t+2} & a_3^{t+2} & \ldots & a_h^{t+2} \\ \vdots & \vdots & \vdots & \ddots & \vdots \\ a_1^{t+L} & a_2^{t+L} & a_3^{t+L} & \ldots & a_h^{t+L} \end{bmatrix} \right). \tag{1}
$$

The $i$-th row of the matrix of the action packet corresponds to the sequence of actions that would constitute the action buffer if the packet reaches the interaction layer at time $t + i$. The reason for using a matrix instead of a vector is that subsequent columns specify which actions to take if a new action packet does not arrive at the interaction layer at a specific time step. For instance, if an action packet arrives at time $t + 2$, then the interaction layer uses the first action in the buffer ($a_1^{t+2}$ in this case). That is, the first column is always used when a new packet arrives at each time step. If no packet arrives for a specific time step, the other columns are used instead (as explained more below in the description of the action buffer). This approach enables adaptivity for the agent: it can generate actions for specific delays without knowing what the delay is going to be ahead of time. Figure 3 illustrates when an action packet arrives at the interaction layer and a row is inserted into the action buffer (3rd row in this case because the packet arrived with a delay of 3).

The capacity of the action buffer is denoted by $h$, the horizon of actions to cover for gaps in the interaction. The number of rows in the matrix, denoted by $L$ (prediction length), is determined by

the agent. If the delay associated with the action packet exceeds the number of rows $L$ in the matrix, that action packet is discarded. Additionally, if an action packet arrives out of order, where $\boldsymbol{a}_t$ arrives after $\boldsymbol{a}_{t'}$ and $t < t'$, $\boldsymbol{a}_t$ is discarded. This process is formally described in Appendix C.

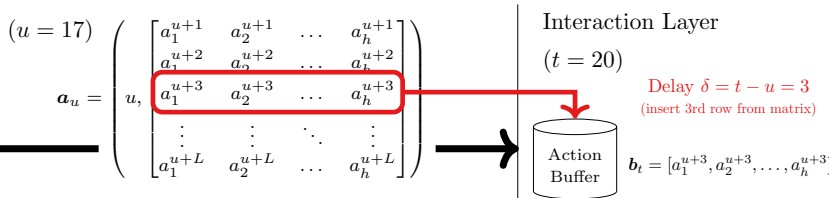

Figure 3: Example (see below for definitions of recovered delay $\delta_t$ and shift counter $c_t$): Suppose an action packet timestamped by the agent with time $u = 17$, $\boldsymbol{a}_u$, arrives at the action layer at time 20. Then, at time $t = 20$, $\delta_{20} = 3$, and $c_{20} = 0$. Now, suppose that 2 time units elapse without any new action packet arriving. Then, at time $t = 22$, $\delta_{22} = 3$ and counter $c_{22} = 2$. Hence, equation $t = u + \delta_{22} + c_{22} = 17 + 3 + 2 = 22$ holds.

While this may appear as if action delays are observable, the action packet only allows us to specify what should happen if it arrives with a certain delay. If the action delay truly was observable, we could use information about delays for previous action packets to get perfect information about which actions will be applied to the underlying state prior to this action packet arriving.

**Action buffer.** The action buffer is responsible for executing an action each time step. If no new action arrives at a time step, the next item in the buffer is used. At the beginning of step $t$, the action buffer contains the following information: $\boldsymbol{b}_t$, a sequence of $h$ actions to be executed next, and $\delta_t$, the delay of the action packet from which the actions $\boldsymbol{b}_t$ were taken. For instance, if an action packet $\boldsymbol{a}_u$ arrives at the action buffer at time $t$, then $\delta_t = t - u$, where $u$ is the time stamp of the action packet $\boldsymbol{a}_u$ that the agent created. If instead no new action packet arrived at time $t$, then $\delta_t = \delta_{t-1}$. To enable the use of an appropriate action even if no new packet arrives at a specific time step, the content of the buffer is shifted one step forward, as shown in Figure 4. Finally, the action buffer includes a counter $c_t$ that records how many steps have passed since the action buffer was updated. The following invariant always holds: $t = u + \delta_t + c_t$. For a concrete example, see the caption of Figure 3.

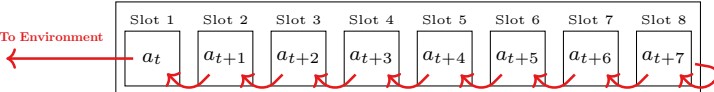

Figure 4: Action buffer shifting actions. Final slot is repeated. (Example: horizon $h = 8$)

**Observation packet.** The observer builds an observation packet $\boldsymbol{o}_t$ at the beginning of step $t$. To this aim, it samples the system state $s_t$, collects information $\boldsymbol{b}_t, \delta_t, c_t$ about the action buffer, forms the observation packet $\boldsymbol{o}_t = (t, s_t, \boldsymbol{b}_t, \delta_t, c_t)$, and sends it to the agent.

Enhancing the information contained in the observation and action packets (compared to the undelayed MDP scenario) allows the agent to make more informed decisions and ensures the system does not run out of actions when action packets experience delays. However, this is insufficient to model our delayed system as an MDP. This is because the agent does not have the knowledge of all the observation and action packets currently in transit. Therefore, we use the formalism of a POMDP to accurately describe the system dynamics.

### 3.3 THE POMDP MODEL

Next, we complete the description of our delayed MDP and model it as a POMDP. To this aim, we remark that the system essentially behaves as if in each step $t$, the agent immediately observes $\boldsymbol{o}_t$ and selects an action packet $\boldsymbol{a}_t$ that arrives at the interaction layer $d_t$ steps after the observation

$\boldsymbol{o}_t$ was made, where $d_t > 0$[2]. We assume that $d_t$ is generated according to some distribution $D$[3]. Furthermore, we assume that observation packets $\boldsymbol{o}_t$ arrive in order at the agent. In this framework—where the agent selects an action packet $\boldsymbol{a}_t$ as soon as the observation $\boldsymbol{o}_t$ is generated—the single round-trip delay $d_t$ represents the three delays $(\tau_{\mathrm{o}}, \tau_{\mathrm{c}}, \tau_{\mathrm{a}})$ as $d_t = \tau_{\mathrm{o}} + \tau_{\mathrm{c}} + \tau_{\mathrm{a}}$.

The time step $t$ is a local time tag from the perspective of the interaction layer. Our POMDP formulation does not assume a synchronized clock between the agent and the interaction layer. The agent acts asynchronously and generates an action packet upon receiving an observation packet.

We define $\mathcal{I}_t$ as the set of action packets in transit at the beginning of step $t$, along with the times at which these packets will arrive at the interaction layer ($\mathcal{I}_t$ is a set containing items on the form $(u + d_u, \boldsymbol{a}_u)$). In reality, delays are observed only when action packets reach the interaction layer, and the agent does not necessarily know whether the action packets already generated have reached the interaction layer. Hence, we must assume that $\mathcal{I}_t$ is not observable by the agent. The framework we just described corresponds to a POMDP, which we formalize in Appendix C in detail.

## 4    Actor-Critic with Delay Adaptation

This section introduces *actor-critic with delay adaptation* (ACDA), a model-based RL algorithm using the interaction layer to adapt on-the-fly to varying unobservable delays, contrasting with state-of-the-art methods that enforce a fixed worst-case delay. A challenge with varying unobservable delays is that the agent lacks perfect information about the actions to be applied in the future. ACDA solves this with a heuristic (Section 4.1) that is effective when delays are temporally correlated.

The actions selected by ACDA will vary in length depending on the delay we are generating actions for. This lends itself poorly to commonly used policy function approximators in deep RL, such as multi-layer perceptrons (MLPs), that assume a fixed size of input. ACDA solves this with a model-based distribution agent (Section 4.2) that embeds the variable-length input into fixed-size embeddings of future state distributions, to which the generated action will be applied. The fixed-size embeddings are fed as input to an MLP to generate actions. ACDA learns a model of the environment dynamics online to compute these embeddings. Section 4.3 shows how we train ACDA.

### 4.1    Heuristic for Assumed Previous Actions

A problem with unobservable delays is that we do not know when our previously sent action packets will arrive at the interaction layer. This means that we do not know which actions are going to be applied to the underlying system between generating the action packet and it arriving at the interaction layer. A naive assumption would be to assume the action buffer contents reported by the observation packet to be the actions that are going to be applied to the underlying system. However, this is unlikely to be true because the action buffer is going to be preempted by action packets already in transit.

ACDA employs a heuristic for estimating these previous actions to be applied to the system between $\boldsymbol{o}_t$ being generated and $\boldsymbol{a}_t$ arriving at the interaction layer. The heuristic assumes that, if $\boldsymbol{a}_t$ arrives at time $t + k$ (it having delay $k$), then previous action packets will also have delay $k$. Such that $\boldsymbol{a}_{t-1}$ will arrive at time $t + k - 1$, $\boldsymbol{a}_{t-2}$ at $t + k - 2$, etc.

Under this assumption, a new action packet will preempt the action buffer at every single time step. This means that, if we assume a delay of $k$, the action applied to the underlying system will be the action in the first column of the $k$-th row in the action packet last received by the interaction layer. By memorizing the action packets

---

**Algorithm 1** Memorized Action Selection

**Input**  $k \in \mathbb{Z}^+$                     (Delay assumption)
        $\boldsymbol{a}_{t-1}, \boldsymbol{a}_{t-2}, \ldots, \boldsymbol{a}_{t-k}$   (Memorized Packets)
1: **for** $i \leftarrow 1$ to $k$ **do**
2:     $(t - i, M^{t-i}) = \boldsymbol{a}_{t-i}$
3:       ▷ Unpacking action matrix M from packets
4: **return** $(\hat{a}_1^{t+k}, \ldots, \hat{a}_k^{t+k}) = (M_{k,1}^{t-k}, \ldots, M_{k,1}^{t-1})$

---

[2]$d_t$ corresponds to the value $\delta_u$ if the action packet reaches the interaction layer at time $u$: $\delta_{t+d_t} = d_t$

[3]For simplicity, we assume that the delay process is markovian and independent of the contents of the action packets and the state of the interaction layer. However, our POMDP formalism can be extended to delay distributions that depend on the previous state, which is more general and realistic.

previously sent, we can under this assump-
tion select the actions that are going to be applied to the system as shown in Algorithm 1. When generating $a_1^{t+k}$, the first action on the $k$-th row in the action packet $\boldsymbol{a}_t$, we use Algorithm 1 to determine the actions $(\hat{a}_1^{t+k}, \ldots, \hat{a}_k^{t+k})$ that will be applied to the observed state $s_t$ before $a_1^{t+k}$ is executed. For the action $a_2^{t+k}$, we know that this is only going to be executed if no new action packet arrived at $t+k+1$. We therefore extend the previous assumption and say that $(\hat{a}_1^{t+k}, \ldots, \hat{a}_k^{t+k}, a_1^{t+k})$ are the actions applied to $s_t$ before $a_2^{t+k}$ is executed.

The main idea here is that the heuristic guesses the applied actions if the delay does not evolve too much over time. If the delay truly was constant, then all guesses would be accurate and ACDA would transform the POMDP problem to a constant-delay MDP. The heuristic's accuracy is compromised during sudden changes in delay, such as network delay spikes. However, as we will see in the evaluation, occasional violations will not significantly impact overall performance.

## 4.2 MODEL-BASED DISTRIBUTION AGENT

The memorized actions used by ACDA are variable in length and therefore cannot be directly used as input to MLPs, which are often used in constant-delay approaches. Instead, ACDA constructs an embedding $z_1^{t+k}$ of the distribution $p(s_{t+k}|s_t, \hat{a}_1^{t+k}, \ldots, \hat{a}_k^{t+k})$, where $\hat{a}_1^{t+k}, \ldots, \hat{a}_k^{t+k}$ are the memorized actions. We then provide $z_1^{t+k}$ as input to an MLP to generate $a_1^{t+k}$. This allows the policy to reason about the possible states in which the generated action will be executed. Note that we are only concerned with the distribution itself and never explicitly sample from it. To compute these embeddings, we learn a model of the system dynamics using three components: EMBED$_\omega$, STEP$_\omega$, and EMIT$_\omega$, where $\omega$ represents learnable parameters.

- $\hat{z}_0 = \text{EMBED}_\omega(s_t)$ embeds a state $s_t$ into a distribution embedding $\hat{z}_0$.
- $\hat{z}_{i+1} = \text{STEP}_\omega(\hat{z}_i, a_{t+i})$ updates the embedded distribution to consider what happens after also applying the action $a_{t+i}$. Such that if $\hat{z}_i$ is an embedding of $p(s_{t+i}|s_t, a_t, \ldots, a_{t+i})$, then $\hat{z}_{i+1}$ is an embedding of $p(s_{t+i+1}|s_t, a_t, \ldots, a_{t+i}, a_{t+i+1})$.
- The final component $\text{EMIT}_\omega(s_{t+i}|\hat{z}_i)$ allows for a state to be sampled from the embedded distribution. This component is not used when generating actions, and is instead only used during training to ensure that $\hat{z}_i$ is a good embedding of $p(s_{t+i}|s_t, a_t, \ldots, a_{t+i})$.

The way these components are used to produce the embedding $z_1^{t+k}$ is illustrated in Figure 5. We use the notation $z_1^{t+k} = \hat{z}_k$ given that we are embedding the selected actions $(\hat{a}_1^{t+k}, \ldots, \hat{a}_k^{t+k})$. We use the notation $\text{STEP}_\omega^k(\text{EMBED}_\omega(s_t), \hat{a}_1^{t+k}, \ldots, \hat{a}_k^{t+k})$ to describe this multi-step embedding process. This notation is formalized in Appendix D.

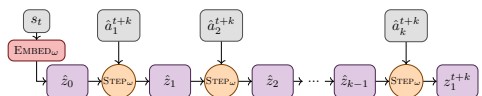

Figure 5: Illustration of the multi-step distribution model embedding $p(s_{t+k}|s_t, \hat{a}_1^{t+k}, \ldots, \hat{a}_k^{t+k})$ as $\text{STEP}_\omega^k(\text{EMBED}_\omega(s_t), \hat{a}_1^{t+k}, \ldots, \hat{a}_k^{t+k})$.

The EMBED$_\omega$ and EMIT$_\omega$ components are implemented as MLPs, while STEP$_\omega$ is implemented as a gated recurrent unit (GRU). We provide detailed descriptions of these components in Appendix D. We learn these components online by collecting information from trajectories about observed states $s_t$ and $s_{t+n}$ and their interleaved actions $a_t, a_{t+1}, \ldots, a_{t+n-1}$ in a replay buffer $\mathcal{R}$. The following loss function $\mathcal{L}(\omega)$ is used to minimize the KL-divergence between the model and the under-

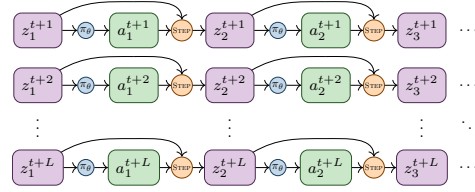

Figure 6: Generating the action packet from the embeddings. Each row in the figure corresponds to a row in the matrix of the action packet $\boldsymbol{a}_t$.

lying system dynamics: $\mathcal{L}(\omega) = \mathbb{E}_{(s_t, a_t, a_{t+1}, \ldots, a_{t+n-1}, s_{t+n}) \sim \mathcal{R}} \left[ -\log \text{EMIT}_\omega(s_{t+n}|z_n) \right]$ where $z_n = \text{STEP}_\omega^n(\text{EMBED}_\omega(s_t), a_t, a_{t+1}, \ldots, a_{t+n-1})$.

Given the embedding $z_1^{t+k}$, we produce $a_1^{t+k}$ in the action packet $\boldsymbol{a}_t$ using a policy $\pi_\theta(a_1^{t+k}|z_1^{t+k})$, i.e., generating actions given the (embedded) distribution over the state that the action will be ap-

plied to. This policy structure allows the agent to reason about uncertainties in future states when generating actions.

By extending the assumptions as shown in Section 4.1, we can also produce the embeddings $z_2^{t+k}$, $z_3^{t+k}$, etc., as illustrated in Figure 6. This process of generating the matrix rows is similar to action chunking Lai et al. (2022); Zhao et al. (2023); Li et al. (2025), though we use them to cover gaps in the interaction, rather than with the expectation that they will all be executed. The complete process of constructing the action packet is formalized in Appendix D. We also discuss the effect that this has on the computational delay in Appendix F, why it is not a problem in our case, and how to handle it if it should become a problem.

This model-based policy can also be applied in the constant-delay setting to achieve decent performance. We evaluate how this compares against a direct MLP function approximator in Appendix E.2, where the model-based policy is implemented in the BPQL algorithm.

### 4.3 TRAINING ALGORITHM

This section describes the training procedure in Algorithm 2, used to optimize the parameters of the networks. It follows an actor-critic setup based on SAC. The training procedure of the critic $Q_\phi$ is similar to BPQL, where $Q_\phi(s, a)$ evaluates the value of actions $a$ on undelayed system states $s$.

Algorithm 2 is split into three parts: trajectory sampling (L3-L12), transition reconstruction (L13), and training (L14-L15). We do this split to reduce the impact that the training procedure can have on the computational delay $\tau_c$ of the system. From the trajectory sampling, we collect POMDP transition information $(\boldsymbol{o}_t, \boldsymbol{a}_t, r_t, \boldsymbol{o}_{t+1})$ where $\Gamma_t$ is used to discern if $\boldsymbol{s}_t$ is in a terminal state.

An important aspect of Algorithm 2 is how trajectory information is reconstructed for training. Specifically, we reconstruct the trajectory $(s_0, a_0, r_0, s_1, a_1, \dots)$ from the perspective of the undelayed MDP, along with the policy input used to generate each action $a_t$. The policy input can be retrospectively recovered by examining the current buffer action delay $(\delta_t)$ and the number of times the buffer has shifted $(c_t)$. This trajectory reconstruction is necessary since we follow the BPQL algorithm's actor-critic setup. The critic $Q_\phi(s_t, a_t)$ estimates values in the undelayed MDP, and we need to be able to regenerate actions $a_t$ using the model-based policy to compute the TD-error. Further details are provided in Appendix D. The hyperparameters used for ACDA, including the prediction length $L$, are located in Appendix B.4.

---

**Algorithm 2** Actor-Critic with Delay Adaptation

1: Init. policy $\pi_\theta$, critic $Q_\phi$, model $\omega$, and replay $\mathcal{R}$
2: **for** each epoch **do**
3:     Reset interaction layer state: $\boldsymbol{s}_0 \sim \boldsymbol{\mu}, t = 0$
4:     Collected trajectory: $\mathcal{T} = \emptyset$
5:     Observe $\boldsymbol{o}_0$
6:     **while** terminal state not reached **do**
7:         **for** $k \leftarrow 1$ to $L$ **do**
8:             Select $\hat{a}_1^{t+k}, \dots, \hat{a}_k^{t+k}$ by Alg. 1
9:             Create the $k$-th row of $\boldsymbol{a}_t$
10:         Send $\boldsymbol{a}_t$, observe $r_t, \boldsymbol{o}_{t+1}, \Gamma_{t+1}$
11:         Add $(\boldsymbol{o}_t, \boldsymbol{a}_t, r_t, \boldsymbol{o}_{t+1}, \Gamma_{t+1})$ to $\mathcal{T}$
12:         $t \leftarrow t + 1$
13:     Reconstruct transition info from $\mathcal{T}$, add to $\mathcal{R}$
14:     **for** $|\mathcal{T}|$ sampled batches from $\mathcal{R}$ **do**
15:         Update $\pi_\theta$, $Q_\phi$ and $\omega$ (by $\mathcal{L}(\omega)$)

---

## 5 EVALUATION AND RESULTS

To assess the benefits of the interaction layer in a delayed setting, we simulate the POMDP described in Section 3.3, wrapping existing environments from the Gymnasium library (Towers et al., 2024) as the underlying system. Specifically, we aim to answer the question of whether our ACDA algorithm, which uses information from the interaction layer, can outperform state-of-the-art algorithms under random delay processes.

We evaluate on the three delay processes shown in Figure 7. The first two delay processes $GE_{1,23}$ and $GE_{4,32}$ follow Gilbert-Elliot models (Gilbert, 1960; Elliott, 1963) where the delay alternates between good and bad states (e.g. a network or computational node being overloaded or having packets dropped). The third delay process MM1 is modeled after an M/M/1 queue (Kleinrock, 1975), where the sampled delay is the time spent in the queue by a network packet. The full definition of these delay processes is located in Appendix B.3. We expect ACDA to perform well under the Gilbert-

Elliot processes that match the temporal assumptions of ACDA. In contrast, we anticipate a worse relative performance of ACDA compared to other baselines under M/M/1 queue delays that fluctuate more.

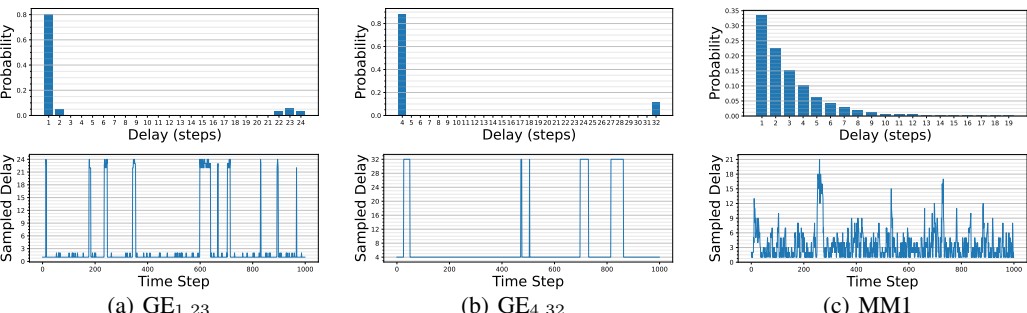

Figure 7: Evaluated delay processes, as a distribution histogram (above) and as a time series sampled delay (below) for each delay process. See Appendix B.3 for their definitions.

The state-of-the-art algorithms we compare against are DCAC (Bouteiller et al., 2021), BPQL (Kim et al., 2023), and VDPO (Wu et al., 2024). As BPQL and VDPO are designed to operate under constant delay, we apply a *constant-delay augmentation* (CDA) to allow them to operate with constant delay in random delay processes. CDA converts the interaction layer POMDP into a constant-delay MDP by making agents act under the worst-case delay of a delay process[4], achieving the same effect as common constant-delay action buffers, as detailed in Appendix A. In addition to the state-of-the-art algorithms, we also evaluate the performance of SAC, both with CDA and when it acts directly on the state from the observation packet (implicitly modeling delays). In Appendix E.3, we evaluate when CDA uses a much lower assumed worst-case delay that holds most of the time, but is occasionally violated. Although this lower assumed delay yields increased performance for in some benchmarks, for other benchmarks the constant-delay algorithms performs worse or inhibit unexpected behavior, which is why we use the conservative worst-case for the evaluation here. We also evaluate the performance of Dreamer when implicitly modeling delays (Karamzade et al., 2024). Further details regarding the evaluation are presented in Appendix B.5.

We evaluate average return over a training period of 1 million steps on MuJoCo environments in Gymnasium, following the procedure from related work in delayed RL. However, an issue with the MuJoCo environments is that they have deterministic transition dynamics, rendering them theoretically unaffected by delay. To better evaluate the effect of delay, we make the transitions stochastic by imposing a 5% noise on the actions. We motivate and specify this in Appendix B.1.

The average return is computed every 10000 steps by freezing the training weights and sampling 10 trajectories under the current policy. We report the best achieved average return—where the return is the sum of rewards over a trajectory—for each algorithm, environment, and delay process in Table 1. All achieved average returns are also presented in Appendix E.1 as time series plots together with tables showing the standard deviation.

Table 1: Best evaluated average return for each algorithm.

| *Gymnasium env.* | Ant-v4 | | | Humanoid-v4 | | | HalfCheetah-v4 | | | Hopper-v4 | | | Walker2d-v4 | | |
|---|---|---|---|---|---|---|---|---|---|---|---|---|---|---|---|
| *Delay process* | $GE_{1,23}$ | $GE_{4,32}$ | MM1 | $GE_{1,23}$ | $GE_{4,32}$ | MM1 | $GE_{1,23}$ | $GE_{4,32}$ | MM1 | $GE_{1,23}$ | $GE_{4,32}$ | MM1 | $GE_{1,23}$ | $GE_{4,32}$ | MM1 |
| SAC | 14.22 | −5.72 | −0.58 | 862.18 | 494.43 | 921.04 | 2064.18 | −158.78 | 20.69 | 306.91 | 279.74 | 333.06 | 708.33 | 60.86 | 604.80 |
| SAC w/ CDA | 69.28 | 18.93 | 102.00 | 414.05 | 230.45 | 613.03 | 128.47 | 591.32 | 550.84 | 426.92 | 315.47 | 627.59 | 428.44 | 257.18 | 2005.76 |
| Dreamer | 1111.73 | 1147.56 | 1121.11 | 1463.07 | 1091.48 | 981.38 | 1796.07 | 2493.19 | 584.40 | 334.30 | 515.36 | 975.72 | 1081.12 | 1233.79 | 1801.81 |
| BPQL | 2691.88 | 2509.52 | **3074.17** | 585.19 | 276.63 | 5435.29 | 4320.20 | 2136.36 | 4660.93 | 1328.71 | 433.29 | 3035.66 | 1215.91 | 875.09 | 3547.73 |
| VDPO | 2163.00 | 2266.99 | 2528.67 | 417.25 | 280.72 | 720.73 | 3144.23 | 3664.30 | 3831.96 | 709.20 | 330.44 | 1459.62 | 846.88 | 344.73 | 2144.25 |
| DCAC | 949.97 | 953.14 | 959.23 | 128.47 | 167.97 | 525.85 | 920.09 | 1123.47 | 35.60 | 16.99 | 57.98 | 1026.45 | 106.70 | 9.23 | 24.48 |
| ACDA | **4112.78** | **2866.93** | 2898.46 | **4608.76** | **3725.59** | **5805.60** | **5984.25** | **4231.15** | **5898.36** | **2094.65** | **1727.79** | **3122.53** | **3863.59** | **1840.58** | **4562.33** |

As shown in Table 1, ACDA outperforms state-of-the-art in all benchmarks except one, with a significant margin in most cases. The improvement is less substantial in Ant-v4, where performance often overlaps, as indicated by the standard deviation (Figure 8).

---

[4]The MM1 delay process does not have a maximum delay. We use a reasonable worst-case of 16 steps.

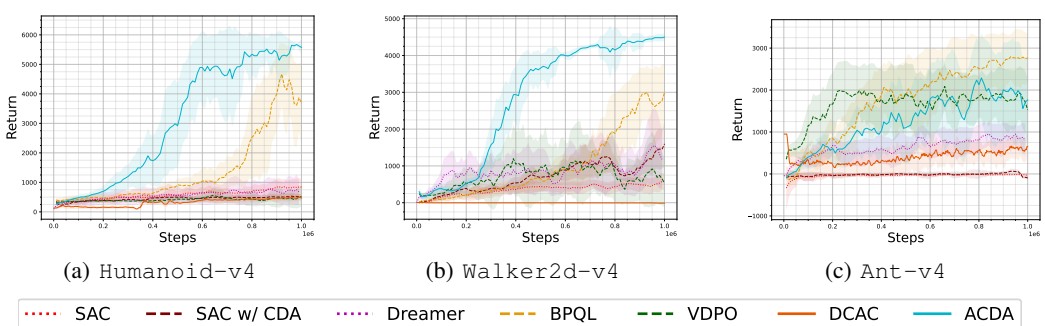

Figure 8: M/M/1 Queue results (all results in Appendix E.1.3). Shaded regions showing std. dev.

Lowering the assumed upper bound delay for constant-delay baselines can also yield better performance for some environments. We present results under lower (incorrect) assumed worst-case delay in Appendix E.3, where constant-delay baselines under $GE_{4,32}$ perform better than ACDA in 3 of the environments. However, across all assumed delays for constant-delay baselines, whose results are presented in Appendix E.5, ACDA is still the best performing algorithm in 10 out of 15 benchmarks.

## 6 CONCLUSION

We introduced the interaction layer, a real-world viable POMDP framework for RL with random unobservable delays. Using the interaction layer, we described and implemented ACDA, a model-based algorithm that significantly outperforms state-of-the-art in delayed RL under random delay processes. Directions of future work include algorithms that can operate on wider areas of delay correlation and on alterations to the interaction layer to handle more complex interaction behavior. Example of alterations include more control of how action packets accepted or discarded, and optimizations to the action packet structure to reduce the amount of computation and transmitted information.

## REPRODUCIBILITY STATEMENT

We provide the source code for all experiments as supplementary material to the paper submission, including the raw measurements used to generate all plots and tables in the paper. The source code is accompanied by instructions for how to run the experiments and how to install the necessary dependencies. All experiments are performed using simulated environments, allowing anyone to reproduce the results shown here in the paper.

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

CONTENTS

# ICLR Statement on LLM Usage

This paper has made use of Large Language Models (LLMs) to polish writing. More specifically, to check for grammatical errors and for phrasing suggestions.

# Outline of the Appendices

Appendix A presents how we implement constant-delay augmentation (CDA) in our framework. This allows agents to act with constant delay using the interaction layer, even if the underlying delay process is stochastic. We primarily use this to provide a fair comparison against related work.

Appendix B presents the evaluation details. In Appendix B.1, we demonstrate that stochastic transitions are necessary to see the effects of delay, both theoretically and with an evaluated example. We also show in Appendix B.1 how we use action noise to convert deterministic transitions into stochastic ones. Further evaluation of the effect that stochasticity has on the best-performing algorithms is presented in Appendix B.2. Appendix B.3 describes the delay distributions used in the benchmarks. Lastly, in Appendix B.4, we present the hyperparameters used, and in Appendix B.5 we present the software and hardware used for running the benchmarks.

Appendix C formalizes the interaction layer as a POMDP. This POMDP is used to simulate the interaction layer in the benchmarks.

Appendix D formalizes the model and its objective, as well as providing an expanded version of the algorithm presented in Section 4.3.

Appendix E contains additional results. Appendix E.1 contains all results presented in the conclusion, with time series plots and standard deviation. In Appendix E.2, we evaluate the model-based distribution agent under CDA, showing that it is the adaptiveness that leads to gains in performance rather than the policy itself. In Appendix E.3, we evaluate the effect of using a lower bound for CDA that holds most of the time, but is occasionally violated. The latter two Appendices E.2 and E.3 show that the adaptiveness of the interaction layer offers gains and stability in performance that cannot be obtained by operating in a constant-delay manner.

Appendix F discusses practical considerations when deploying the interaction layer to real-world environments.

Appendix G shows an alternative, more realistic definition of delayed MDPs, which uses real-valued delays rather than discrete.

# A  CONSTANT-DELAY AUGMENTATION

To be able to evaluate and compare fairly with state-of-the-art algorithms, we make it possible to have constant delay augmentation within our framework. Specifically, to allow algorithms such as BPQL and VDPO that expect a constant delay when the underlying delay process is stochastic, we apply a *constant-delay augmentation* (CDA) on top of the interaction layer. CDA converts the interaction layer POMDP into a constant-delay MDP, under the assumption that the maximum delay does not exceed $h$ steps. This augmentation ensures that we evaluate state-of-the-art as intended when comparing their performance against ACDA.

CDA is implemented on top of the interaction layer by simply arranging the contents of the action packet matrix such that, no matter when action packet $\boldsymbol{a}_t$ arrives (between $t+1$ and $t+h$), each action will be executed $h$ steps after it was generated. We illustrate this procedure of constructing the action packet in Algorithm 3.

---

**Algorithm 3** Constant-Delay Augmentation using the Interaction Layer

---

**Input**  $\boldsymbol{o}_t = (t, s_t, \boldsymbol{b}_t, \delta_t, c_t)$   (Observation packet)
$\quad\quad\quad \pi$   (Constant-delay policy operating on the horizon $h$)

1: $a_t, \ldots, a_{t+h-1} = \boldsymbol{b}_t$
2: $a_{t+h} \sim \pi(\cdot | s_t, a_t, \ldots, a_{t+h-1})$

3: $\boldsymbol{a}_t = \left( t, \begin{bmatrix} a_{t+1} & a_{t+2} & a_{t+3} & \cdots & a_{t+h-2} & a_{t+h-1} & a_{t+h} \\ a_{t+2} & a_{t+3} & a_{t+4} & \cdots & a_{t+h-1} & a_{t+h} & a_{t+h} \\ a_{t+3} & a_{t+4} & a_{t+5} & \cdots & a_{t+h} & a_{t+h} & a_{t+h} \\ \vdots & \vdots & \vdots & \ddots & \vdots & \vdots & \vdots \\ a_{t+h-1} & a_{t+h} & a_{t+h} & \cdots & a_{t+h} & a_{t+h} & a_{t+h} \\ a_{t+h} & a_{t+h} & a_{t+h} & \cdots & a_{t+h} & a_{t+h} & a_{t+h} \end{bmatrix} \right)$

4: **return** $\boldsymbol{a}_t$

---

This states that if $\boldsymbol{a}_t$ arrives at $t+i$, then the actions to be applied are $a_{t+i}, a_{t+\min(h,i+1)}, a_{t+\min(h,i+2)}, \ldots, a_{t+\min(h,i+(h-3))}, a_{t+\min(h,i+(h-2))}, a_{t+h}$, which ensures that $\boldsymbol{b}_t$ always is a correct guess of the actions to be applied next. For $i > 1$, we pad with $a_{t+h}$ to the right on each row to represent the shifting behavior. Forming the action packets in this way ensures that $a_{t+h}$ always gets executed at time $t+h$, given that the delay does not exceed $h$.

The policy $\pi$ can be any constant-delay augmented policy. We can also apply the model-based distribution policy from the ACDA algorithm to the CDA setting, by letting $\pi(a_{t+h}|s_t, a_t, \ldots, a_{t+h-1}) = \pi_\theta(a_{t+h}|z_h)$, where $z_h = \text{STEP}_\omega^h(\text{EMBED}_\omega(s_t), a_t, \ldots, a_{t+h-1})$. We present the results of this policy in Appendix E.2.

This assumes the horizon $h$ is a valid upper bound of the delay. We can still perform the augmentation if $h$ is less than the upper bound, but then we are no longer guaranteed the MDP properties of constant delay. We present the results of this in Appendix E.3.

# B EVALUATION DETAILS

This section provides a more complete overview of the evaluation and the results. We provide justification for choosing the 5% noise on environments (Appendix B.1), the delay processes used (Appendix B.3), the hyperparameters used and neural network architectures used (Appendix B.4), as well as the software and hardware used during evaluation (Appendix B.5). The complete results for all benchmarks are presented separately in Appendix E.

## B.1 ACTION NOISE AND ITS EFFECT ON PERFORMANCE

The benchmark environments used, as defined in Gymnasium, have fully deterministic transitions. As a result, they are theoretically unaffected by delay: the optimal value achievable in the delayed MDP is identical to that of the undelayed MDP. This follows trivially from the fact that, with a perfect deterministic model of the MDP dynamics, the agent can precisely predict the future state in which its action will be applied. Consequently, the agent can plan as if there were no delay at all.

The same is not true for MDPs with stochastic transition dynamics. To show this, consider the MDP with $S = \{H, T\}, A = \{H, T\}, r(H, H) = 1, r(T, T) = 1, r(H, T) = 0, r(T, H) = 0$, where $\forall s', s, a \quad p(s'|s, a) = 0.5$. This MDP models flipping a fair coin, where the agent is given a reward of 1 if it can correctly identify the face of the current coin. Consider this MDP with a constant delay of 1 time step. Now, the agent instead has to guess the face of the next coin, on which it can do no better than a 50/50 guess. Therefore, in this example, the value of an optimal agent in the delayed MDP is half of the value of an optimal agent in the undelayed MDP.

To better highlight the practical issues with delay, we add uncertainty to transitions in the Gymnasium environments by adding noise to the actions prior to being applied to the environment. Let $\beta$ be the noise factor indicating how much noise we add relative to the span of values that the action can take. Then we add noise to the actions $a$ as follows:

$$\text{Assume } a = [a(1), a(2), \ldots, a(n)] \tag{2}$$

$$a(i)_{\max} = \text{ maximum value for a(i)} \tag{3}$$

$$a(i)_{\min} = \text{ minimum value for a(i)} \tag{4}$$

$$\nu(i) = \beta \cdot (a(i)_{\max} - a(i)_{\min}) \cdot \xi \Big|_{\xi \sim \mathcal{N}(0,1)} \tag{5}$$

$$\tilde{a}(i) = \text{clip}\left(a(i) + \nu(i), a(i)_{\min}, a(i)_{\max}\right) \tag{6}$$

$$\tilde{a} = [\tilde{a}(1), \tilde{a}(2), \ldots, \tilde{a}(n)] \tag{7}$$

Here, we assume that the actions are continuous, which works since all environments in our evaluation are of this nature. Then the transitions become $p(s'|s, \tilde{a})$, with the noisy action applied instead of the original one. We use the noise factor $\beta = 0.05$ in all our noisy environments evaluated here.

To see the effect that this noise has on delayed RL in practice, we evaluate the performance of BPQL when trained over different constant delays, with and without noise. The results are plotted in Figure 9. The evaluation is done by training a BPQL policy on a specific constant delay and action noise, evaluating the policy's average return every 10000 steps, and reporting the best achieved average return as the performance. This evaluation procedure, which is used by all evaluations in the paper, is further described in Appendix B.5.

The results without noise for constant delays of 3, 6, and 9, shown in Table 2, are representative of those reported by Kim et al. (2023) (8100 ± 543.4, 6334.6 ± 245.3, and 5887.5 ± 270.5 for constant delays of 3, 6, and 9 respectively). This suggests that our implementation is faithful to their approach. Notably, we observe that in the deterministic setting, the impact of delay, while causing a significant initial drop in performance, does not lead to significant degradation over longer time horizons. This behavior contrasts with the noisy environments, where the performance declines more noticeably as the delay increases.

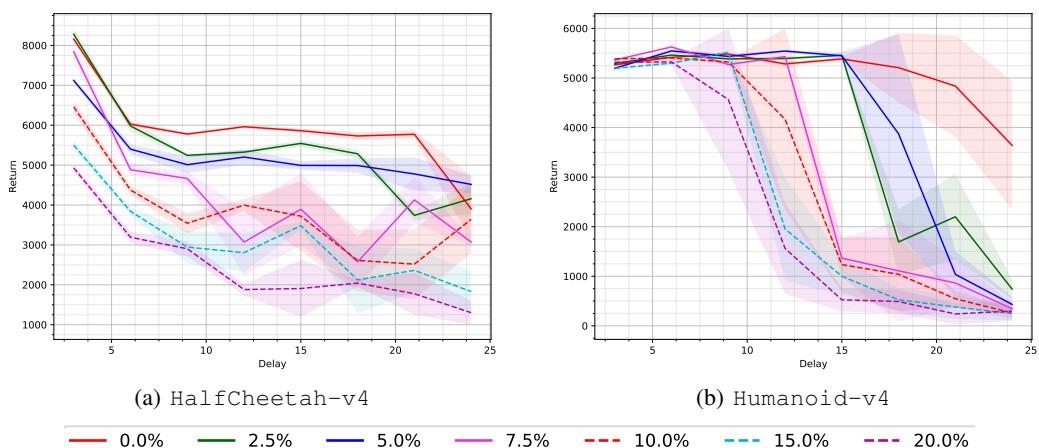

(a) `HalfCheetah-v4`  (b) `Humanoid-v4`

| 0.0% | 2.5% | 5.0% | 7.5% | 10.0% | 15.0% | 20.0% |

Figure 9: Best evaluated performance of BPQL after training over $10^6$ timesteps when different noise is applied to the `HalfCheetah-v4` and `Humanoid-v4` environments. Each line represents when a specific action noise is induced on that environment, indicated by the % in the legend (e.g. 2.5% means $\beta = 0.025$). Each plotted point represents the best evaluated average return when BPQL is trained on that noisy environment with that constant delay. The shaded regions represent the standard deviation.

Table 2: The noise evaluation measurements shown in Figure 9.

`HalfCheetah-v4`

| | Delay 3 | Delay 6 | Delay 9 | Delay 12 | Delay 15 | Delay 18 | Delay 21 | Delay 24 |
|---|---|---|---|---|---|---|---|---|
| 0% noise | 8159.28 ± 50.12 | **6027.13 ± 50.79** | 5778.11 ± 43.85 | **5961.74 ± 34.98** | **5862.30 ± 55.13** | 5730.15 ± 101.62 | **5773.66 ± 106.13** | 3902.62 ± 822.27 |
| 2.5% noise | **8281.45 ± 143.90** | 5981.20 ± 137.60 | 5244.27 ± 43.00 | 5325.65 ± 62.94 | 5544.09 ± 78.04 | 5284.89 ± 103.65 | 3740.87 ± 127.23 | 4158.25 ± 382.17 |
| 5% noise | 7121.49 ± 58.07 | 5399.96 ± 122.38 | 5010.51 ± 211.44 | 5201.90 ± 197.06 | 4992.80 ± 110.72 | 4988.70 ± 179.34 | 4778.82 ± 419.50 | **4516.67 ± 222.24** |
| 7.5% noise | 7839.51 ± 169.20 | 4883.81 ± 138.27 | 4665.25 ± 159.32 | 3069.77 ± 828.40 | 3893.18 ± 914.63 | 2575.36 ± 627.12 | 4128.39 ± 580.59 | 3071.16 ± 104.56 |
| 10% noise | 6459.33 ± 158.08 | 4376.52 ± 114.97 | 3543.70 ± 257.16 | 3996.73 ± 153.34 | 3721.40 ± 873.19 | 2610.57 ± 777.65 | 2517.84 ± 875.47 | 3641.14 ± 851.81 |
| 15% noise | 5500.11 ± 125.74 | 3841.51 ± 100.05 | 2946.32 ± 409.99 | 2805.37 ± 491.86 | 3483.07 ± 91.09 | 2122.89 ± 842.33 | 2363.06 ± 544.60 | 1830.24 ± 517.49 |
| 20% noise | 4929.20 ± 90.98 | 3192.02 ± 142.30 | 2903.07 ± 192.16 | 1881.11 ± 149.04 | 1907.26 ± 716.61 | 2042.48 ± 91.94 | 1776.34 ± 528.88 | 1300.96 ± 298.35 |

`Humanoid-v4`

| | Delay 3 | Delay 6 | Delay 9 | Delay 12 | Delay 15 | Delay 18 | Delay 21 | Delay 24 |
|---|---|---|---|---|---|---|---|---|
| 0.0% noise | 5265.75 ± 16.10 | 5416.97 ± 4.56 | 5483.19 ± 11.20 | 5283.45 ± 34.02 | 5383.76 ± 49.09 | **5207.48 ± 704.16** | **4836.92 ± 1019.75** | **3639.32 ± 1295.49** |
| 2.5% noise | 5292.31 ± 96.10 | 5462.32 ± 19.79 | 5385.21 ± 12.65 | 5392.97 ± 38.69 | **5456.68 ± 18.84** | 1692.16 ± 660.11 | 2199.76 ± 866.64 | 739.62 ± 198.24 |
| 5.0% noise | 5194.99 ± 8.40 | 5545.78 ± 46.43 | 5435.19 ± 40.38 | **5543.46 ± 19.15** | 5448.85 ± 77.24 | 3873.11 ± 2007.65 | 1038.41 ± 466.64 | 431.75 ± 154.26 |
| 7.5% noise | 5362.93 ± 10.58 | **5625.55 ± 30.05** | 5271.80 ± 24.04 | 5430.81 ± 33.14 | 1364.47 ± 391.72 | 1110.10 ± 1023.93 | 863.93 ± 490.55 | 346.29 ± 109.18 |
| 10.0% noise | **5386.11 ± 18.15** | 5403.12 ± 8.27 | 5322.27 ± 45.70 | 4167.17 ± 1831.92 | 1235.33 ± 563.62 | 1039.16 ± 747.36 | 545.63 ± 162.01 | 258.85 ± 116.50 |
| 15.0% noise | 5194.31 ± 44.01 | 5299.12 ± 29.14 | **5516.88 ± 42.33** | 1955.98 ± 1041.45 | 1001.01 ± 383.28 | 526.00 ± 235.22 | 378.92 ± 243.38 | 247.76 ± 135.52 |
| 20.0% noise | 5317.08 ± 29.38 | 5323.95 ± 58.41 | 4566.51 ± 1432.43 | 1561.88 ± 916.61 | 525.42 ± 236.13 | 490.00 ± 237.10 | 239.58 ± 214.93 | 297.82 ± 199.20 |

We therefore conclude that a fair evaluation of delayed RL should be done in environments with stochastic dynamics. Further practical evaluation of the effect that noise has on performance across different training algorithms is shown in Appendix B.2.

## B.2 FURTHER EVALUATION OF NOISE EFFECTS

To see how noise affects the best performing delay-aware algorithms, BPQL, VDPO, and ACDA, we evaluate their performance as the noise varies from 0% to 25% on the same set of chosen environments across the three delay processes. As explained in Appendix B.5, each measurement represents the best average return for a policy trained on the combination of environment, action noise, and delay process. The results for delay processes $GE_{1,23}$, $GE_{4,32}$, and MM1 are shown in Tables 3, 4, and 5, respectively.

Note that VDPO uses a fixed seed when resetting an environment. Therefore, VDPO has a standard deviation of 0 when evaluating without action noise, as all sampled trajectories are deterministic. This is due to us using the original VDPO implementation with as few modifications as possible, as further explained in Appendix B.5.

The results show a trend that ACDA performs even better as the noise increases. As ACDA adapts to the sampled delays, this performance increase is expected because noisy environments can still perform well for lower delays, as shown in Figure 9.

There are some outliers in the results, where the performance is slightly better for a higher action noise. We believe that these are due to randomness and that they would not be present if we sampled more trajectories during evaluation and averaged across multiple trained policies.

Table 3: Best returns from the $GE_{1,23}$ delay process over different noise.

| Ant-v4 | | | | | |
| --- | --- | --- | --- | --- | --- |
| 0% noise | 5% noise | 10% noise | 15% noise | 20% noise | 25% noise |
| BPQL 3736.70 ± 108.62 | 2691.88 ± 129.84 | 1421.78 ± 297.93 | 640.97 ± 316.38 | 69.69 ± 75.58 | 1.31 ± 18.92 |
| VDPO 3492.27 ± 0.00 | 2163.00 ± 53.04 | 1162.88 ± 603.92 | 644.82 ± 373.70 | 296.89 ± 131.34 | 34.66 ± 39.05 |
| ACDA **4719.08 ± 658.29** | **4112.78 ± 818.44** | **2780.25 ± 761.75** | **1209.94 ± 832.61** | **536.10 ± 400.29** | **192.79 ± 105.91** |

| Humanoid-v4 | | | | | |
| --- | --- | --- | --- | --- | --- |
| 0% noise | 5% noise | 10% noise | 15% noise | 20% noise | 25% noise |
| BPQL 2462.64 ± 1341.26 | 585.19 ± 163.49 | 261.12 ± 125.97 | 365.19 ± 172.41 | 298.31 ± 200.46 | 237.62 ± 161.17 |
| VDPO 464.39 ± 0.00 | 417.25 ± 210.09 | 312.26 ± 145.72 | 285.21 ± 186.51 | 276.49 ± 174.40 | 325.55 ± 130.45 |
| ACDA **4842.13 ± 861.55** | **4608.76 ± 1084.52** | **3751.03 ± 1552.10** | **3638.29 ± 1849.70** | **3852.50 ± 1237.91** | **1597.85 ± 1143.37** |

| HalfCheetah-v4 | | | | | |
| --- | --- | --- | --- | --- | --- |
| 0% noise | 5% noise | 10% noise | 15% noise | 20% noise | 25% noise |
| BPQL 4176.16 ± 897.01 | 4320.20 ± 1028.52 | 3908.33 ± 77.24 | 1810.59 ± 635.08 | 1899.01 ± 89.66 | 1206.78 ± 154.01 |
| VDPO 4976.10 ± 0.00 | 3144.23 ± 1156.52 | 2240.86 ± 591.26 | 1664.11 ± 144.79 | 1049.28 ± 167.12 | 799.84 ± 212.99 |
| ACDA **6087.67 ± 1142.66** | **5984.25 ± 1885.78** | **5838.64 ± 724.34** | **4656.72 ± 693.20** | **4446.73 ± 627.70** | **2783.35 ± 194.57** |

| Hopper-v4 | | | | | |
| --- | --- | --- | --- | --- | --- |
| 0% noise | 5% noise | 10% noise | 15% noise | 20% noise | 25% noise |
| BPQL 3176.22 ± 48.33 | 1328.71 ± 937.67 | 549.60 ± 541.58 | 232.25 ± 176.51 | 135.24 ± 100.37 | 111.19 ± 92.11 |
| VDPO **3477.61 ± 0.00** | 709.20 ± 522.01 | 181.60 ± 60.08 | 150.74 ± 94.33 | 96.75 ± 54.71 | 77.14 ± 73.00 |
| ACDA 2381.98 ± 1226.41 | **2094.65 ± 944.20** | **2344.23 ± 1167.03** | **1330.55 ± 895.65** | **1636.10 ± 1134.22** | **1057.21 ± 949.42** |

| Walker2d-v4 | | | | | |
| --- | --- | --- | --- | --- | --- |
| 0% noise | 5% noise | 10% noise | 15% noise | 20% noise | 25% noise |
| BPQL 1287.71 ± 754.84 | 1215.91 ± 776.93 | 652.90 ± 501.11 | 316.01 ± 216.53 | 595.48 ± 668.53 | 314.27 ± 417.49 |
| VDPO 2005.31 ± 0.00 | 846.88 ± 808.67 | 810.89 ± 1173.36 | 283.58 ± 334.85 | 186.02 ± 307.16 | 199.08 ± 375.33 |
| ACDA **4030.01 ± 82.46** | **3863.59 ± 232.52** | **4295.73 ± 128.36** | **4045.24 ± 51.37** | **3199.49 ± 231.87** | **3234.59 ± 623.28** |

Table 4: Best returns from the $GE_{4,32}$ delay process over different noise.

| Ant-v4 | | | | | |
| --- | --- | --- | --- | --- | --- |
| 0% noise | 5% noise | 10% noise | 15% noise | 20% noise | 25% noise |
| BPQL 3523.32 ± 146.78 | 2509.52 ± 117.37 | **1456.09 ± 299.72** | 547.34 ± 237.21 | 67.78 ± 122.29 | 0.28 ± 8.70 |
| VDPO **3574.87 ± 0.00** | 2266.99 ± 90.89 | 1167.24 ± 473.41 | **647.07 ± 346.39** | **244.78 ± 127.86** | 26.36 ± 40.46 |
| ACDA 2658.50 ± 285.82 | **2866.93 ± 1172.46** | 1406.31 ± 712.09 | 547.27 ± 329.31 | 138.69 ± 94.21 | **29.26 ± 32.36** |

| Humanoid-v4 | | | | | |
| --- | --- | --- | --- | --- | --- |
| 0% noise | 5% noise | 10% noise | 15% noise | 20% noise | 25% noise |
| BPQL 876.89 ± 69.04 | 276.63 ± 131.70 | 301.68 ± 134.63 | 254.22 ± 131.77 | 201.02 ± 109.72 | 195.88 ± 123.20 |
| VDPO 442.03 ± 0.00 | 280.72 ± 169.85 | 262.74 ± 131.86 | 233.62 ± 145.57 | 206.56 ± 159.37 | 194.34 ± 113.40 |
| ACDA **3877.77 ± 1776.04** | **3725.59 ± 1513.38** | **3454.60 ± 1567.23** | **3092.70 ± 1752.58** | **1043.06 ± 339.16** | **649.32 ± 270.74** |

| HalfCheetah-v4 | | | | | |
| --- | --- | --- | --- | --- | --- |
| 0% noise | 5% noise | 10% noise | 15% noise | 20% noise | 25% noise |
| BPQL 4894.08 ± 99.20 | 2136.36 ± 547.04 | 2019.46 ± 758.99 | 1785.40 ± 655.73 | 1920.19 ± 126.28 | 1286.61 ± 141.72 |
| VDPO **5059.93 ± 0.00** | 3664.30 ± 929.25 | 1923.50 ± 379.20 | 1510.93 ± 435.71 | 1177.83 ± 283.21 | 790.87 ± 174.17 |
| ACDA 4203.13 ± 279.18 | **4231.15 ± 333.69** | **3239.61 ± 199.78** | **3149.08 ± 367.98** | **3218.57 ± 154.02** | **1826.06 ± 214.04** |

| Hopper-v4 | | | | | |
| --- | --- | --- | --- | --- | --- |
| 0% noise | 5% noise | 10% noise | 15% noise | 20% noise | 25% noise |
| BPQL 2668.14 ± 711.84 | 433.29 ± 381.79 | 190.79 ± 135.89 | 120.03 ± 137.44 | 76.54 ± 66.87 | 77.79 ± 72.13 |
| VDPO **3403.46 ± 0.00** | 330.44 ± 263.74 | 138.13 ± 128.53 | 86.52 ± 81.11 | 77.02 ± 86.06 | 59.12 ± 60.31 |
| ACDA 2947.10 ± 929.71 | **1727.79 ± 959.50** | **1434.94 ± 805.19** | **1814.74 ± 1187.29** | **1128.22 ± 1015.20** | **532.97 ± 630.97** |

| Walker2d-v4 | | | | | |
| --- | --- | --- | --- | --- | --- |
| 0% noise | 5% noise | 10% noise | 15% noise | 20% noise | 25% noise |
| BPQL 1352.44 ± 328.51 | 875.09 ± 747.72 | 343.62 ± 314.86 | 275.07 ± 207.17 | 191.56 ± 383.58 | 134.20 ± 125.93 |
| VDPO 1779.39 ± 0.00 | 344.73 ± 316.82 | 123.18 ± 160.70 | 147.64 ± 258.22 | 73.78 ± 142.23 | 56.70 ± 148.33 |
| ACDA **3945.33 ± 148.28** | **1840.58 ± 386.78** | **1409.42 ± 281.81** | **1322.32 ± 305.35** | **1149.82 ± 345.44** | **850.48 ± 172.82** |

Table 5: Best returns from the MM1 delay process over different noise.

| `Ant-v4` | | | | | | |
|---|---|---|---|---|---|---|
| | 0% noise | 5% noise | 10% noise | 15% noise | 20% noise | 25% noise |
| BPQL | **3717.34 ± 125.79** | **3074.17 ± 106.78** | 1680.23 ± 307.70 | **764.00 ± 295.30** | 123.27 ± 122.10 | 4.06 ± 13.70 |
| VDPO | 3638.48 ± 9.30 | 2528.67 ± 144.63 | 1319.55 ± 537.44 | 709.82 ± 254.73 | **334.61 ± 133.75** | 32.48 ± 27.59 |
| ACDA | 2593.23 ± 88.19 | 2898.46 ± 838.07 | **1941.68 ± 567.39** | 724.43 ± 487.25 | 306.75 ± 319.66 | **76.13 ± 110.11** |
| `Humanoid-v4` | | | | | | |
| | 0% noise | 5% noise | 10% noise | 15% noise | 20% noise | 25% noise |
| BPQL | 2926.45 ± 1042.65 | 5435.29 ± 68.34 | 733.25 ± 410.45 | 554.66 ± 205.97 | 423.77 ± 227.06 | 406.50 ± 195.39 |
| VDPO | 762.75 ± 0.00 | 720.73 ± 634.35 | 544.09 ± 424.14 | 423.67 ± 252.84 | 526.95 ± 394.22 | 354.99 ± 129.91 |
| ACDA | **5238.97 ± 332.60** | **5805.60 ± 23.04** | **5548.29 ± 39.58** | **5343.60 ± 227.82** | **1752.02 ± 616.85** | **1585.00 ± 538.02** |
| `HalfCheetah-v4` | | | | | | |
| | 0% noise | 5% noise | 10% noise | 15% noise | 20% noise | 25% noise |
| BPQL | 5627.41 ± 64.54 | 4660.93 ± 448.10 | 2291.63 ± 857.39 | 2171.03 ± 626.53 | 2026.23 ± 523.82 | 1574.84 ± 167.54 |
| VDPO | 4684.17 ± 0.00 | 3831.96 ± 960.07 | 2454.03 ± 415.56 | 1896.91 ± 419.64 | 1277.00 ± 244.75 | 939.40 ± 251.26 |
| ACDA | **6309.62 ± 356.30** | **5898.36 ± 409.10** | **4998.40 ± 414.15** | **4173.81 ± 96.85** | **2547.28 ± 106.75** | **2243.67 ± 122.99** |
| `Hopper-v4` | | | | | | |
| | 0% noise | 5% noise | 10% noise | 15% noise | 20% noise | 25% noise |
| BPQL | 3130.47 ± 29.44 | 3035.66 ± 103.80 | 1106.35 ± 490.33 | 397.27 ± 249.50 | 319.40 ± 198.93 | 196.96 ± 105.23 |
| VDPO | **3797.33 ± 0.00** | 1459.88 ± 933.11 | 389.41 ± 247.52 | 201.10 ± 121.99 | 156.50 ± 87.59 | 152.03 ± 96.74 |
| ACDA | 3029.85 ± 565.47 | **3122.53 ± 417.37** | **2245.07 ± 1166.01** | **2250.64 ± 901.67** | **1079.67 ± 690.26** | **1189.84 ± 677.60** |
| `Walker2d-v4` | | | | | | |
| | 0% noise | 5% noise | 10% noise | 15% noise | 20% noise | 25% noise |
| BPQL | 3815.63 ± 39.18 | 3547.73 ± 133.51 | 2182.64 ± 763.72 | 883.42 ± 187.43 | 691.82 ± 431.84 | 758.42 ± 558.27 |
| VDPO | **5202.62 ± 0.00** | 2144.25 ± 1650.85 | 1416.45 ± 1845.96 | 1538.01 ± 1867.52 | 1227.54 ± 1220.62 | 637.52 ± 1228.54 |
| ACDA | 4485.74 ± 55.72 | **4562.33 ± 87.98** | **3653.42 ± 35.72** | **3892.52 ± 52.42** | **3036.04 ± 631.82** | **1608.60 ± 877.13** |

## B.3 EVALUATED DELAY DISTRIBUTIONS

As mentioned in Section 5, we evaluate on delay processes following the Gilbert-Elliot and M/M/1 models. We formally define these processes in this section, as well as their conservative and optimistic worst-case delay assumptions (high and low CDA). Appendix E.2 and E.3 evaluate the performance under high and low CDA, respectively.

We consider $GE_{1,23}$ and $GE_{4,32}$, two Gilbert-Elliot models. These are Markovian processes alternating between two states, a good state $s_{good}$ and a bad state $s_{bad}$, as illustrated in Figure 10. We describe the models in Table 6 as a two-state Markov process, where they initially start in the good state. The notation $D(d|s)$ is used to describe the probability of sampling the delay $d$ in the Gilbert-Elliot state $s$. We set the opportunistic low CDA to be the maximum delay that can be sampled in the $s_{good}$ state.

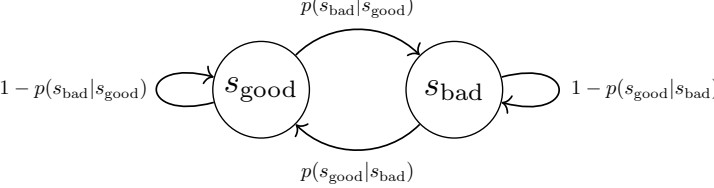

Figure 10: Illustration of transitions in a Gilbert-Elliot model.

We plot the distribution histogram and time series over 1000 samples of the Gilbert-Elliot processes in Figure 11.

The M/M/1 queue process is described by simulating an M/M/1 queue according to the pseudocode in Algorithm 4. We set the arrival rate $\lambda_{arrive} = 0.33$ and the service rate $\lambda_{service} = 0.75$. The arrivals and departures are dictated by independent Poisson processes parametrized by these values (e.g., the time between two arrivals is a r.v. with an exponential distribution of mean $\lambda_{arrive}$). Note that there is no upper bound on this delay process, and it is therefore impossible to convert this to a constant-delay MDP through Constant Delay Augmentation (CDA). We can still apply the CDA

Table 6: Description of the Gilbert-Elliot delay processes used during evaluation.

| **Property** | $GE_{1,23}$ | $GE_{4,32}$ |
|---|---|---|
| $p(s_{\text{bad}}\|s_{\text{good}})$ | $\dfrac{1}{125}$ | $\dfrac{1}{250}$ |
| $p(s_{\text{good}}\|s_{\text{bad}})$ | $\dfrac{1}{20}$ | $\dfrac{1}{32}$ |
| $D(d\|s_{\text{good}})$ | $\Pr[d=1] = \frac{15}{16}$ $\Pr[d=2] = \frac{1}{16}$ | $\Pr[d=4] = 1$ |
| $D(d\|s_{\text{bad}})$ | $\Pr[d=22] = \frac{3}{11}$ $\Pr[d=23] = \frac{5}{11}$ $\Pr[d=24] = \frac{3}{11}$ | $\Pr[d=32] = 1$ |
| Low CDA | 2 | 4 |
| High CDA | 24 | 32 |

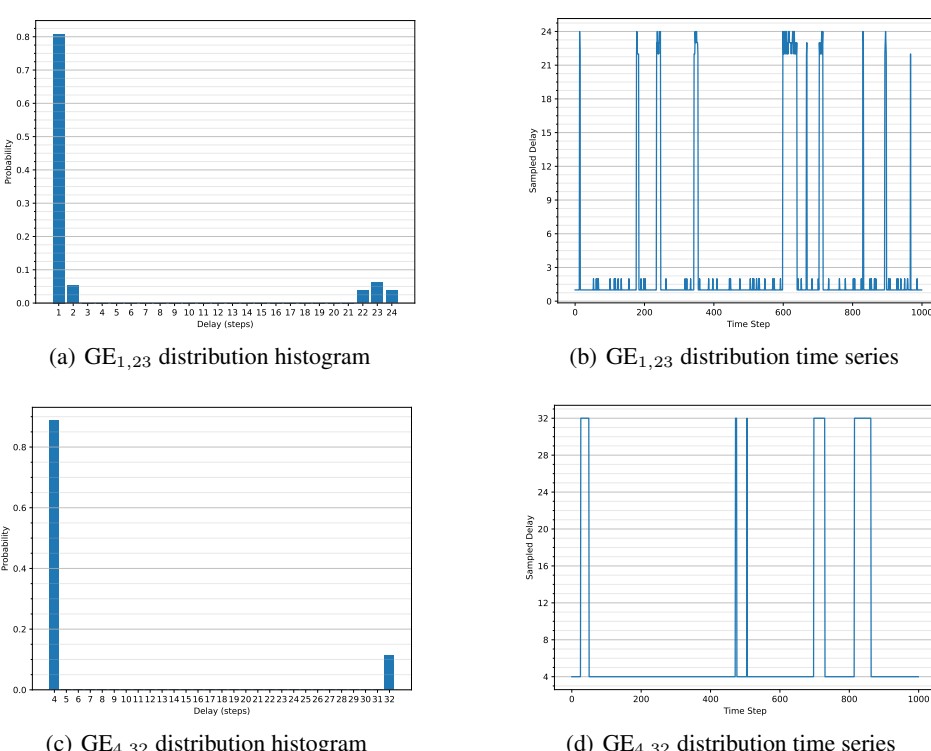

(a) $GE_{1,23}$ distribution histogram

(b) $GE_{1,23}$ distribution time series

(c) $GE_{4,32}$ distribution histogram

(d) $GE_{4,32}$ distribution time series

Figure 11: The Gilbert-Elliot delay processes.

conversion from Appendix A to apply constant-delay methodologies on this delay process, though they are no longer operating on an MDP, as the true delay may exceed the assumed upper bound.

**Algorithm 4** M/M/1 Queue Delay Generator

---

**Initial State:** $t_{\text{arrival}} \sim \text{Exp}(\cdot | \lambda_{\text{arrive}})$   (Time of arrival of the first packet)
                   $t_{\text{service}} \leftarrow \emptyset$          (Cannot serve anything yet)
                   $Q \leftarrow \text{FIFOqueue}()$     (Empty queue initially)

1: **procedure** SAMPLEDELAY
2:     **if** $t_{\text{service}} = \emptyset$ **then**
3:         $t \leftarrow t_{\text{arrival}}$
4:         $Q.\text{insert}(t)$
5:         $t_{\text{arrival}} \sim \text{Exp}(\cdot | \lambda_{\text{arrive}}) + t$
6:         $t_{\text{service}} \sim \text{Exp}(\cdot | \lambda_{\text{service}}) + t$
7:     **while** $t_{\text{arrival}} < t_{\text{service}}$ **do**
8:         $t \leftarrow t_{\text{arrival}}$
9:         $Q.\text{insert}(t)$
10:       $t_{\text{arrival}} \sim \text{Exp}(\cdot | \lambda_{\text{arrive}}) + t$
11:     $t \leftarrow t_{\text{service}}$
12:     $t_{\text{inserted}} \leftarrow Q.\text{pop}()$
13:     $d \leftarrow \lceil t - t_{\text{inserted}} \rceil$
14:     **if** $Q.\text{isempty}()$ **then**
15:         $t_{\text{service}} \leftarrow \emptyset$
16:     **else**
17:         $t_{\text{service}} \sim \text{Exp}(\cdot | \lambda_{\text{service}}) + t$
18:     **return** $d$

---

We plot delays of the M/M/1 queue in Figure 12. We set the conservative delay (high CDA) to be 16, and the opportunistic delay (low CDA) to be 4 for the M/M/1 queue.

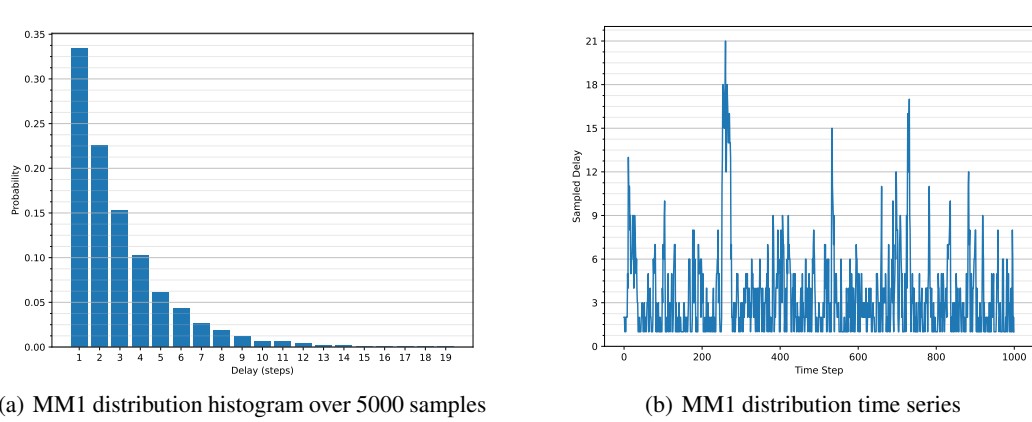

(a) MM1 distribution histogram over 5000 samples          (b) MM1 distribution time series

Figure 12: Delays from an M/M/1 queue when $\lambda_{\text{arrive}} = 0.33$ and $\lambda_{\text{service}} = 0.75$.

### B.4 HYPERPARAMETERS AND NEURAL NETWORK STRUCTURE

Hyperparameters used for training the policy $\pi_\theta$ and the critic $Q_\phi$ in ACDA follow the common learning rates used for SAC, and are therefore shared between SAC, BPQL, and ACDA. We show these together with the model hyperparameters used for ACDA in Tables 7(a) and 7(b). The hyperparameters for the model share the same replay size and batch size. We use slightly different parameters for the model when learning the dynamics on the 2D environments (`HalfCheetah-v4`, `Hopper-v4`, and `Walker2d-v4`) and the 3D environments (`Ant-v4` and `Humanoid-v4`).

The $\pi_\theta, Q_\phi$, and EMBED$_\omega$ networks are all implemented as MLPs with 2 hidden layers of dimension 256 each. The policy outputs the mean and standard deviation of a Gaussian distribution, which is put through tanh and scaled to exactly cover the action space. The EMIT$_\omega$ network consists of 2

Table 7(a): SAC Hyperparameters

| Parameter | Value |
|---|---|
| Policy ($\theta$) learning rate | $3 \cdot 10^{-4}$ |
| Critic ($\phi$) Learning rate | $3 \cdot 10^{-4}$ |
| Temperature ($\alpha$) learning rate | $3 \cdot 10^{-4}$ |
| Starting temperature ($\alpha$) | 0.2 |
| Temperature threshold $\mathcal{H}$ | $-dim(A)$ |
| Target smoothing coefficient | 0.005 |
| Replay buffer size | $10^6$ |
| Discount $\gamma$ | 0.99 |
| Minibatch size | 256 |
| Optimizer (policy, critic, temp.) | Adam |
| Activation (policy, critic) | ReLU |

Table 7(b): Model Hyperparameters

| Parameter | Value (2D) | Value (3D) |
|---|---|---|
| Model ($\omega$) learning rate | $10^{-4}$ | $5 \cdot 10^{-5}$ |
| Model training window $n$ | 16 | 16 |
| Latent GRU dimensionality | 384 | 512 |
| Angle clamping | Yes | No |
| Optimizer | Adam | Adam |
| Activation | ClipSiLU | ClipSiLU |

common layers of dimension 256 each, with additional "head" layers of dimension 256 each for outputting the mean and standard deviation of a Gaussian distribution.

Both MLPs (EMBED$_\omega$ and EMIT$_\omega$) in the model make use of a clipped version of SiLU (Hendrycks & Gimpel, 2023) as their activation function, where ClipSiLU$(x) = $ SiLU$(\max(-20, x))$. We found that the use of ClipSiLU significantly improved the model performance. Earlier experiments with models using ReLU activation did not manage to achieve good performance when used with ACDA.

The angle clamping mentioned in Table 7(b) constrains all components of the state space that represent an angle to reside in the range $[-\pi, \pi)$. We only apply this to 2D environments.

The model is trained using sub-trajectories $T_n = (s_t, a_t, s_{t+1}, a_{t+1}, \ldots, s_{t+n-1}, a_{t+n-1}, s_{t+n})$, where $n$ is the model training window in Table 7(b). We optimize the model parameters $\omega$ with sub-trajectories using the following equation

$$\nabla_\omega \mathbb{E}_{T_n \sim \mathcal{R}} \left[ \frac{1}{n+1} \sum_{k=0}^{n} -\log \text{EMIT}_\omega \left( s_{t+k} | \text{STEP}_\omega^k (\text{EMBED}_\omega(s_t), a_t, \ldots, a_{t+k-1}) \right) \right] \quad (8)$$

where for $k = 0$, we just evaluate the embedder as EMIT$_\omega(s_t | \text{EMBED}_\omega(s_t))$.

We set the prediction length $L$ in ACDA to the assumed upper bound of the delay process in all benchmarks. For each baseline, the horizon $h$ of the action buffer is also set to the assumed upper bound of the delay process, unless the baseline uses constant-delay augmentation (CDA), in which the assumed delay used for the CDA is also used for the action buffer horizon $h$.

B.5 PRACTICAL EVALUATION DETAILS

The evaluation methodology used for all performance measurements is that of the maximum average return for a trained policy. A policy is trained on a total of $10^6$ steps from the environment. Every 10000 training steps, all network weights are frozen, and the policy is evaluated by sampling 10 trajectories from the environment. These trajectories are discarded after evaluation and not used for training. Each policy is evaluated on the same underlying environment, delay process, and noise process as it was trained on. The average return (unweighted average sum of rewards $\frac{1}{10} \sum_{i=1}^{10} \sum_{(s_t, a_t, r_t) \in \tau_i} r_t$) is reported as the performance of the policy. For time series plots with an x-axis *Steps* and a y-axis *Return*, we refer to the average return evaluated after that number of training steps. The maximum average return, as shown in the tables, is the maximum evaluated average return achieved at any point during the training process.

The training algorithms for SAC, SAC /w CDA, BPQL, and ACDA are implemented in our own framework. Dreamer, DCAC, and VDPO are evaluated using the authors' own implementations [5][6][7], with small modifications to accommodate our delay processes, noise processes, and the interaction layer.

In our framework, we use PyTorch for deep learning functionality and Gymnasium for RL functionality. The interaction layer is implemented as a Gymnasium environment wrapper, based on the formalism described in Appendix C, that extends the Gymnasium API to support action and observation packets. The constant-delay augmentation (CDA) in Appendix A is implemented as a wrapper on top of the interaction layer wrapper, reducing the API back to the original Gymnasium API. We implement ACDA to explicitly make use of the extended interaction layer definitions, whereas we implement BPQL and SAC w/ CDA to operate directly on the regular Gymnasium API using the CDA wrapper. A pass-through wrapper for the interaction layer is used for evaluating SAC (without CDA), where the action packet is filled with the provided action.

All dependencies for our framework are provided as conda YAML files.

For VDPO we reuse the code artifact from their original article. Modifications to their implementation include the addition of action noise, our interaction layer wrapper (with CDA), and additional statistical reporting. These modifications are documented in the artifact. The reason for adding our own interaction layer wrapper to VDPO is to capture effects from the M/M/1 delay process when the maximum delay assumption is violated.

The Dreamer baseline uses the Dreamer v3 implementation. We add the action noise and the interaction layer wrappers on top of the environment, with the pass-through wrapper used to allow it to operate using the regular Gymnasium API. This follows a similar procedure used by Karamzade et al. (2024).

As DCAC already has a framework for random delays, we do not add our interaction layer wrapper to their implementation. Instead, we only add the action noise and the delay processes. To adapt our single delay to their split observation and action delay, we set the observation delay to 0 and set the sampled delay as the action delay. This matches the formalism in the interaction layer framework, as we only consider the full round-trip delay.

Each benchmark, meaning a single algorithm training on a single environment with a single delay process, is run using a single Nvidia A40 GPU and 16 CPU cores. ACDA and VDPO benchmarks take around 18-24 hours each to complete. BPQL and SAC benchmarks take around 6 hours each to complete. Each Dreamer and DCAC benchmark takes roughly 3-5 days to complete.

---

[5]https://github.com/danijar/dreamerv3
[6]https://github.com/rmst/rlrd
[7]https://github.com/QingyuanWuNothing/VDPO

## C FORMAL DESCRIPTION OF THE INTERACTION LAYER POMDP

This section describes the POMDP of the interaction layer introduced in Section 3.3. The POMDP formalizes the interaction between the agent and the interaction layer that wraps the underlying system. We assume that the underlying system can be described by an MDP $\mathcal{M} = (S, A, r, p, \mu)$ where $S$ is the state space, $A$ the action space, $r(s, a)$ the reward function, $p(s'|s, a)$ the transition distribution, and $\mu(s)$ the initial state distribution.

The interaction layer wraps the MDP $\mathcal{M}$, where the interaction delay is described by the delay process $D$. Samples $d \sim D(\cdot)$ are not necessarily independent. To fully define the POMDP of the interaction layer, we also need the action buffer horizon $h$ as well as the default action $a_{\text{init}}$. Given this information, the POMDP is described as the tuple $\mathcal{P} = (\boldsymbol{S}, \boldsymbol{A}, \boldsymbol{p}, \boldsymbol{r}, \boldsymbol{\mu}, \Omega, O)$. We use the notation $\boldsymbol{s} \in \boldsymbol{S}$, $\boldsymbol{a} \in \boldsymbol{A}$, and $\boldsymbol{o} \in \Omega$ to denote members of these sets. We also refer to items $\boldsymbol{a} \in \boldsymbol{A}$ as *action packets* and items $\boldsymbol{o} \in \Omega$ as *observation packets*.

An observation is described as the tuple $\boldsymbol{o} = (t, s, \boldsymbol{b}, \delta, c)$. This describes the state at the interaction layer at time $t$, where

- $t$ is the time at the interaction layer when the observation was generated,
- $s$ the underlying system state observed at the same time step,
- $\boldsymbol{b} = (b_1, b_2, \ldots, b_h)$ are action buffer contents at time $t$ ($b_1$ is immediately applied to $s$),
- $\delta$ is the delay of the action packet used to update the action buffer $\boldsymbol{b}$, and
- $c$ is the number of time steps without a new action packet replacing the action buffer contents.

When referring to the state of the action buffer at different time steps, e.g., $\boldsymbol{b}_t$ and $\boldsymbol{b}_{t+1}$, we use the notation $b_{t,i}$ and $b_{t+1,i}$ to refer to the $i$-th action in $\boldsymbol{b}_t$ and $\boldsymbol{b}_{t+1}$, respectively.

Delays are referred to using different notations, $d_t$ or $\delta_t$, depending on the context:

- $d_t$ is the unobserved delay sampled at time $t$. This is the delay of the action packet $\boldsymbol{a}_t$.
- $\delta_t$ is the delay recovered in hindsight. See description of the observation packet above for more information. This hindsight delay is related to $d_t$ by the equation $\delta_t = d_{t-(\delta_t+c_t)}$.

The individual components of $\mathcal{P}$ are defined in Equations 9-18:

$$\text{Action space } \boldsymbol{A} = \mathbb{N} \times \bigcup_{k=1}^{\infty} A^{k \times h} \tag{9}$$

$$\text{Observation space } \Omega = \mathbb{N} \times S \times A^h \times \mathbb{Z}^+ \times \mathbb{N} \tag{10}$$

$$\text{State space } \boldsymbol{S} = \Omega \times 2^{(\mathbb{Z}^+ \times \boldsymbol{A})} \tag{11}$$

$$\text{Initial state distribution } \boldsymbol{\mu}(\boldsymbol{s}) = \begin{cases} \mu(s) & \text{if } \boldsymbol{s} = ((0, s, (a_{\text{init}}, \ldots), 1, 0), \emptyset) \\ 0 & \text{otherwise} \end{cases} \tag{12}$$

$$\text{Observation distribution } O(\boldsymbol{o}|\boldsymbol{s}) = \begin{cases} 1 & \text{if } \boldsymbol{s} = (\boldsymbol{o}, \mathcal{I}) \\ 0 & \text{otherwise} \end{cases} \tag{13}$$

$$\text{Reward function } \boldsymbol{r}(\boldsymbol{s}_t, \boldsymbol{a}_t) = r(s_t, b_1) \tag{14}$$
$$\text{where } \boldsymbol{s}_t = ((t, s_t, (b_1, b_2, \ldots, b_h), \delta_t, c_t), \mathcal{I}_t)$$

Equation 11 defines states as tuples $\boldsymbol{s}_t = (\boldsymbol{o}_t, \mathcal{I}_t)$, where $\boldsymbol{o}_t$ is the state of the interaction layer (observable) and $\mathcal{I}_t$ is the set of action packets in transit (not observable). The transit set $\mathcal{I}_t \in 2^{(\mathbb{Z}^+ \times \boldsymbol{A})}$ contains tuples of action packets and their arrival time.

Note that, by the definition of $\boldsymbol{\mu}$ in Equation 12, we can always check if the action buffer contains the initial actions by $t - (\delta_t + c_t) < 0$. This holds until the first action packet is received.

The transition dynamics $\boldsymbol{p}(\boldsymbol{s}_{t+1}|\boldsymbol{s}_t, \boldsymbol{a}_t)$ are described in Equation 18 below. While this is simple to describe in text and with examples, it becomes complicated to define formally. We first define a couple of auxiliary functions below to help define the transition dynamics. We define the function

TRANSMIT$(\mathcal{I}_t, \boldsymbol{a}_t, d)$ that adds the action $\boldsymbol{a}_t$ with delay $d$ to the transit set, together with the $\min \mathcal{I}$ and $\min_t \mathcal{I}$ operations to get the action packet with the nearest arrival:

$$\text{TRANSMIT}(\mathcal{I}_t, \boldsymbol{a}_t, d) = \{(t+d, \boldsymbol{a}_t)\} \cup \{(t', \boldsymbol{a}') \in \mathcal{I}_t : t' < t + d\} \tag{15}$$

$$\min_t \mathcal{I} = \min\{t' : (t', \boldsymbol{a}') \in \mathcal{I}\} \tag{16}$$

$$\min \mathcal{I} = \begin{cases} \emptyset & \text{if } \mathcal{I} = \emptyset \\ (t', \boldsymbol{a}') \in \mathcal{I} & \text{if } t' = \min_t \mathcal{I} \end{cases} \tag{17}$$

One aspect of the behavior of the interaction layer, modeled by the TRANSMIT$(\mathcal{I}_t, \boldsymbol{a}_t, d)$ function in Equation 15, is that outdated action packets arriving at the interaction layer will be discarded. For example, if $\boldsymbol{a}_t$ has delay $d_t = 4$, and $\boldsymbol{a}_{t+1}$ has delay $d_{t+1} = 2$, then $\boldsymbol{a}_{t+1}$ will arrive at time $t + 3$, whereas $\boldsymbol{a}_t$ will arrive after at time $t + 4$. When $\boldsymbol{a}_t$ arrives, the interaction layer will see that the contents of the action buffer are based on information from $\boldsymbol{o}_{t+1}$, whereas the action packet $\boldsymbol{a}_t$ is based on information from $\boldsymbol{o}_t$. Therefore, $\boldsymbol{a}_t$ is considered outdated and will be discarded. Also note that a consequence of this is that $d_t$ will never be observed, not even in hindsight.

Using these functions, we define the transition probabilities below in Equation 18. The probabilities themselves are simple to describe as $p(s_{t+1}|s_t, b_1) \times D(d)$; the complexity arises from checking that the new POMDP state is compatible with the possible sampled delays. The first case covers when no new action packet arrives at the interaction layer at time $t + 1$, the second case is when the received action packet $\boldsymbol{a}_t$ has too few rows in the matrix to update the action buffer for the sampled delay $d$, and the third case is when a received action packet is used to update the action buffer.

$$\boldsymbol{p}(\boldsymbol{s}_{t+1}|\boldsymbol{s}_t, \boldsymbol{a}_t) = \begin{cases} p(s_{t+1}|s_t, b_1) \cdot D(d) & \text{if } \mathcal{I}_{t+1} = \text{TRANSMIT}(\mathcal{I}_t, \boldsymbol{a}_t, d) \wedge \min_t \mathcal{I}_{t+1} > t + 1 \wedge \\ & \quad c_{t+1} = c_t + 1 \wedge \delta_{t+1} = \delta_t \wedge \\ & \quad \boldsymbol{b}_{t+1} = (b_2, b_3, \ldots, b_{h-1}, b_h, b_h) \\[1em] p(s_{t+1}|s_t, b_1) \cdot D(d) & \text{if } \mathcal{I}_{t+1} = \mathcal{I}_{\text{cand}} \setminus \{\min \mathcal{I}_{\text{cand}}\} \wedge \min_t \mathcal{I}_{\text{cand}} = t + 1 \wedge \\ & \quad (t + 1 - u) > L \wedge c_{t+1} = c_t + 1 \wedge \delta_{t+1} = \delta_t \wedge \\ & \quad \boldsymbol{b}_{t+1} = (b_2, b_3, \ldots, b_{h-1}, b_h, b_h) \\[0.5em] & \quad \text{where } \mathcal{I}_{\text{cand}} = \text{TRANSMIT}(\mathcal{I}_t, \boldsymbol{a}_t, d) \\ & \quad \quad (t + 1, \boldsymbol{a}_u) = \min \mathcal{I}_{\text{cand}} \\ & \quad \quad (u, M^u) = \boldsymbol{a}_u \\ & \quad \quad M^u \in A^{L \times h} \\[1em] p(s_{t+1}|s_t, b_1) \cdot D(d) & \text{if } \mathcal{I}_{t+1} = \mathcal{I}_{\text{cand}} \setminus \{\min \mathcal{I}_{\text{cand}}\} \wedge \min_t \mathcal{I}_{\text{cand}} = t + 1 \wedge \\ & \quad (t + 1 - u) \leq L \wedge c_{t+1} = 0 \wedge \delta_{t+1} = (t + 1 - u) \wedge \\ & \quad \boldsymbol{b}_{t+1} = M^u_{(t+1-u)} \\[0.5em] & \quad \text{where } \mathcal{I}_{\text{cand}} = \text{TRANSMIT}(\mathcal{I}_t, \boldsymbol{a}_t, d) \\ & \quad \quad (t + 1, \boldsymbol{a}_u) = \min \mathcal{I}_{\text{cand}} \\ & \quad \quad (u, M^u) = \boldsymbol{a}_u \\ & \quad \quad M^u \in A^{L \times h} \\[1em] 0 & \text{otherwise} \end{cases} \tag{18}$$

$$\begin{aligned} \text{where } \boldsymbol{s}_t &= (\boldsymbol{o}_t, \mathcal{I}_t) \\ \boldsymbol{s}_{t+1} &= (\boldsymbol{o}_{t+1}, \mathcal{I}_{t+1}) \\ \boldsymbol{o}_t &= (t, s_t, \boldsymbol{b}_t, \delta_t, c_t) \\ \boldsymbol{o}_{t+1} &= (t + 1, s_{t+1}, \boldsymbol{b}_{t+1}, \delta_{t+1}, c_{t+1}) \\ \boldsymbol{b}_t &= (b_1, b_2, \ldots, b_{h-2}, b_{h-1}, b_h) \end{aligned}$$

# D  DETAILED MODEL DESCRIPTION

This section provides formal definitions of the model introduced in Section 4.2, as well as a detailed definition of the training algorithm presented in Section 4.3. Appendix D.1 presents the formal model definition. Appendix D.2 presents the full training algorithm.

We use the variables $\theta$, $\phi$, and $\omega$ to denote the parameters of the policy, critic, and model, respectively. In practice, these are large vectors of real numbers where different parts of the vector contain the parameters for components in a deep neural network.

## D.1  MODEL COMPONENTS AND OBJECTIVE

The primary purpose of the model is to overcome the limitation on fixed-size inputs of MLPs. The idea is that, instead of generating actions directly with the augmented state input:

$$a_{t+k} \sim \pi_\theta(\cdot|s_t, a_t, a_{t+1}, \ldots, a_{t+k-1}), \tag{19}$$

we generate actions using the distribution over the state that the action will be applied to as policy input:

$$a_{t+k} \sim \pi_\theta(\cdot|p(\cdot|s_t, a_t, a_{t+1}, \ldots, a_{t+k-1})), \tag{20}$$

where $p(\cdot|s_t, a_t, a_{t+1}, \ldots, a_{t+k-1})$ represents the distribution over states after applying the action sequence $a_t, a_{t+1}, \ldots, a_{t+k-1}$, in order, to the state $s_t$. We represent $p(\cdot|s_t, a_t, a_{t+1}, \ldots, a_{t+k-1})$ as a fixed-size latent representation, and thanks to this representation, we can generate actions with MLPs for variable-size inputs. The purpose of the model is to create these embeddings (defining the mapping between $p(\cdot|s_t, a_t, a_{t+1}, \ldots, a_{t+k-1})$ and the corresponding latent representation). Since this kind of policy makes decisions using a distribution over the state, and that the distribution is embedded as a latent representation using a model, we refer to agents using this kind of policy as *model-based distribution agents* (MDA).

The model consists of three components: $\text{EMBED}_\omega(s_t)$, $\text{STEP}_\omega(z_i, a_{t+i})$, and $\text{EMIT}_\omega(\hat{s}_{t+i}|z_{t+i})$.

$\text{EMBED}_\omega(s_t)$ embeds the state $s_t$ into a latent representation $z_0$. In a perfect model, $z_0$ would be an embedding of the Dirac delta distribution $\delta(x - s_t)$.

$\text{STEP}_\omega(z_i, a_{t+i})$ updates the latent representation $z_i$ to include information about what happens if the action $a_{t+i}$ is also applied. Such that, if $z_i$ is a latent representation of $p(\cdot|s_t, a_t, \ldots, a_{t+i-1})$, then $z_{i+1} = \text{STEP}_\omega(z_i, a_{t+i})$ is a latent representation of $p(\cdot|s_t, a_t, \ldots, a_{t+i-1}, a_{t+i})$.

$\text{EMIT}_\omega(\hat{s}_{t+i}|z_{t+i})$ converts the latent representation $z_{t+i}$ back to a regular parameterized distribution. We use a normal distribution in our model, where $\text{EMIT}_\omega$ outputs the mean and standard deviation for each component of the MDP state. We never sample from this distribution. This component is only used to ensure that we have a good latent representation.

To make the notation more compact, we use the multi-step notation $\text{STEP}_\omega^k$ where

$$\text{STEP}_\omega^0(z) = z \tag{21}$$

$$\text{STEP}_\omega^k(z, a_0, a_1, \ldots, a_{k-1}) = \text{STEP}_\omega^{k-1}(\text{STEP}_\omega(z, a_0), a_1, \ldots, a_{k-1}) \tag{22}$$

With this notation, we say that $\text{STEP}_\omega^k(\text{EMBED}_\omega(s_t), a_t, a_{t+1}, \ldots, a_{t+k-1})$ embeds the distribution $p(\cdot|s_t, a_t, a_{t+1}, \ldots, a_{t+k-1})$. We optimize the model to minimize the KL-divergence between the embedded distribution and the true distribution. That is

$$\min_\omega D_{\text{KL}}\left(p(\cdot|s_t, a_t, \ldots, a_{t+n-1}) \,\|\, \text{EMIT}_\omega(\cdot|\text{STEP}_\omega^n(\text{EMBED}_\omega(s_t), a_t, \ldots, a_{t+n-1}))\right) \tag{23}$$

for all possible states $s_t$ and sequences of actions $a_t, \ldots, a_{t+n-1}$. The loss function $\mathcal{L}(\omega)$ from Section 4.2 is a Monte Carlo estimate of this objective:

$$\mathcal{L}(\omega) = \mathbb{E}_{(s_t, a_t, a_{t+1}, \ldots, a_{t+n-1}, s_{t+n}) \sim \mathcal{R}}\left[-\log \text{EMIT}_\omega(s_{t+n}|z_n)\right] \tag{24}$$

$$\text{where } z_n = \text{STEP}_\omega^n(\text{EMBED}_\omega(s_t), a_t, a_{t+1}, \ldots, a_{t+n-1})$$

and $\mathcal{R}$ is a replay buffer with experiences collected online.

## D.2  TRAINING ALGORITHM

This section presents the full version of the training algorithm from Section 4.3. As with SAC, we assume that $\pi_\theta$ is represented as a reparameterizable policy that is a deterministic function with independent noise input.

---

**Algorithm 5** Actor-Critic with Delay Adaptation

---

1: Initialize policy $\pi_\theta$, critics $Q_{\phi_1}, Q_{\phi_2}$, model $\omega$, temperature $\alpha$, target networks $\phi_1', \phi_2'$, and replay $\mathcal{R}$

2: **for** each epoch **do**

3: *// Stage 1: Sample trajectory*

4: Collected trajectory: $\mathcal{T} = \emptyset$

5: $t \leftarrow 0$

6: Reset interaction layer state: $s_0 \sim \boldsymbol{\mu}$, Observe $\boldsymbol{o}_0$

7: **while** terminal state not reached **do**

8:  $(t, s_t, \boldsymbol{b}_t, \delta_t, c_t) = \boldsymbol{o}_t$

9:  **for** $k \leftarrow 1$ to $L$ **do**

10:   Select $\hat{a}_1^{t+k}, \ldots, \hat{a}_k^{t+k}$ by Algorithm 1

11:   $y_0 \leftarrow (\hat{a}_1^{t+k}, \ldots, \hat{a}_k^{t+k})$

12:   **for** $i \leftarrow 1$ to $h$ **do**

13:    $a_i^{t+k} \sim \pi_\theta(\cdot | \text{STEP}_\omega^{k+i-1}(\text{EMBED}_\omega(s_t), y_{i-1}))$

14:    $y_i \leftarrow (y_{i-1}, a_i^{t+k})$

15:  $\boldsymbol{a}_t \leftarrow \left( t, \begin{bmatrix} a_1^{t+1} & a_2^{t+1} & a_3^{t+1} & \ldots & a_h^{t+1} \\ a_1^{t+2} & a_2^{t+2} & a_3^{t+2} & \ldots & a_h^{t+2} \\ \vdots & \vdots & \vdots & \ddots & \vdots \\ a_1^{t+L} & a_2^{t+L} & a_3^{t+L} & \ldots & a_h^{t+L} \end{bmatrix} \right)$

16:  Send $\boldsymbol{a}_t$ to interaction layer, observe $r_t, \boldsymbol{o}_{t+1}, \Gamma_{t+1}$

17:  Add $(\boldsymbol{o}_t, \boldsymbol{a}_t, r_t, \boldsymbol{o}_{t+1}, \Gamma_{t+1})$ to $\mathcal{T}$

18:  $t \leftarrow t + 1$

19: *// Stage 2: Reconstruct transition info*

20: **for** $(\boldsymbol{o}_i, \boldsymbol{a}_i, r_i, \boldsymbol{o}_{i+1}, \Gamma_{i+1}) \in \mathcal{T}$ **do**

21:  $(i, s_i, \boldsymbol{b}_i, \delta_i, c_i) = \boldsymbol{o}_i, \quad (i+1, s_{i+1}, \boldsymbol{b}_{i+1}, \delta_{i+1}, c_{i+1}) = \boldsymbol{o}_{i+1}$

22:  $a_i = b_{i,1}$

23:  **if** $i - (\delta_i + c_i) \geq 0$ **then**

24:   *// (Recover the input used by $\pi_\theta$ and $\text{STEP}_\omega^k$ to generate $b_{i,1}$ and $b_{i+1,1}$)*

25:   $j \leftarrow i - (\delta_i + c_i), \quad j' \leftarrow (i+1) - (\delta_{i+1} + c_{i+1})$

26:   Reconstruct $\hat{a}_1^{j+\delta_i}, \ldots, \hat{a}_{\delta_i}^{j+\delta_i}$, choose $a_1^{j+\delta_i}, \ldots, a_{c_i}^{j+\delta_i}$ from $\boldsymbol{a}_j$

27:   Reconstruct $\hat{a}_1^{j'+\delta_{i+1}}, \ldots, \hat{a}_{\delta_{i+1}}^{j'+\delta_{i+1}}$, choose $a_1^{j'+\delta_{i+1}}, \ldots, a_{c_{i+1}}^{j'+\delta_{i+1}}$ from $\boldsymbol{a}_{j'}$

28:   $y_i \leftarrow (\hat{a}_1^{j+\delta_i}, \ldots, \hat{a}_{\delta_i}^{j+\delta_i}, a_1^{j+\delta_i}, \ldots, a_{c_i}^{j+\delta_i})$

29:   $y_{i+1} \leftarrow (\hat{a}_1^{j'+\delta_{i+1}}, \ldots, \hat{a}_{\delta_{i+1}}^{j'+\delta_{i+1}}, a_1^{j'+\delta_{i+1}}, \ldots, a_{c_{i+1}}^{j'+\delta_{i+1}})$

30:   *// We denote their lengths as $|y_i| = \delta_i + c_i$*

31:   Add $(s_i, a_i, r_i, s_{i+1}, \Gamma_{i+1}, y_i, y_{i+1})$ to $\mathcal{R}$

32: *// Stage 3: Update network weights*

33: **for** $|\mathcal{T}|$ sampled batches of $(s, a, r, s', \Gamma, y, y')$ from $\mathcal{R}$ **do**

34:  *// These are computed in expectation of samples from $\mathcal{R}$*

35:  $\hat{a}' \sim \pi_\theta(\cdot | \text{STEP}_\omega^{|y'|}(\text{EMBED}_\omega(s'), y'))$

36:  $x = r + \gamma(1 - \Gamma)(\min(Q_{\phi_1'}(s', \hat{a}'), Q_{\phi_2'}(s', \hat{a}') - \alpha \log \pi_\theta(\hat{a}' | \ldots))$

37:  Do gradient descent step on $\nabla_{\phi_1}(Q_{\phi_1}(s, a) - x)^2$ and $\nabla_{\phi_2}(Q_{\phi_2}(s, a) - x)^2$

38:  $\hat{a} \sim \pi_\theta(\cdot | \text{STEP}_\omega^{|y|}(\text{EMBED}_\omega(s), y))$

39:  Do gradient ascent step on $\nabla_\theta(\min(Q_{\phi_1}(s, \hat{a}), Q_{\phi_2}(s, \hat{a})) - \alpha \log \pi_\theta(\hat{a} | \ldots))$

40:  Update $\alpha$ according to SAC

41:  Compute $\nabla_\omega$ by Equation 8 and do gradient descent step

42:  Update target networks $\phi_1', \phi_2'$

---

# E    ADDITIONAL RESULTS

This section presents additional results to complement those presented in Section 5. Appendix E.1 presents the results from Table 1 as time series plots, showing how the mean and standard deviation of the evaluated return change over the training process. In subsequent tables, the standard deviation is shown following the $\pm$ symbol.

Appendix E.2 and E.3 answer two questions that are not part of the evaluation in Section 5. The questions that these appendices answer are:

- Appendix E.2 answers the question whether the gain in performance is due to the model-based policy introduced in Section 4.2 or due to the adaptiveness of ACDA. We evaluate this by modifying the BPQL algorithm such that it uses the MDA policy instead of a direct MLP policy, and compare how that performs against ACDA. The results clearly show that it is the adaptiveness of the interaction layer that is the reason for the strong performance of ACDA.

- Appendix E.3 answers the question whether acting with CDA under a different horizon $h$ than the worst-case delay is better than trying to adapt to varying delays. We set up this demonstration by applying CDA with a horizon $h$ that is greater than or equal to the sampled delay most of the time, but occasionally a sampled delay exceeds this horizon. This kind of CDA does not result in a constant-delay MDP that BPQL and VDPO expect, hence we did not include this in the main evaluation. The results in Appendix E.3 show that while VDPO and BPQL occasionally gain performance in this setting, ACDA is still the best performing algorithm in most cases. ACDA is always close to the highest performing algorithm in the few cases where ACDA does not achieve the best mean return. These results show that the adaptiveness of the interaction layer provides an increase in performance that cannot be achieved through constant-delay approaches.

In addition to the results from Appendix E.3, we also evaluate when the horizon $h$ is based on the average sampled delay under each evaluated delay process. These average delay results are presented in Appendix E.4. Using the average delay as the $h$ never results in the best average return for a benchmark. However, in a few instances, the average delay does result in the best performance amongst different horizon values within a specific constant-delay algorithm.

The results from all evaluated baselines are presented in Appendix E.5.

## E.1    TIME SERIES RESULTS

This section presents the results from the evaluation in Section 5 as time series plots, including standard deviation bands. Unless otherwise specified, the evaluation methodology follows that described in Section 5. To reduce noise and highlight trends, each time series is smoothed using a running average over five evaluation points.

The results in this appendix are split into the delay processes used. Appendix E.1.1 presents results for the $GE_{1,23}$ delay process, Appendix E.1.2 for the $GE_{4,32}$ delay process, and Appendix E.1.3 for the M/M/1 queue delay process.

### E.1.1 Performance Evaluation under the $GE_{1,23}$ Delay Process

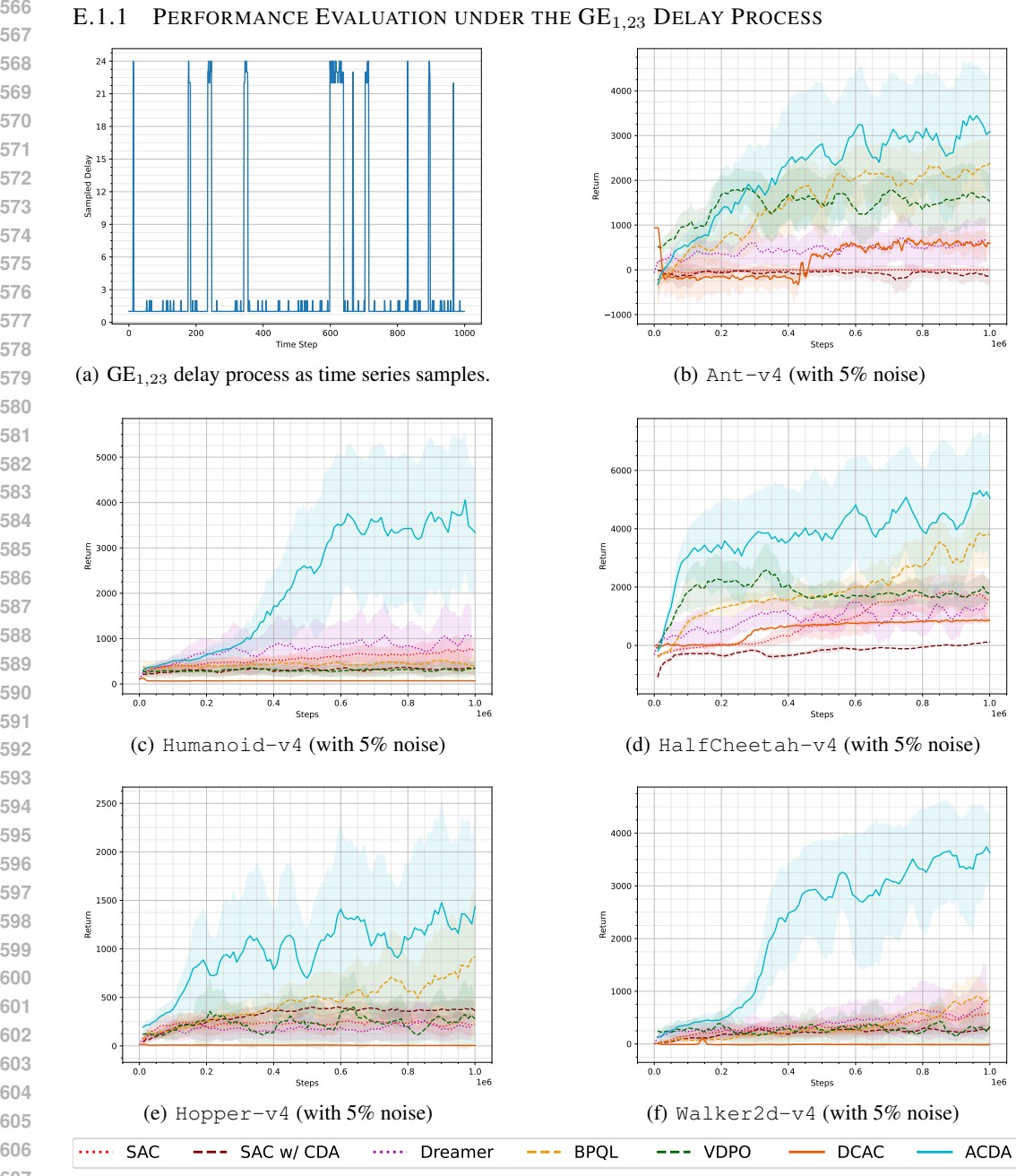

(a) $GE_{1,23}$ delay process as time series samples.

(b) `Ant-v4` (with 5% noise)

(c) `Humanoid-v4` (with 5% noise)

(d) `HalfCheetah-v4` (with 5% noise)

(e) `Hopper-v4` (with 5% noise)

(f) `Walker2d-v4` (with 5% noise)

······ SAC    --- SAC w/ CDA    ······ Dreamer    --- BPQL    --- VDPO    — DCAC    — ACDA

Figure 13: Time series evaluation during training on the $GE_{1,23}$ delay process. All environments have added 5% noise to the actions.

Table 8: Best returns from the $GE_{1,23}$ delay process.

|  | Ant-v4 | Humanoid-v4 | HalfCheetah-v4 | Hopper-v4 | Walker2d-v4 |
|---|---|---|---|---|---|
| SAC | $14.22 \pm 14.89$ | $862.18 \pm 266.21$ | $2064.18 \pm 223.48$ | $306.91 \pm 51.26$ | $708.33 \pm 221.53$ |
| SAC w/ CDA | $69.28 \pm 114.47$ | $414.05 \pm 204.20$ | $128.47 \pm 9.55$ | $426.92 \pm 27.93$ | $428.44 \pm 509.44$ |
| Dreamer | $1111.73 \pm 412.35$ | $1463.07 \pm 649.56$ | $1796.07 \pm 381.47$ | $334.30 \pm 245.42$ | $1081.12 \pm 905.94$ |
| BPQL | $2691.88 \pm 129.84$ | $585.19 \pm 163.49$ | $4320.20 \pm 1028.52$ | $1328.71 \pm 937.67$ | $1215.91 \pm 776.93$ |
| VDPO | $2163.00 \pm 53.04$ | $417.25 \pm 210.09$ | $3144.23 \pm 1156.52$ | $709.20 \pm 522.01$ | $846.88 \pm 808.67$ |
| DCAC | $949.97 \pm 11.87$ | $128.47 \pm 36.09$ | $920.09 \pm 33.05$ | $16.99 \pm 15.94$ | $106.70 \pm 53.84$ |
| **ACDA** | $\mathbf{4112.78 \pm 818.44}$ | $\mathbf{4608.76 \pm 1084.52}$ | $\mathbf{5984.25 \pm 1885.78}$ | $\mathbf{2094.65 \pm 944.20}$ | $\mathbf{3863.59 \pm 232.52}$ |

E.1.2   PERFORMANCE EVALUATION UNDER THE $GE_{4,32}$ DELAY PROCESS

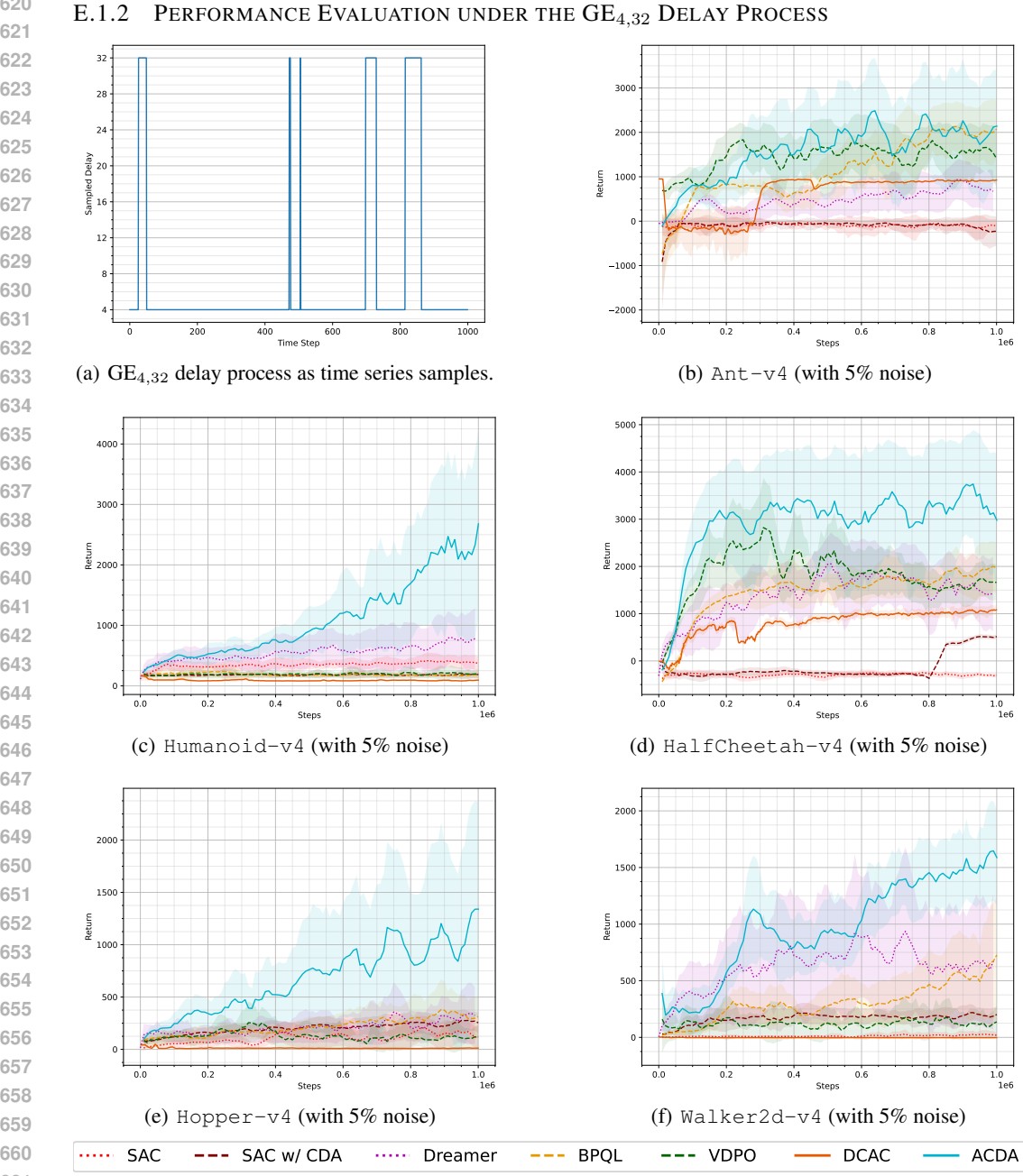

(a) $GE_{4,32}$ delay process as time series samples.

(b) Ant-v4 (with 5% noise)

(c) Humanoid-v4 (with 5% noise)

(d) HalfCheetah-v4 (with 5% noise)

(e) Hopper-v4 (with 5% noise)

(f) Walker2d-v4 (with 5% noise)

······ SAC    — — SAC w/ CDA    ······ Dreamer    — — BPQL    — — VDPO    —— DCAC    —— ACDA

Figure 14: Time series evaluation during training on the $GE_{4,32}$ delay process. All environments have added 5% noise to the actions.

Table 9: Best returns from the $GE_{4,32}$ delay process.

|  | Ant-v4 | Humanoid-v4 | HalfCheetah-v4 | Hopper-v4 | Walker2d-v4 |
|---|---|---|---|---|---|
| SAC | $-5.72 \pm 19.62$ | $494.43 \pm 156.01$ | $-158.78 \pm 65.86$ | $279.74 \pm 109.83$ | $60.86 \pm 75.59$ |
| SAC w/ CDA | $18.93 \pm 23.64$ | $230.45 \pm 99.22$ | $591.32 \pm 36.39$ | $315.47 \pm 51.49$ | $257.18 \pm 73.42$ |
| Dreamer | $1147.56 \pm 371.15$ | $1091.48 \pm 577.52$ | $2493.19 \pm 231.22$ | $515.36 \pm 430.72$ | $1233.79 \pm 802.47$ |
| BPQL | $2509.52 \pm 117.37$ | $276.63 \pm 131.70$ | $2136.36 \pm 547.04$ | $433.29 \pm 381.79$ | $875.09 \pm 747.72$ |
| VDPO | $2266.99 \pm 90.89$ | $280.72 \pm 169.85$ | $3664.30 \pm 929.25$ | $330.44 \pm 263.74$ | $344.73 \pm 316.82$ |
| DCAC | $953.14 \pm 12.06$ | $167.97 \pm 82.14$ | $1123.47 \pm 100.34$ | $57.98 \pm 28.04$ | $9.23 \pm 20.35$ |
| ACDA | $\mathbf{2866.93 \pm 1172.46}$ | $\mathbf{3725.59 \pm 1513.38}$ | $\mathbf{4231.15 \pm 333.69}$ | $\mathbf{1727.79 \pm 959.50}$ | $\mathbf{1840.58 \pm 386.78}$ |

### E.1.3 PERFORMANCE EVALUATION UNDER THE M/M/1 QUEUE DELAY PROCESS

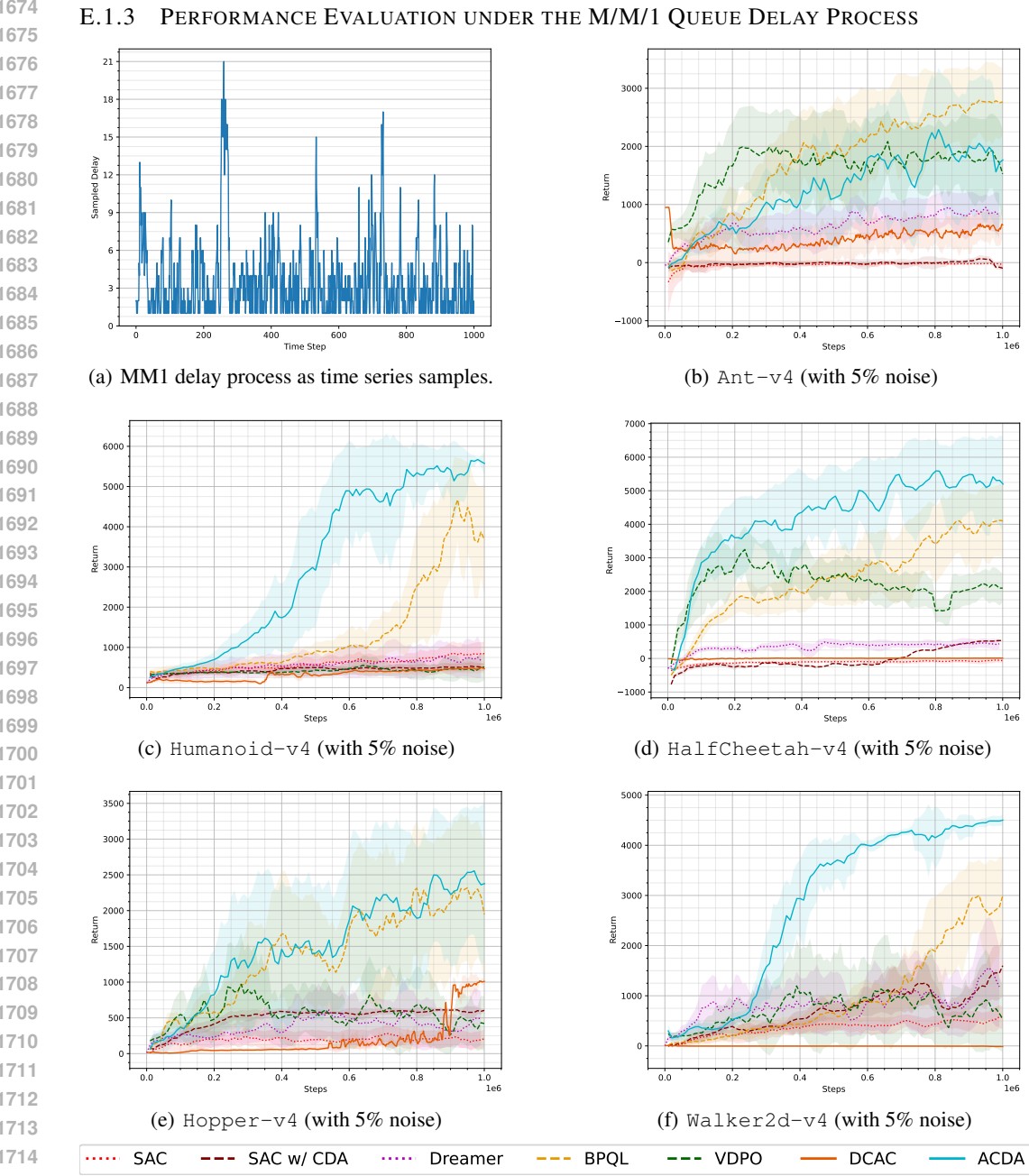

(a) MM1 delay process as time series samples.

(b) `Ant-v4` (with 5% noise)

(c) `Humanoid-v4` (with 5% noise)

(d) `HalfCheetah-v4` (with 5% noise)

(e) `Hopper-v4` (with 5% noise)

(f) `Walker2d-v4` (with 5% noise)

······ SAC    --- SAC w/ CDA    ······ Dreamer    --- BPQL    --- VDPO    --- DCAC    --- ACDA

Figure 15: Time series evaluation during training on the MM1 delay process. All environments have added 5% noise to the actions.

Table 10: Best returns from the MM1 delay process.

|  | Ant-v4 | Humanoid-v4 | HalfCheetah-v4 | Hopper-v4 | Walker2d-v4 |
|---|---|---|---|---|---|
| SAC | $-0.58 \pm 8.66$ | $921.04 \pm 299.47$ | $20.69 \pm 94.91$ | $333.06 \pm 96.04$ | $604.80 \pm 212.37$ |
| SAC w/ CDA | $102.00 \pm 33.77$ | $613.03 \pm 157.68$ | $550.84 \pm 16.28$ | $627.59 \pm 24.62$ | $2005.76 \pm 341.30$ |
| Dreamer | $1121.11 \pm 58.69$ | $981.38 \pm 597.01$ | $584.40 \pm 72.26$ | $975.72 \pm 650.05$ | $1801.81 \pm 1158.73$ |
| BPQL | $\mathbf{3074.17 \pm 106.78}$ | $5435.29 \pm 68.34$ | $4660.93 \pm 448.10$ | $3035.66 \pm 103.80$ | $3547.73 \pm 133.51$ |
| VDPO | $2528.67 \pm 144.63$ | $720.73 \pm 634.35$ | $3831.96 \pm 960.07$ | $1459.88 \pm 933.11$ | $2144.25 \pm 1650.85$ |
| DCAC | $959.23 \pm 13.54$ | $525.85 \pm 135.36$ | $35.60 \pm 21.42$ | $1026.45 \pm 2.96$ | $24.48 \pm 45.46$ |
| ACDA | $2898.46 \pm 838.07$ | $\mathbf{5805.60 \pm 23.04}$ | $\mathbf{5898.36 \pm 409.10}$ | $\mathbf{3122.53 \pm 417.37}$ | $\mathbf{4562.33 \pm 87.98}$ |

## E.2    MODEL-BASED DISTRIBUTION AGENT VS. ADAPTIVENESS

The purpose of this Appendix is to answer the question of whether it is the model-based distribution agent (MDA) or the adaptivity of ACDA that leads to its high performance. To answer this, we modify the BPQL algorithm to use the MDA policy instead of the direct MLP policy that they used in their original paper.

It is necessary to modify the BPQL algorithm itself since the optimization of the MDA policy is split into two steps, with different kinds of samples from the replay buffer. If the delay truly was constant, then BPQL with MDA would be the same as the performance of ACDA, due to the perfect conditions for the memorized action selection. However, it is necessary to split these into two algorithms since this cannot capture the M/M/1 queue delay process, which cannot be represented as a true constant-delay MDP.

Like Appendix E.1, results are split based on the delay process used. Appendix E.2.1 presents results for the $GE_{1,23}$ delay process, Appendix E.2.2 for the $GE_{4,32}$ delay process, and Appendix E.2.3 for the M/M/1 queue delay process.

These results show that, while BPQL sometimes performs better using the MDA policy, ACDA, with its adaptivity, is still the best-performing algorithm.

## E.2.1 PERFORMANCE OF MDA UNDER THE $GE_{1,23}$ DELAY PROCESS

(a) $GE_{1,23}$ delay process as time series samples. The red line indicates assumed upper bound for CDA.

(b) `Ant-v4` (with 5% noise)

(c) `Humanoid-v4` (with 5% noise)

(d) `HalfCheetah-v4` (with 5% noise)

(e) `Hopper-v4` (with 5% noise)

(f) `Walker2d-v4` (with 5% noise)

--- BPQL   --- VDPO   —— ACDA   --- BPQL w/ MDA

Figure 16: Time series evaluation during training on the $GE_{1,23}$ delay process. All environments have added 5% noise to the actions.

Table 11: Best returns from the $GE_{1,23}$ delay process.

| | Ant-v4 | Humanoid-v4 | HalfCheetah-v4 | Hopper-v4 | Walker2d-v4 |
|---|---|---|---|---|---|
| BPQL | $2691.88 \pm 129.84$ | $585.19 \pm 163.49$ | $4320.20 \pm 1028.52$ | $1328.71 \pm 937.67$ | $1215.91 \pm 776.93$ |
| VDPO | $2163.00 \pm 53.04$ | $417.25 \pm 210.09$ | $3144.23 \pm 1156.52$ | $709.20 \pm 522.01$ | $846.88 \pm 808.67$ |
| ACDA | $\mathbf{4112.78 \pm 818.44}$ | $\mathbf{4608.76 \pm 1084.52}$ | $\mathbf{5984.25 \pm 1885.78}$ | $\mathbf{2094.65 \pm 944.20}$ | $\mathbf{3863.59 \pm 232.52}$ |
| BPQL w/ MDA | $1795.29 \pm 23.78$ | $563.36 \pm 96.11$ | $4926.36 \pm 60.08$ | $465.14 \pm 138.86$ | $3681.39 \pm 126.41$ |

### E.2.2 PERFORMANCE OF MDA UNDER THE $GE_{4,32}$ DELAY PROCESS

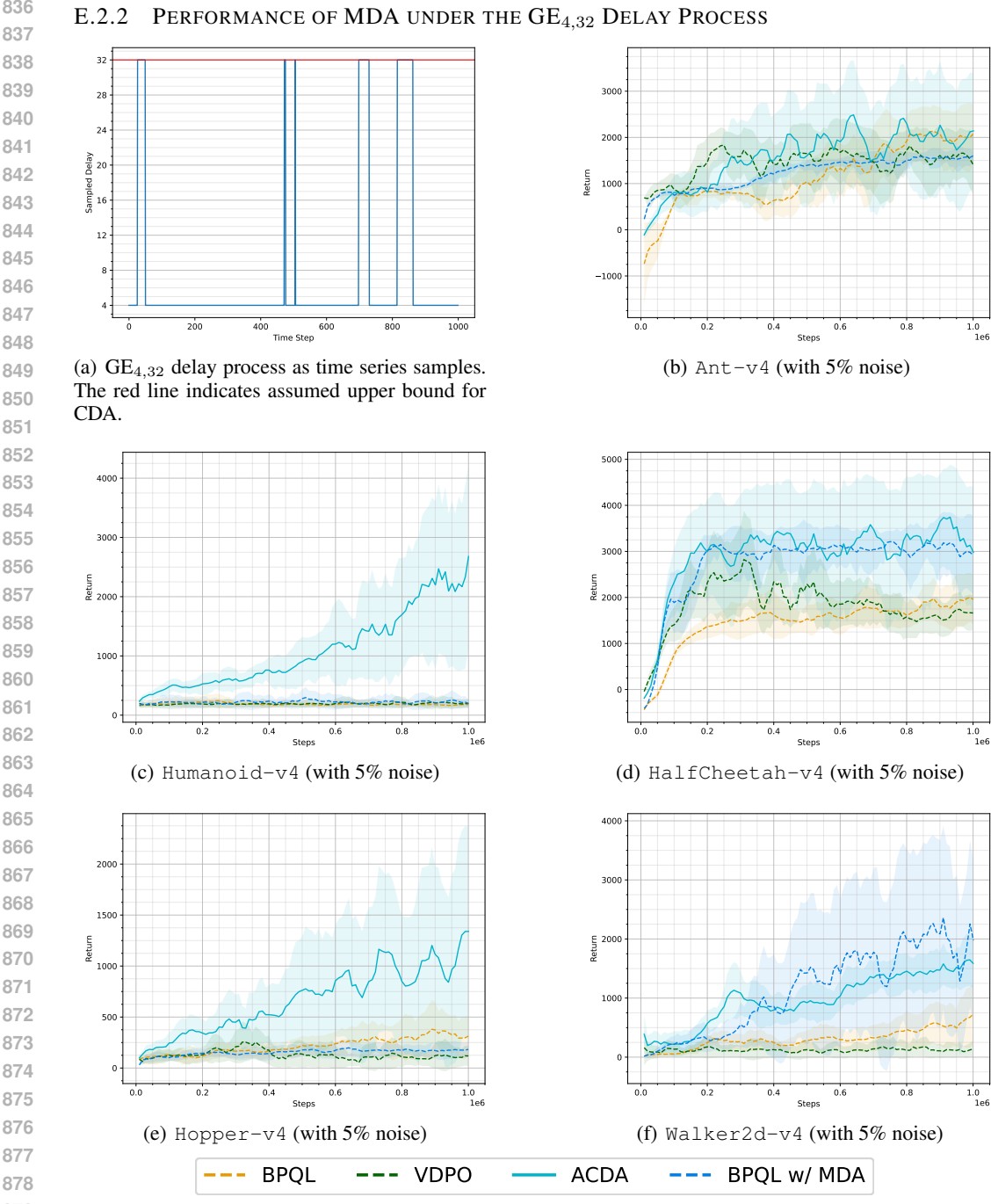

(a) $GE_{4,32}$ delay process as time series samples. The red line indicates assumed upper bound for CDA.

(b) `Ant-v4` (with 5% noise)

(c) `Humanoid-v4` (with 5% noise)

(d) `HalfCheetah-v4` (with 5% noise)

(e) `Hopper-v4` (with 5% noise)

(f) `Walker2d-v4` (with 5% noise)

--- BPQL    --- VDPO    —— ACDA    --- BPQL w/ MDA

Figure 17: Time series evaluation during training on the $GE_{4,32}$ delay process. All environments have added 5% noise to the actions.

Table 12: Best returns from the $GE_{4,32}$ delay process.

|  | Ant-v4 | Humanoid-v4 | HalfCheetah-v4 | Hopper-v4 | Walker2d-v4 |
|---|---|---|---|---|---|
| BPQL | 2509.52 ± 117.37 | 276.63 ± 131.70 | 2136.36 ± 547.04 | 433.29 ± 381.79 | 875.09 ± 747.72 |
| VDPO | 2266.99 ± 90.89 | 280.72 ± 169.85 | 3664.30 ± 929.25 | 330.44 ± 263.74 | 344.73 ± 316.82 |
| ACDA | **2866.93 ± 1172.46** | **3725.59 ± 1513.38** | **4231.15 ± 333.69** | **1727.79 ± 959.50** | 1840.58 ± 386.78 |
| BPQL w/ MDA | 1661.41 ± 43.42 | 359.46 ± 156.86 | 3609.45 ± 328.37 | 224.34 ± 90.12 | **3015.45 ± 1428.05** |

### E.2.3 PERFORMANCE OF MDA UNDER THE M/M/1 QUEUE DELAY PROCESS

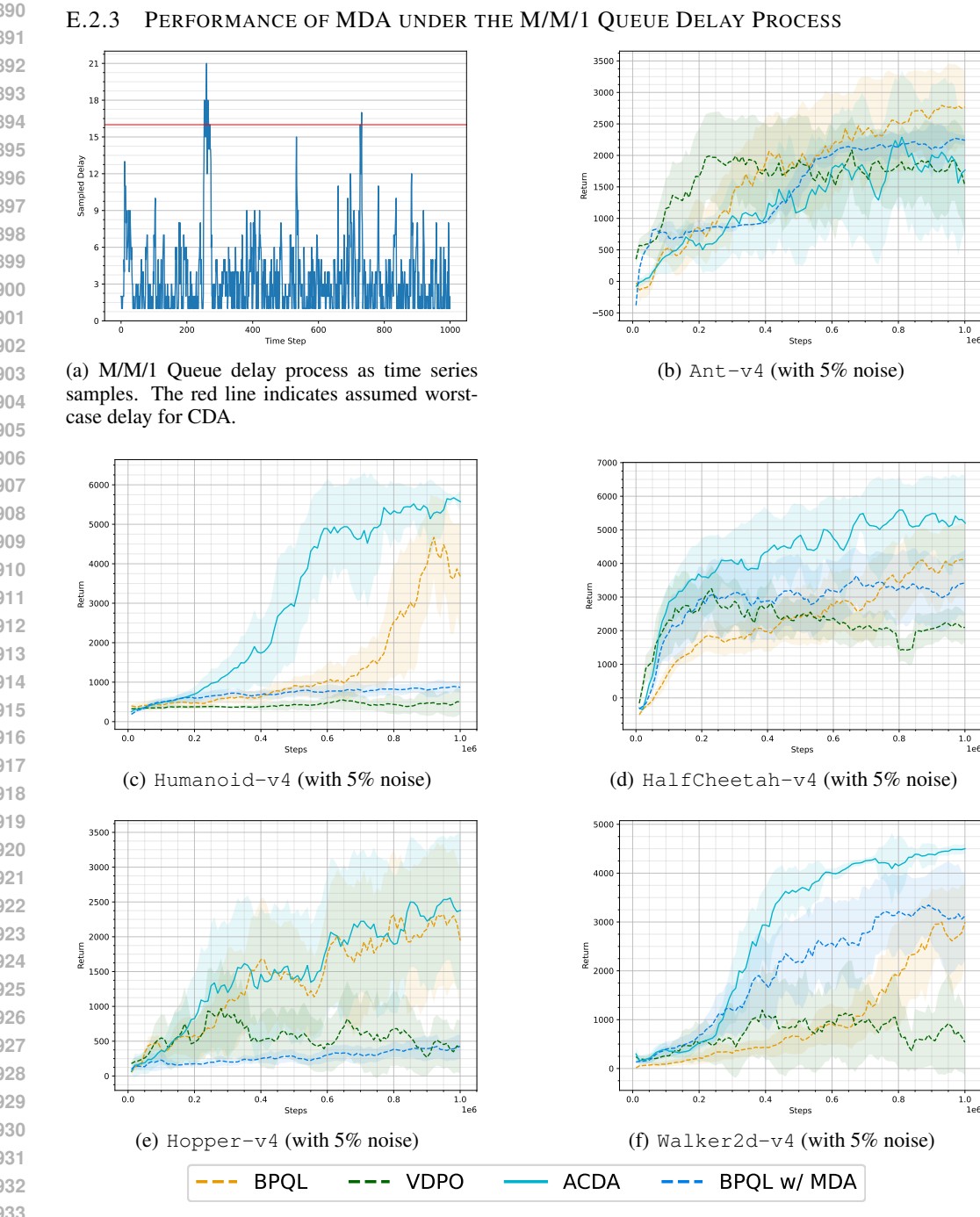

(a) M/M/1 Queue delay process as time series samples. The red line indicates assumed worst-case delay for CDA.

(b) `Ant-v4` (with 5% noise)

(c) `Humanoid-v4` (with 5% noise)

(d) `HalfCheetah-v4` (with 5% noise)

(e) `Hopper-v4` (with 5% noise)

(f) `Walker2d-v4` (with 5% noise)

BPQL — — — VDPO — — — ACDA ——— BPQL w/ MDA — — —

Figure 18: Time series evaluation during training on the MM1 delay process. All environments have added 5% noise to the actions.

Table 13: Best returns from the MM1 delay process.

|  | Ant-v4 | Humanoid-v4 | HalfCheetah-v4 | Hopper-v4 | Walker2d-v4 |
|---|---|---|---|---|---|
| BPQL | **3074.17 ± 106.78** | 5435.29 ± 68.34 | 4660.93 ± 448.10 | 3035.66 ± 103.80 | 3547.73 ± 133.51 |
| VDPO | 2528.67 ± 144.63 | 720.73 ± 634.35 | 3831.96 ± 960.07 | 1459.88 ± 933.11 | 2144.25 ± 1650.85 |
| ACDA | 2898.46 ± 838.07 | **5805.60 ± 23.04** | **5898.36 ± 409.10** | **3122.53 ± 417.37** | **4562.33 ± 87.98** |
| BPQL w/ MDA | 2308.18 ± 75.13 | 944.56 ± 92.79 | 3953.69 ± 351.34 | 512.43 ± 346.18 | 3667.61 ± 142.48 |

### E.3 RESULTS WHEN VIOLATING THE UPPER BOUND ASSUMPTIONS

The $GE_{1,23}$ and $GE_{4,32}$ delay processes occasionally have a very high delay, but most of the time, the delays of these are very low. To convert these to a constant-delay MDP, we need to assume the worst-case possible delay of the process. We do this using the CDA method described in Appendix A. CDA can be applied regardless of whether the constant $h$ that we wish to act under is a worst-case delay or not. Though CDA only guarantees the MDP property if $h$ is a true upper bound of the delay process.

A natural question is how state-of-the-art approaches perform if we apply CDA to a more favorable constant $h$, which holds most of the time and is much lower than the worst-case delay. We answer this by evaluating BPQL and VDPO under more opportunistic constants $h$. These are compared against the performance of ACDA, which can still adapt to much larger delays. We also include an evaluation of BPQL with the MDA policy, as in Appendix E.2, but now when that acts under the opportunistic constant $h$ instead. We present the results of this evaluation in Appendices E.3.1, E.3.2, and E.3.3, which are split based on the delay process used. The opportunistic constant $h$ used is highlighted as a red line in a time series samples plot for each delay process.

The results show that ACDA still outperforms state-of-the-art in most benchmarks. While BPQL and VDPO can achieve better performance in some cases, ACDA is still performing close to the best algorithm. Also, operating under these opportunistic constants can have significant negative consequences. This is best highlighted by the results in Figure 19(d), where VDPO experiences a collapse in performance under the `HalfCheetah-v4` environment using the $GE_{1,23}$ delay process.

Based on these results, we conclude that the adaptiveness provided by the interaction layer is a necessity to be able to achieve high performance under random unobservable delays. While it is possible to sacrifice the MDP property to gain performance in the constant-delay setting, state-of-the-art still does not outperform the adaptive ACDA algorithm.

### E.3.1 $GE_{1,23}$ DELAY PROCESS WITH LOW CDA

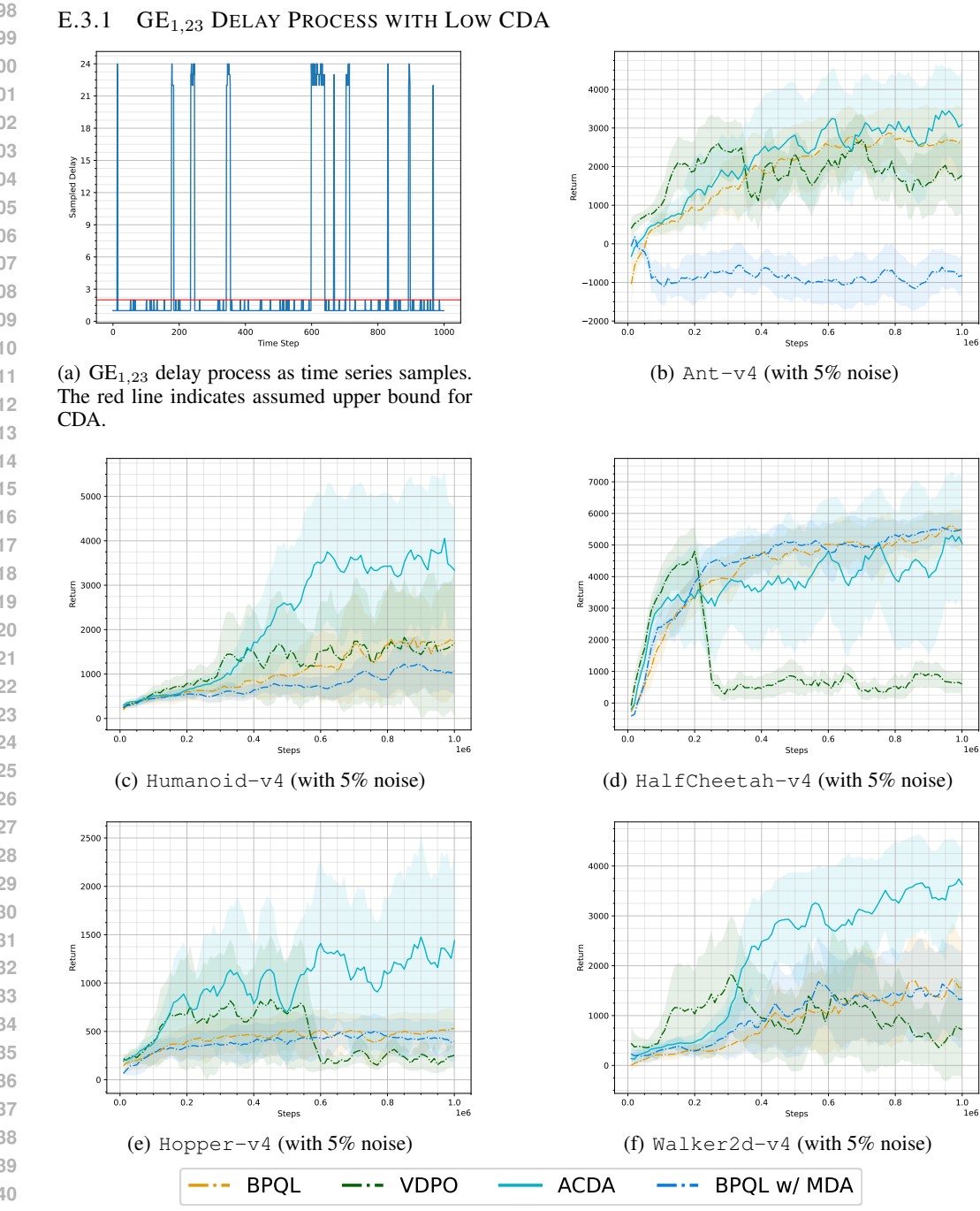

(a) $GE_{1,23}$ delay process as time series samples. The red line indicates assumed upper bound for CDA.

(b) `Ant-v4` (with 5% noise)

(c) `Humanoid-v4` (with 5% noise)

(d) `HalfCheetah-v4` (with 5% noise)

(e) `Hopper-v4` (with 5% noise)

(f) `Walker2d-v4` (with 5% noise)

—·— BPQL    —··— VDPO    —— ACDA    —··— BPQL w/ MDA

Figure 19: Time series evaluation during training on the $GE_{1,23}$ delay process for opportunistic delay assumptions of $h = 2$ (except for ACDA). All environments have added 5% noise to the actions.

Table 14: Best returns from the $GE_{1,23}$ delay process.

|  | Ant-v4 | Humanoid-v4 | HalfCheetah-v4 | Hopper-v4 | Walker2d-v4 |
|---|---|---|---|---|---|
| BPQL | 3359.65 ± 288.24 | 2469.96 ± 1375.23 | 5944.67 ± 395.56 | 642.54 ± 420.04 | 2043.32 ± 923.00 |
| VDPO | 3103.10 ± 252.19 | 2658.48 ± 2044.40 | 5625.06 ± 524.23 | 1113.51 ± 836.72 | 2756.73 ± 1693.64 |
| ACDA | **4112.78 ± 818.44** | **4608.76 ± 1084.52** | **5984.25 ± 1885.78** | **2094.65 ± 944.20** | **3863.59 ± 232.52** |
| BPQL w/ MDA | 423.46 ± 73.63 | 1543.10 ± 536.14 | 5710.20 ± 566.48 | 545.01 ± 116.07 | 1923.28 ± 838.62 |

## E.3.2 $GE_{4,32}$ Delay Process with Low CDA

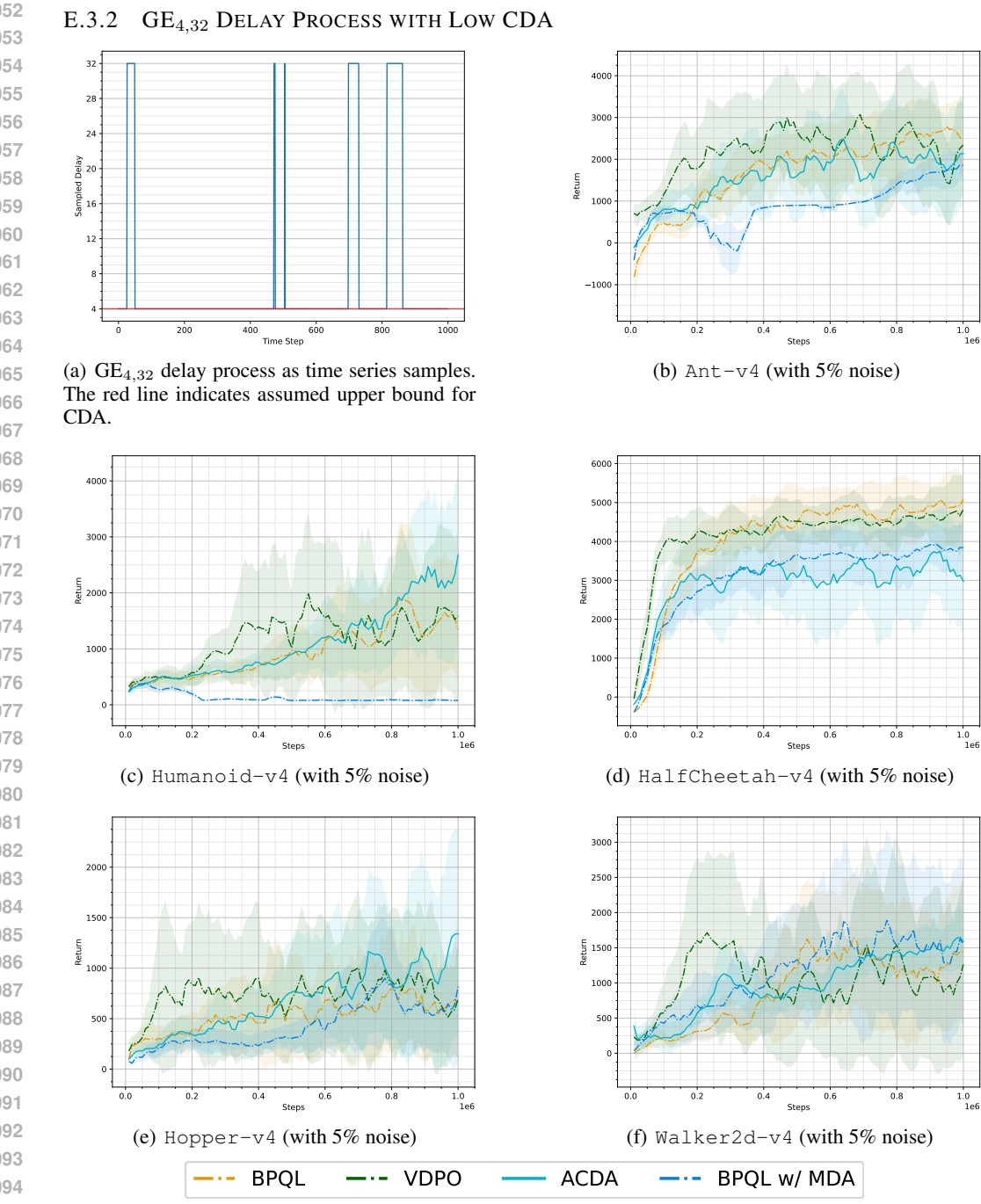

(a) $GE_{4,32}$ delay process as time series samples. The red line indicates assumed upper bound for CDA.

(b) `Ant-v4` (with 5% noise)

(c) `Humanoid-v4` (with 5% noise)

(d) `HalfCheetah-v4` (with 5% noise)

(e) `Hopper-v4` (with 5% noise)

(f) `Walker2d-v4` (with 5% noise)

$$-\cdot- \text{BPQL} \quad -\cdot- \text{VDPO} \quad \text{——} \text{ACDA} \quad -\cdot- \text{BPQL w/ MDA}$$

Figure 20: Time series evaluation during training on the $GE_{4,32}$ delay process for opportunistic delay assumptions of $h = 4$ (except for ACDA). All environments have added 5% noise to the actions.

Table 15: Best returns from the $GE_{4,32}$ delay process.

|  | Ant-v4 | Humanoid-v4 | HalfCheetah-v4 | Hopper-v4 | Walker2d-v4 |
|---|---|---|---|---|---|
| BPQL | $3000.00 \pm 754.09$ | $2949.17 \pm 1933.62$ | $5315.27 \pm 559.25$ | $1243.58 \pm 704.53$ | $2107.39 \pm 1219.55$ |
| VDPO | $\mathbf{3979.56 \pm 331.76}$ | $2956.52 \pm 2322.74$ | $\mathbf{5424.96 \pm 306.06}$ | $1360.87 \pm 627.11$ | $2234.83 \pm 1776.21$ |
| ACDA | $2866.93 \pm 1172.46$ | $\mathbf{3725.59 \pm 1513.38}$ | $4231.15 \pm 333.69$ | $\mathbf{1727.79 \pm 959.50}$ | $1840.58 \pm 386.78$ |
| BPQL w/ MDA | $2155.47 \pm 84.22$ | $465.58 \pm 82.59$ | $4082.54 \pm 411.47$ | $1312.37 \pm 868.12$ | $\mathbf{2355.23 \pm 1145.73}$ |

### E.3.3 M/M/1 Queue Delay Process with Low CDA

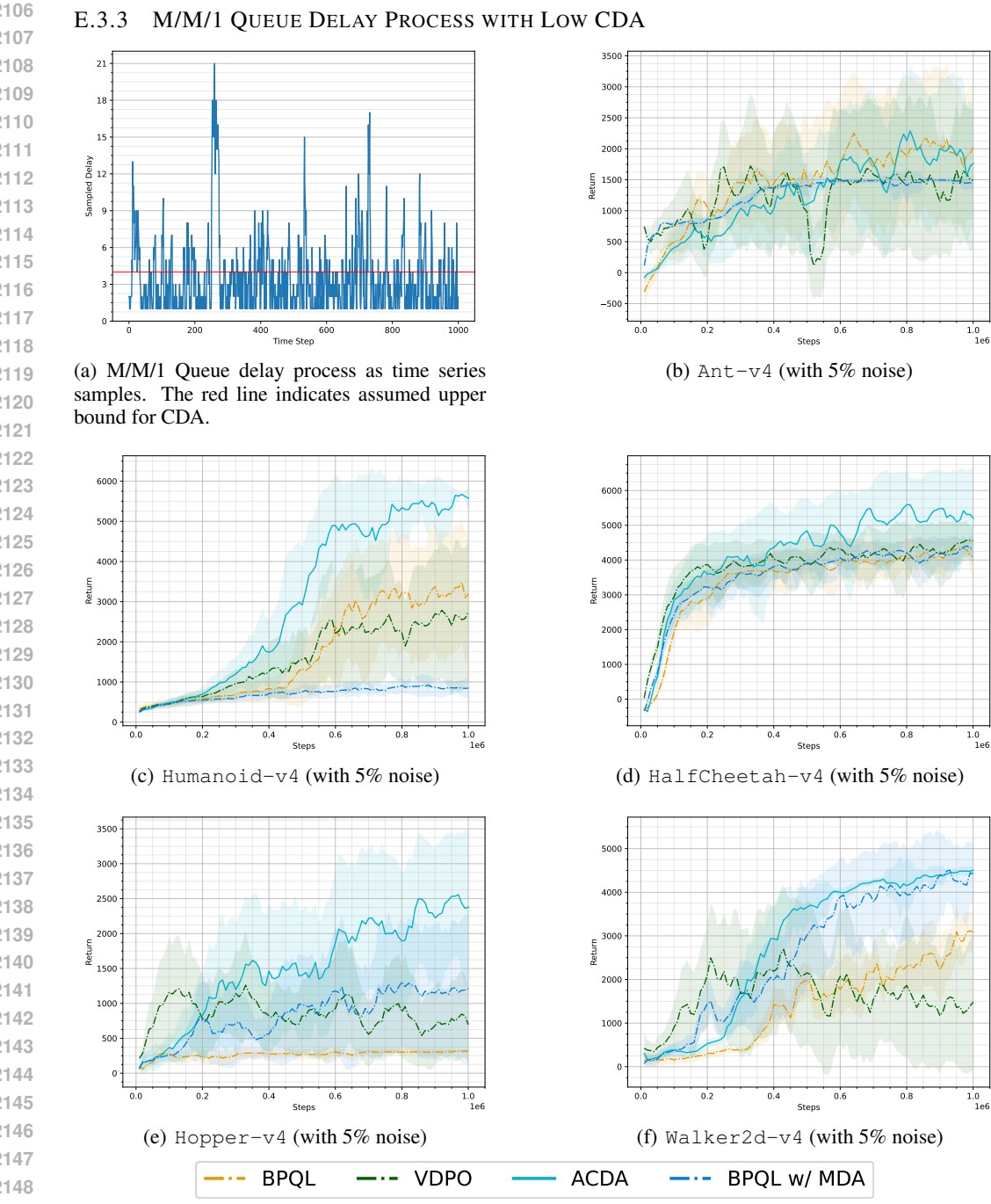

(a) M/M/1 Queue delay process as time series samples. The red line indicates assumed upper bound for CDA.

(b) `Ant-v4` (with 5% noise)

(c) `Humanoid-v4` (with 5% noise)

(d) `HalfCheetah-v4` (with 5% noise)

(e) `Hopper-v4` (with 5% noise)

(f) `Walker2d-v4` (with 5% noise)

$$ \cdots \text{ BPQL} \qquad \cdots \text{ VDPO} \qquad \text{---- ACDA} \qquad \cdots \text{ BPQL w/ MDA} $$

Figure 21: Time series evaluation during training on the MM1 delay process for opportunistic delay assumptions of $h = 4$ (except for ACDA). All environments have added 5% noise to the actions.

Table 16: Best returns from the MM1 delay process.

|  | Ant-v4 | Humanoid-v4 | HalfCheetah-v4 | Hopper-v4 | Walker2d-v4 |
|---|---|---|---|---|---|
| BPQL | 2577.34 ± 1217.11 | 4158.16 ± 981.22 | 4478.64 ± 193.82 | 407.91 ± 179.00 | 3475.80 ± 586.89 |
| VDPO | 2278.08 ± 726.85 | 3349.31 ± 1668.64 | 4857.65 ± 335.88 | 1628.83 ± 880.65 | 3554.77 ± 1278.22 |
| ACDA | **2898.46 ± 838.07** | **5805.60 ± 23.04** | **5898.36 ± 409.10** | **3122.53 ± 417.37** | 4562.33 ± 87.98 |
| BPQL w/ MDA | 1541.32 ± 16.71 | 1030.55 ± 185.86 | 4502.92 ± 214.25 | 1665.66 ± 1210.69 | **4825.75 ± 126.28** |

### E.4 RESULTS FOR AVERAGE DELAYS

This sections performs an evaluation similar to that found in Appendix E.3, but using the average delay as the assumed delay. Although the results in Appendix E.3 are believed to be more favorable for the constant-delay processes, we also evaluate under the average delay to reduce bias in the results.

The average delay for each delay process are:

$$\mathbb{E}[GE_{1,23}] \approx 4.088 \tag{25}$$

$$\mathbb{E}[GE_{4,32}] \approx 7.177 \tag{26}$$

$$\mathbb{E}[MM1] \approx 2.916 \quad \text{(Monte-Carlo estimated from } 10^7 \text{ samples)} \tag{27}$$

As the average delays are not whole numbers, we evaluate both when the average delays are rounded up and when they are rounded down. Although the average delay for MM1 is slightly different to the theoretical average for M/M/1 queues ($\frac{1}{0.75-0.33} \approx 2.381$) due to discretization, this does not affect the rounding.

We present the average CDA results for delay processes $GE_{1,23}$, $GE_{4,32}$, and MM1 in Appendices E.4.1, E.4.2, and E.4.3 respectively. From these results, we see that ACDA is best in 13 out of 15 benchmarks, compared to 11 out of 15 benchmarks for low CDA (presented in Appendix E.3). Additionally, as shown later in Appendix E.5, the average CDA never yields the overall best-performing baseline for any benchmark.

### E.4.1  $GE_{1,23}$ Delay Process with Average CDA

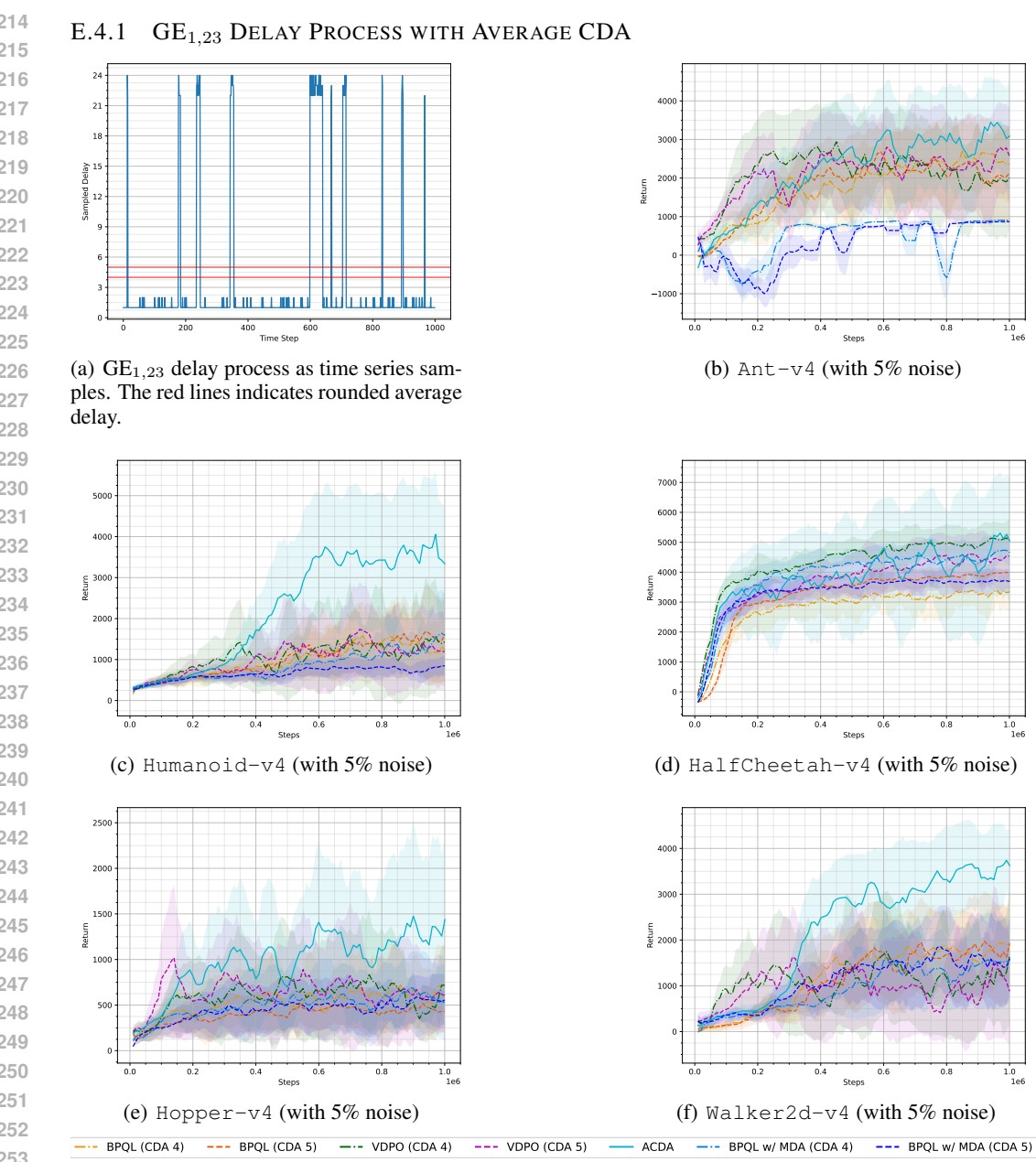

(a) $GE_{1,23}$ delay process as time series samples. The red lines indicates rounded average delay.

(b) `Ant-v4` (with 5% noise)

(c) `Humanoid-v4` (with 5% noise)

(d) `HalfCheetah-v4` (with 5% noise)

(e) `Hopper-v4` (with 5% noise)

(f) `Walker2d-v4` (with 5% noise)

BPQL (CDA 4) — BPQL (CDA 5) — VDPO (CDA 4) — VDPO (CDA 5) — ACDA — BPQL w/ MDA (CDA 4) — BPQL w/ MDA (CDA 5)

Figure 22: Time series evaluation during training on the $GE_{1,23}$ delay process for average delay assumptions of $h = 4$ and $h = 5$. All environments have added 5% noise to the actions.

Table 17: Best returns from the $GE_{1,23}$ delay process. Average delay results below the solid line.

|  | Ant-v4 | Humanoid-v4 | HalfCheetah-v4 | Hopper-v4 | Walker2d-v4 |
|---|---|---|---|---|---|
| BPQL (CDA 4) | $3076.02 \pm 443.41$ | $2287.35 \pm 1047.81$ | $3740.65 \pm 340.21$ | $1097.92 \pm 610.78$ | $2317.55 \pm 499.71$ |
| BPQL (CDA 5) | $3127.63 \pm 440.96$ | $2204.08 \pm 1004.49$ | $4102.93 \pm 555.25$ | $603.92 \pm 114.92$ | $2704.21 \pm 746.97$ |
| VDPO (CDA 4) | $3240.02 \pm 1026.94$ | $2725.54 \pm 1404.37$ | $5327.23 \pm 317.71$ | $1254.58 \pm 513.41$ | $2608.30 \pm 2130.91$ |
| VDPO (CDA 5) | $3439.84 \pm 1045.19$ | $2072.25 \pm 1421.87$ | $4950.22 \pm 602.94$ | $1507.23 \pm 955.98$ | $2093.83 \pm 1835.53$ |
| ACDA | $\mathbf{4112.78 \pm 818.44}$ | $\mathbf{4608.76 \pm 1084.52}$ | $\mathbf{5984.25 \pm 1885.78}$ | $\mathbf{2094.65 \pm 944.20}$ | $\mathbf{3863.59 \pm 232.52}$ |
| BPQL w/ MDA (CDA 4) | $918.30 \pm 5.80$ | $1971.09 \pm 1698.64$ | $4826.55 \pm 339.02$ | $951.58 \pm 241.48$ | $2211.10 \pm 1042.52$ |
| BPQL w/ MDA (CDA 5) | $881.15 \pm 7.17$ | $949.03 \pm 282.09$ | $3886.75 \pm 289.39$ | $726.35 \pm 189.88$ | $2219.83 \pm 967.03$ |

### E.4.2 GE$_{4,32}$ DELAY PROCESS WITH AVERAGE CDA

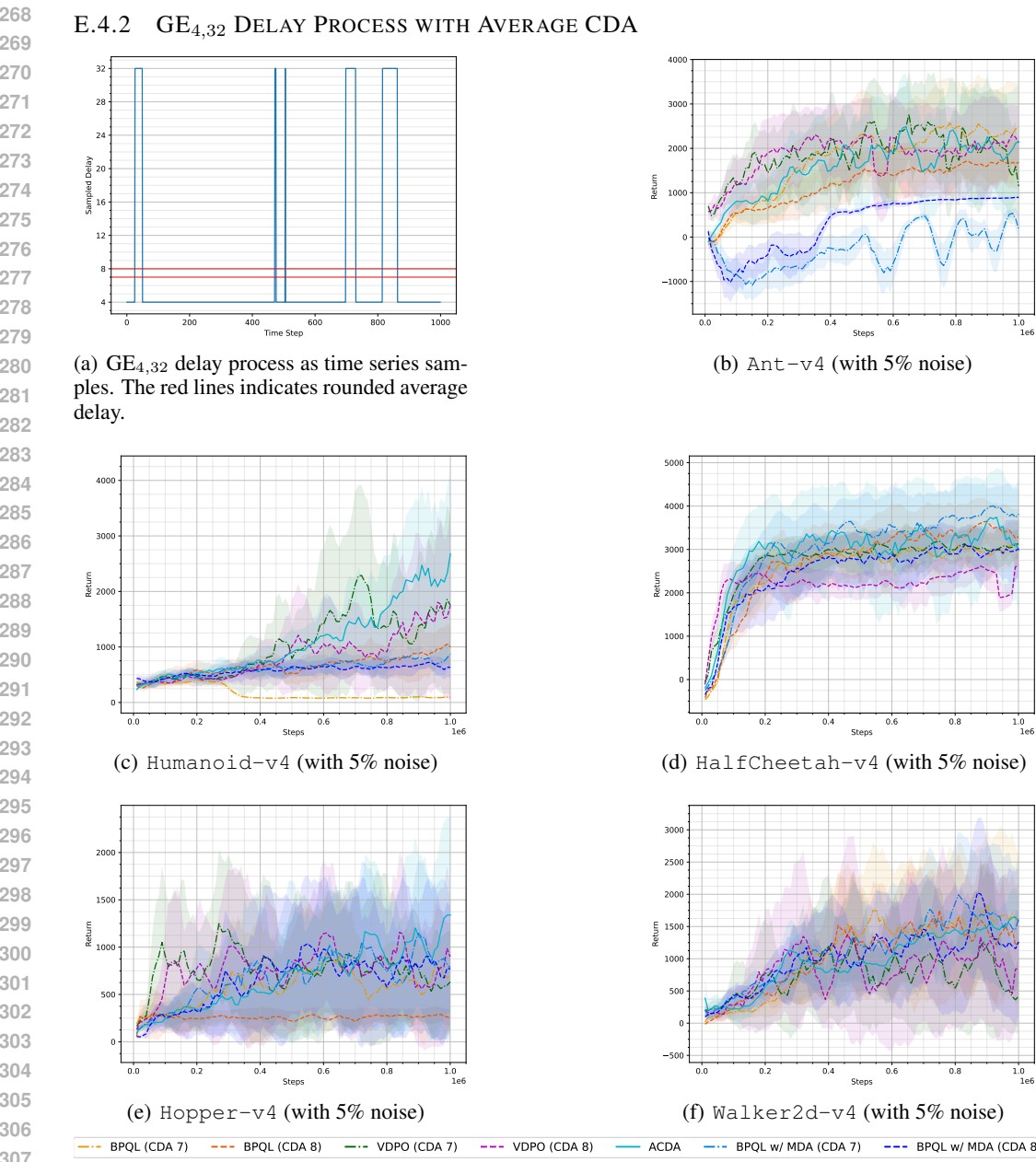

(a) GE$_{4,32}$ delay process as time series samples. The red lines indicates rounded average delay.

(b) `Ant-v4` (with 5% noise)

(c) `Humanoid-v4` (with 5% noise)

(d) `HalfCheetah-v4` (with 5% noise)

(e) `Hopper-v4` (with 5% noise)

(f) `Walker2d-v4` (with 5% noise)

BPQL (CDA 7) — BPQL (CDA 8) — VDPO (CDA 7) — VDPO (CDA 8) — ACDA — BPQL w/ MDA (CDA 7) — BPQL w/ MDA (CDA 8)

Figure 23: Time series evaluation during training on the GE$_{4,32}$ delay process for average delay assumptions of $h = 7$ and $h = 8$. All environments have added 5% noise to the actions.

Table 18: Best returns from the GE$_{4,32}$ delay process. Average delay results below the solid line.

| | Ant-v4 | Humanoid-v4 | HalfCheetah-v4 | Hopper-v4 | Walker2d-v4 |
|---|---|---|---|---|---|
| BPQL (CDA 7) | $3295.31 \pm 500.78$ | $466.90 \pm 92.62$ | $3363.96 \pm 440.71$ | $1279.14 \pm 751.20$ | $\mathbf{2573.74 \pm 1335.71}$ |
| BPQL (CDA 8) | $1936.41 \pm 141.68$ | $1168.42 \pm 247.85$ | $3815.76 \pm 390.89$ | $477.15 \pm 319.04$ | $2112.47 \pm 938.36$ |
| VDPO (CDA 7) | $\mathbf{3449.78 \pm 216.11}$ | $3343.71 \pm 1853.87$ | $3465.86 \pm 255.54$ | $1577.83 \pm 561.66$ | $1845.23 \pm 1890.44$ |
| VDPO (CDA 8) | $2836.59 \pm 391.05$ | $2630.14 \pm 1755.00$ | $2743.14 \pm 304.28$ | $1673.32 \pm 765.39$ | $2443.61 \pm 1789.56$ |
| ACDA | $2866.93 \pm 1172.46$ | $\mathbf{3725.59 \pm 1513.38}$ | $\mathbf{4231.15 \pm 333.69}$ | $\mathbf{1727.79 \pm 959.50}$ | $1840.58 \pm 386.78$ |
| BPQL w/ MDA (CDA 7) | $639.19 \pm 50.33$ | $972.62 \pm 198.05$ | $4149.78 \pm 286.66$ | $1430.83 \pm 802.87$ | $2279.09 \pm 987.54$ |
| BPQL w/ MDA (CDA 8) | $917.01 \pm 30.49$ | $841.34 \pm 158.81$ | $3324.03 \pm 363.93$ | $1664.18 \pm 732.19$ | $2456.04 \pm 1239.15$ |

### E.4.3 M/M/1 Queue Delay Process with Average CDA

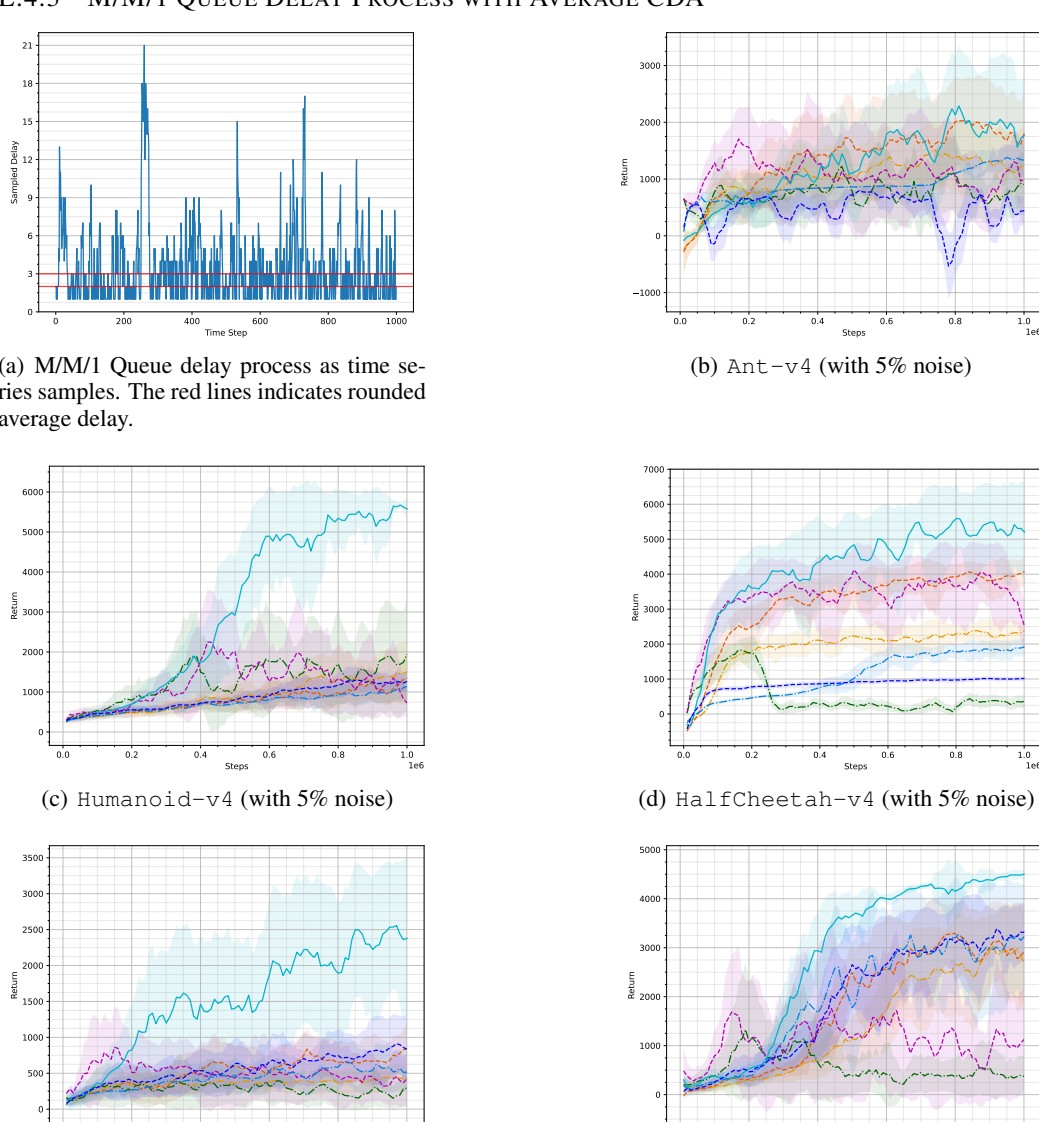

(a) M/M/1 Queue delay process as time series samples. The red lines indicates rounded average delay.

(b) `Ant-v4` (with 5% noise)

(c) `Humanoid-v4` (with 5% noise)

(d) `HalfCheetah-v4` (with 5% noise)

(e) `Hopper-v4` (with 5% noise)

(f) `Walker2d-v4` (with 5% noise)

BPQL (CDA 7) ··· BPQL (CDA 8) ··· VDPO (CDA 7) ··· VDPO (CDA 8) ── ACDA ─·· BPQL w/ MDA (CDA 7) ── BPQL w/ MDA (CDA 8)

Figure 24: Time series evaluation during training on the MM1 delay process for average delay assumptions of $h = 2$ and $h = 3$. All environments have added 5% noise to the actions.

Table 19: Best returns from the MM1 delay process. Average delay results below the solid line.

|  | Ant-v4 | Humanoid-v4 | HalfCheetah-v4 | Hopper-v4 | Walker2d-v4 |
|---|---|---|---|---|---|
| BPQL (CDA 2) | 1675.16 ± 613.61 | 1728.83 ± 946.59 | 2509.78 ± 159.44 | 518.81 ± 131.09 | 3377.82 ± 174.89 |
| BPQL (CDA 3) | 2504.20 ± 399.72 | 1507.54 ± 653.85 | 4204.65 ± 295.58 | 980.68 ± 331.18 | 3639.76 ± 106.58 |
| VDPO (CDA 2) | 1649.20 ± 522.32 | 2856.47 ± 1621.48 | 1937.06 ± 351.08 | 703.85 ± 558.47 | 1848.79 ± 1572.29 |
| VDPO (CDA 3) | 2187.20 ± 386.29 | 3217.84 ± 1761.53 | 4469.93 ± 489.60 | 1249.77 ± 716.85 | 2778.95 ± 1371.81 |
| ACDA | **2898.46 ± 838.07** | **5805.60 ± 23.04** | **5898.36 ± 409.10** | **3122.53 ± 417.37** | **4562.33 ± 87.98** |
| BPQL w/ MDA (CDA 2) | 1475.42 ± 31.45 | 1370.12 ± 474.63 | 2058.15 ± 133.25 | 864.57 ± 569.87 | 3892.97 ± 84.39 |
| BPQL w/ MDA (CDA 3) | 815.85 ± 5.94 | 1443.72 ± 299.19 | 1040.09 ± 46.75 | 994.34 ± 348.21 | 3541.48 ± 44.42 |

## E.5 RESULTS FROM ALL EVALUATED BASELINES

This section presents all results for every evaluated baseline, allowing for a wholistic comparison of performance. These results are previously presented in Appendices E.1, E.2, E.3, and E.4.

### E.5.1 GE$_{1,23}$ DELAY PROCESS FOR ALL BASELINES

Table 20: Best returns under the GE$_{1,23}$ delay process for all baselines.

| | Ant-v4 | Humanoid-v4 | HalfCheetah-v4 | Hopper-v4 | Walker2d-v4 |
|---|---|---|---|---|---|
| SAC | 14.22 ± 14.89 | 862.18 ± 266.21 | 2064.18 ± 223.48 | 306.91 ± 51.26 | 708.33 ± 221.53 |
| SAC w/ CDA | 69.28 ± 114.47 | 414.05 ± 204.20 | 128.47 ± 9.55 | 426.92 ± 27.93 | 428.44 ± 509.44 |
| Dreamer | 1111.73 ± 412.35 | 1463.07 ± 649.56 | 1796.07 ± 381.47 | 334.30 ± 245.42 | 1081.12 ± 905.94 |
| BPQL (Low CDA) | 3359.65 ± 288.24 | 2469.96 ± 1375.23 | 5944.67 ± 395.56 | 642.54 ± 420.04 | 2043.32 ± 923.00 |
| BPQL (High CDA) | 2691.88 ± 129.84 | 585.19 ± 163.49 | 4320.20 ± 1028.52 | 1328.71 ± 937.67 | 1215.91 ± 776.93 |
| BPQL (CDA 4) | 3076.02 ± 443.41 | 2287.35 ± 1047.81 | 3740.65 ± 340.21 | 1097.92 ± 610.78 | 2317.55 ± 499.71 |
| BPQL (CDA 5) | 3127.63 ± 440.96 | 2204.08 ± 1004.49 | 4102.93 ± 555.25 | 603.92 ± 114.92 | 2704.21 ± 746.97 |
| VDPO (Low CDA) | 3103.10 ± 252.19 | 2658.48 ± 2044.40 | 5625.06 ± 524.23 | 1113.51 ± 836.72 | 2756.73 ± 1693.64 |
| VDPO (High CDA) | 2163.00 ± 53.04 | 417.25 ± 210.09 | 3144.23 ± 1156.52 | 709.20 ± 522.01 | 846.88 ± 808.67 |
| VDPO (CDA 4) | 3240.02 ± 1026.94 | 2725.54 ± 1404.37 | 5327.23 ± 317.71 | 1254.58 ± 513.41 | 2608.30 ± 2130.91 |
| VDPO (CDA 5) | 3439.84 ± 1045.19 | 2072.25 ± 1421.87 | 4950.22 ± 602.94 | 1507.23 ± 955.98 | 2093.83 ± 1835.53 |
| BPQL w/ MDA (Low CDA) | 423.46 ± 73.63 | 1543.10 ± 536.14 | 5710.20 ± 566.48 | 545.01 ± 116.07 | 1923.28 ± 838.62 |
| BPQL w/ MDA (High CDA) | 1795.29 ± 23.78 | 563.36 ± 96.11 | 4926.36 ± 60.08 | 465.14 ± 138.86 | 3681.39 ± 126.41 |
| BPQL w/ MDA (CDA 4) | 918.30 ± 5.80 | 1971.09 ± 1698.64 | 4826.55 ± 339.02 | 951.58 ± 241.48 | 2211.10 ± 1042.52 |
| BPQL w/ MDA (CDA 5) | 881.15 ± 7.17 | 949.03 ± 282.09 | 3886.75 ± 289.39 | 726.35 ± 189.88 | 2219.83 ± 967.03 |
| DCAC | 949.97 ± 11.87 | 128.47 ± 36.09 | 920.09 ± 33.05 | 16.99 ± 15.94 | 106.70 ± 53.84 |
| ACDA | **4112.78 ± 818.44** | **4608.76 ± 1084.52** | **5984.25 ± 1885.78** | **2094.65 ± 944.20** | **3863.59 ± 232.52** |

## E.5.2 $\text{GE}_{4,32}$ Delay Process for All Baselines

Table 21: Best returns under the $\text{GE}_{4,32}$ delay process for all baselines.

| | Ant-v4 | Humanoid-v4 | HalfCheetah-v4 | Hopper-v4 | Walker2d-v4 |
|---|---|---|---|---|---|
| SAC | $-5.72 \pm 19.62$ | $494.43 \pm 156.01$ | $-158.78 \pm 65.86$ | $279.74 \pm 109.83$ | $60.86 \pm 75.59$ |
| SAC w/ CDA | $18.93 \pm 23.64$ | $230.45 \pm 99.22$ | $591.32 \pm 36.39$ | $315.47 \pm 51.49$ | $257.18 \pm 73.42$ |
| Dreamer | $1147.56 \pm 371.15$ | $1091.48 \pm 577.52$ | $2493.19 \pm 231.22$ | $515.36 \pm 430.72$ | $1233.79 \pm 802.47$ |
| BPQL (Low CDA) | $3000.00 \pm 754.09$ | $2949.17 \pm 1933.62$ | $5315.27 \pm 559.25$ | $1243.58 \pm 704.53$ | $2107.39 \pm 1219.55$ |
| BPQL (High CDA) | $2509.52 \pm 117.37$ | $276.63 \pm 131.70$ | $2136.36 \pm 547.04$ | $433.29 \pm 381.79$ | $875.09 \pm 747.72$ |
| BPQL (CDA 7) | $3295.31 \pm 500.78$ | $466.90 \pm 92.62$ | $3363.96 \pm 440.71$ | $1279.14 \pm 751.20$ | $2573.74 \pm 1335.71$ |
| BPQL (CDA 8) | $1936.41 \pm 141.68$ | $1168.42 \pm 247.85$ | $3815.76 \pm 390.89$ | $477.15 \pm 319.04$ | $2112.47 \pm 938.36$ |
| VDPO (Low CDA) | $\mathbf{3979.56 \pm 331.76}$ | $2956.52 \pm 2322.74$ | $\mathbf{5424.96 \pm 306.06}$ | $1360.87 \pm 627.11$ | $2234.83 \pm 1776.21$ |
| VDPO (High CDA) | $2266.99 \pm 90.89$ | $280.72 \pm 169.85$ | $3664.30 \pm 929.25$ | $330.44 \pm 263.74$ | $344.73 \pm 316.82$ |
| VDPO (CDA 7) | $3449.78 \pm 216.11$ | $3343.71 \pm 1853.87$ | $3465.86 \pm 255.54$ | $1577.83 \pm 561.66$ | $1845.23 \pm 1890.44$ |
| VDPO (CDA 8) | $2836.59 \pm 391.05$ | $2630.14 \pm 1755.00$ | $2743.14 \pm 304.28$ | $1673.32 \pm 765.39$ | $2443.61 \pm 1789.56$ |
| BPQL w/ MDA (Low CDA) | $2155.47 \pm 84.22$ | $465.58 \pm 82.59$ | $4082.54 \pm 411.47$ | $1312.37 \pm 868.12$ | $2355.23 \pm 1145.73$ |
| BPQL w/ MDA (High CDA) | $1661.41 \pm 43.42$ | $359.46 \pm 156.86$ | $3609.45 \pm 328.37$ | $224.34 \pm 90.12$ | $\mathbf{3015.45 \pm 1428.05}$ |
| BPQL w/ MDA (CDA 7) | $639.19 \pm 50.33$ | $972.62 \pm 198.05$ | $4149.78 \pm 286.66$ | $1430.83 \pm 802.87$ | $2279.09 \pm 987.54$ |
| BPQL w/ MDA (CDA 8) | $917.01 \pm 30.49$ | $841.34 \pm 158.81$ | $3324.03 \pm 363.93$ | $1664.18 \pm 732.19$ | $2456.04 \pm 1239.15$ |
| DCAC | $953.14 \pm 12.06$ | $167.97 \pm 82.14$ | $1123.47 \pm 100.34$ | $57.98 \pm 28.04$ | $9.23 \pm 20.35$ |
| ACDA | $2866.93 \pm 1172.46$ | $\mathbf{3725.59 \pm 1513.38}$ | $4231.15 \pm 333.69$ | $\mathbf{1727.79 \pm 959.50}$ | $1840.58 \pm 386.78$ |

## E.5.3 M/M/1 Queue Delay Process for All Baselines

Table 22: Best returns under the MM1 delay process for all baselines.

| | Ant-v4 | Humanoid-v4 | HalfCheetah-v4 | Hopper-v4 | Walker2d-v4 |
|---|---|---|---|---|---|
| SAC | $-0.58 \pm 8.66$ | $921.04 \pm 299.47$ | $20.69 \pm 94.91$ | $333.06 \pm 96.04$ | $604.80 \pm 212.37$ |
| SAC w/ CDA | $102.00 \pm 33.77$ | $613.03 \pm 157.68$ | $550.84 \pm 16.28$ | $627.59 \pm 24.62$ | $2005.76 \pm 341.30$ |
| Dreamer | $1121.11 \pm 58.69$ | $981.38 \pm 597.01$ | $584.40 \pm 72.26$ | $975.72 \pm 650.05$ | $1801.81 \pm 1158.73$ |
| BPQL (Low CDA) | $2577.34 \pm 1217.11$ | $4158.16 \pm 981.22$ | $4478.64 \pm 193.82$ | $407.91 \pm 179.00$ | $3475.80 \pm 586.89$ |
| BPQL (High CDA) | $\mathbf{3074.17 \pm 106.78}$ | $5435.29 \pm 68.34$ | $4660.93 \pm 448.10$ | $3035.66 \pm 103.80$ | $3547.73 \pm 133.51$ |
| BPQL (CDA 2) | $1675.16 \pm 613.61$ | $1728.83 \pm 946.59$ | $2509.78 \pm 159.44$ | $518.81 \pm 131.09$ | $3377.82 \pm 174.89$ |
| BPQL (CDA 3) | $2504.20 \pm 399.72$ | $1507.54 \pm 653.85$ | $4204.65 \pm 295.58$ | $980.68 \pm 331.18$ | $3639.76 \pm 106.58$ |
| VDPO (Low CDA) | $2278.08 \pm 726.85$ | $3349.31 \pm 1668.64$ | $4857.65 \pm 335.88$ | $1628.83 \pm 880.65$ | $3554.77 \pm 1278.22$ |
| VDPO (High CDA) | $2528.67 \pm 144.63$ | $720.73 \pm 634.35$ | $3831.96 \pm 960.07$ | $1459.88 \pm 933.11$ | $2144.25 \pm 1650.85$ |
| VDPO (CDA 2) | $1649.20 \pm 522.32$ | $2856.47 \pm 1621.48$ | $1937.06 \pm 351.08$ | $703.85 \pm 558.47$ | $1848.79 \pm 1572.29$ |
| VDPO (CDA 3) | $2187.20 \pm 386.29$ | $3217.84 \pm 1761.53$ | $4469.93 \pm 489.60$ | $1249.77 \pm 716.85$ | $2778.95 \pm 1371.81$ |
| BPQL w/ MDA (Low CDA) | $1541.32 \pm 16.71$ | $1030.55 \pm 185.86$ | $4502.92 \pm 214.25$ | $1665.66 \pm 1210.69$ | $\mathbf{4825.75 \pm 126.28}$ |
| BPQL w/ MDA (High CDA) | $2308.18 \pm 75.13$ | $944.56 \pm 92.79$ | $3953.69 \pm 351.34$ | $512.43 \pm 346.18$ | $3667.61 \pm 142.48$ |
| BPQL w/ MDA (CDA 2) | $1475.42 \pm 31.45$ | $1370.12 \pm 474.63$ | $2058.15 \pm 133.25$ | $864.57 \pm 569.87$ | $3892.97 \pm 84.39$ |
| BPQL w/ MDA (CDA 3) | $815.85 \pm 5.94$ | $1443.72 \pm 299.19$ | $1040.09 \pm 46.75$ | $994.34 \pm 348.21$ | $3541.48 \pm 44.42$ |
| DCAC | $959.23 \pm 13.54$ | $525.85 \pm 135.36$ | $35.60 \pm 21.42$ | $1026.45 \pm 2.96$ | $24.48 \pm 45.46$ |
| ACDA | $2898.46 \pm 838.07$ | $\mathbf{5805.60 \pm 23.04}$ | $\mathbf{5898.36 \pm 409.10}$ | $\mathbf{3122.53 \pm 417.37}$ | $4562.33 \pm 87.98$ |

## F  PRACTICAL CONSIDERATIONS OF THE INTERACTION LAYER

This section discusses practical considerations when deploying ACDA and the interaction layer to real-world environments. Specifically, we discuss the delays not handled by the interaction layer in Appendix F.1, the effect that ACDA has on computational delays in Appendix F.2, how to mitigate computational delays in Appendix F.3, and the effect that ACDA has on transmission bandwidth in Appendix F.4.

### F.1  CONSIDERATIONS FOR NON-INTERACTION DELAYS

As illustrated in Figure 1 in the introduction, there are additional delays not handled by the interaction layer. Namely, the delays between the interaction layer and the system itself. In this paper, we presume that these delays are negligible or otherwise accounted for. If these delays are not negligible and must be accounted for, then this can most likely be handled by prematurely sensing and actuating the system. As the interaction layer by design is located close to the excited system, any delays between them will likely take place over controlled channels (such as USB or SPI), meaning that any delay over these channels is stable.

### F.2  EFFECT OF ACTION PACKET ON COMPUTATIONAL DELAY

Although ACDA handles interaction delays, the computation of the action packet matrix itself does add computational delay, compared to constant-delay approaches that only compute a single action. This could cause concern for real-world applications if this additional delay is too significant. If the computational delay is longer than the excitation period of the environment, the policy cannot generate actions fast enough, and the interaction will stall. In this section, we discuss the effect that the computation of the action packet can have on the delay, present measurements of computational delay, and put that into the context of the evaluated environments.

There are a few remarks about the action packet itself:

- In ACDA, the computation of the rows is done in parallel. The effective computational time is linear instead of quadratic. More specifically, the number of sequential computations is proportional to the sum of the horizon and the prediction length, $h + L$.
- The action packet contains horizons of actions, allowing for gaps in the interaction. For a practical scenario, in the event of not having enough time to compute subsequent action packets, it is possible to only generate action packets based on the latest observation packet that has reached the agent.

With this in mind, we measure the average computational time of action packets for the network structure used when evaluating the `Ant-v4` environment under the MM1 delay process. We measure the time taken in the training loop to generate a single action packet, as well as the execution time of the individual network components. All measurements are done in our framework implemented in PyTorch, collected on the same system used to run the benchmarks.

Table 23: Execution time measurements.

| Aspect | Measured Time |
|---|---:|
| Generating a random action packet | 24 ms |
| Generating an action packet using the MDA | 68 ms |
| Randomly filling an action packet matrix | 39 $\mu$s |
| Single GRU forward pass | 164 $\mu$s |
| Single policy forward pass | 61 $\mu$s |

A notable aspect of the measurements in Table 23 is the difference in generating a random action packet in the training loop, compared to directly filling an action packet matrix in PyTorch (24 ms vs 61 $\mu$s). This hints at that our current implementation is not optimized, that there are further gains to

be made, and that $68$ ms is not a representative time for generating an action packet in an optimized implementation.

The worst-case computation for a row in the action packet under the MM1 delay process is for the 16th row (delay 16). This consists of first embedding the observed state, then embedding the 16 guessed preceding actions into a distribution, and then sequentially generating 16 actions and embedding the next distribution, excluding the distribution after the final action, as that is not needed. The effective computation time for the action packet is then as shown in Equation 28:

$$t_{\text{action packet}} = t_{\text{embed}} + 16 \cdot t_{\text{GRU}} + 15 \cdot (t_{\text{policy}} + t_{\text{GRU}}) + t_{\text{policy}} \tag{28}$$

As the embedding network is comparable in size to the policy network, we use the time for the policy as a proxy for the state embedding. Inserting the measurements from Table 23 into the formula from Equation 28, we get an estimated average execution time of $6.1$ ms for an action packet. In the context of the `Ant-v4`, environment which uses a $50$ ms actuation period, the computational delay is less than the time for a single step in the environment. The computational delay is also lower than the smallest actuation period for any environment used in the evaluation, the smallest period being $8$ ms for the `Walker2d-v4` environment.

## F.3 COMPENSATING FOR EXCESSIVE COMPUTATIONAL DELAYS

Even though the expected computational delay estimated in Appendix F.2 is smaller than the actuation periods of the environments evaluated in this paper, other environments may require an even shorter actuation period that is smaller than the computational delay. This shorter period will cause the ACDA agent to stall, as it cannot produce action packets faster than the rate at which observation packets arrive.

A possible solution to this problem is for the ACDA agent to compute action packets based solely on the last received observation packet and to disregard all previous observation packets. These disregarded observation packets will not be used to compute any action packets, resulting in gaps in the interaction between the agent and the interaction layer. However, as each row in the matrix of an action packet describes a horizon of actions, ACDA already provides a solution for filling in these gaps in the interaction. The only aspect that has to change in the original algorithm is to make the memorized action prediction aware that a packet has been skipped. This process, which we refer to as *packet skipping*, is illustrated in Figure 25.

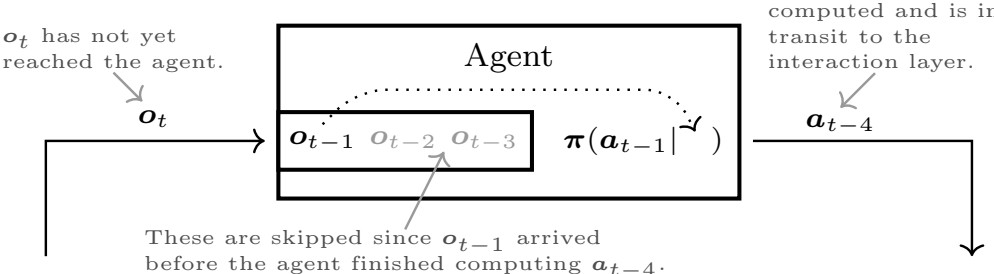

Figure 25: Illustration of packet skipping when observations are reaching the agent faster than it can compute action packets. In the illustrated example, while $a_{t-4}$ was being computed, $o_{t-3}$, $o_{t-2}$, and $o_{t-1}$ had arrived at the agent. The action packets $a_{t-3}$ and $a_{t-2}$ are never computed, and the agent directly starts to compute $a_{t-1}$ instead.

To evaluate the plausibility of packet skipping, we train and evaluate ACDA when an action packet is computed based on every 2nd, 4th, and 8th incoming observation packet. These results are presented in Table 24, where they are put into the context of the results from the main body of the paper.

Although there is a noticeable drop-off in performance for some benchmarks, such as `Humanoid-v4` and `Hopper-v4` under MM1 delays, ACDA still manages to achieve good performance in most benchmarks, even when skipping the computation of action packets. It should also be

Table 24: Best average return when only every 2nd, 4th, and 8th observation packet is used by ACDA to compute an action packet.

| Gymnasium env. | Ant-v4 | | | Humanoid-v4 | | | HalfCheetah-v4 | | | Hopper-v4 | | | Walker2d-v4 | | |
|---|---|---|---|---|---|---|---|---|---|---|---|---|---|---|---|
| Delay process | $GE_{1,23}$ | $GE_{4,32}$ | MM1 | $GE_{1,23}$ | $GE_{4,32}$ | MM1 | $GE_{1,23}$ | $GE_{4,32}$ | MM1 | $GE_{1,23}$ | $GE_{4,32}$ | MM1 | $GE_{1,23}$ | $GE_{4,32}$ | MM1 |
| SAC | 14.22 | −5.72 | −0.58 | 862.18 | 494.43 | 921.04 | 2064.18 | −158.78 | 20.69 | 306.91 | 279.74 | 333.06 | 708.33 | 60.86 | 604.80 |
| SAC w/ CDA | 69.28 | 18.93 | 102.00 | 414.05 | 230.45 | 613.03 | 128.47 | 591.32 | 550.84 | 426.92 | 315.47 | 627.59 | 428.44 | 257.18 | 2005.76 |
| Dreamer | 1111.73 | 1147.56 | 1121.11 | 1463.07 | 1091.48 | 981.38 | 1796.07 | 2493.19 | 584.40 | 334.30 | 515.36 | 975.72 | 1081.12 | 1233.79 | 1801.81 |
| BPQL | 2691.88 | 2509.52 | **3074.17** | 585.19 | 276.63 | 5435.29 | 4320.20 | 2136.36 | 4660.93 | 1328.71 | 433.29 | 3035.66 | 1215.91 | 875.09 | 3547.73 |
| VDPO | 2163.00 | 2266.99 | 2528.67 | 417.25 | 280.72 | 720.73 | 3144.23 | 3664.30 | 3831.96 | 709.20 | 330.44 | 1459.88 | 846.88 | 344.73 | 2144.25 |
| DCAC | 949.97 | 953.14 | 959.23 | 128.47 | 167.97 | 525.85 | 920.09 | 1123.47 | 35.60 | 16.99 | 57.98 | 1026.45 | 106.70 | 9.23 | 24.48 |
| ACDA | **4112.78** | 2866.93 | 2898.46 | **4608.76** | 3725.59 | **5805.60** | 5984.25 | **4231.15** | **5898.36** | 2094.65 | **1727.79** | **3122.53** | 3863.59 | 1840.58 | **4562.33** |
| ACDA (every 2nd) | 3760.33 | **3215.65** | 2932.27 | 4108.88 | **3986.12** | 3270.50 | **6205.55** | 4026.24 | 3773.21 | 1983.83 | 1533.11 | 2022.95 | 3697.84 | **3790.17** | 3818.12 |
| ACDA (every 4th) | 3341.16 | 3180.04 | 2448.11 | 4044.15 | 2342.79 | 3688.68 | 4949.92 | 3838.81 | 3653.31 | 1940.75 | 1710.44 | 1383.08 | **4634.92** | 1682.69 | 3944.06 |
| ACDA (every 8th) | 3263.72 | 2246.90 | 2588.54 | 4311.94 | 2672.57 | 1315.06 | 4311.85 | 3418.11 | 3334.57 | **2132.22** | 1493.36 | 829.54 | 3379.25 | 3552.03 | 4519.21 |

noted that we do not believe the computational delay to be a practical problem for these benchmarks, and that scenarios such as only using every 8th incoming observation packet are excessive; however, these results demonstrate the viability of the approach. It should also be noted that although some benchmarks, such as `Walker2d-v4` under $GE_{4,32}$, exhibit high variance, ACDA still achieves a high average return across all benchmarks.

## F.4 EFFECT OF ACTION PACKET ON COMMUNICATION

While the action packet enables adaptability to interaction delays, it also introduces bandwidth overhead due to sending a matrix of actions rather than a single action. The worst-case bandwidth overhead in the evaluated environments is in the `Humanoid-v4` under the $GE_{4,32}$ delay, where each action in the $32 \times 32$ action packet matrix consists of 17 32-bit floating point numbers, which are sent to the interaction layer every 15 ms. This benchmark requires a bandwidth of approximately 4.5 MiB/s, as opposed to sending a single action, which only requires a bandwidth of 4.5 kiB/s. While many communication channels, such as WiFi, support bandwidths well above 4.5 MiB/s, this becomes problematic for low-bandwidth communication channels such as Bluetooth.

$$32 \cdot 32 \cdot 17 \cdot 4 \cdot \frac{1000}{15} \approx 4.43 \cdot 2^{20} \tag{29}$$

We do not attempt to optimize the bandwidth in this paper; instead, we assume it is sufficient for all our benchmarks. It is possible to implement methods on the current interaction layer framework that mitigate this issue, such as the action skipping method described in Appendix F.3. It is also possible to reduce the size of the action packet by reducing the action buffer horizon $h$, which will not impact performance unless the gaps in the interaction exceed $h$.

There is also room for alterations to the interaction layer setup to reduce the size of the action packets, while still providing the adaptability to random delays. In the case of $GE_{4,32}$, only rows 4 and 32 in the action packet matrix will ever be selected. Therefore, we could optimize the interaction by only sending the rows for the delays we believe will actually occur, discarding or artificially delaying the packet if it did not contain a row for the actual delay.

# G INTERACTION-DELAYED REINFORCEMENT LEARNING WITH REAL-VALUED DELAY

In Section 3.1, we introduced the delayed MDP using discrete delays measured in steps within the MDP. For real systems described by an MDP, each step corresponds to some amount of real-valued time, possibly controlled by a clock. Any interaction delay with the real system will also correspond to some amount of real-valued time that does not necessarily align with the time taken for a step in the MDP. Therefore, it makes sense to talk about delay directly as real-valued time when considering interaction delays for systems in the real world.

In Appendix G.1, we describe the effect that delays have when they are described as real-valued delays. We describe in Appendix G.2 how to implement the interaction layer to handle these real-valued delays.

## G.1 ORIGIN AND EFFECT OF DELAY AS CONTINUOUS TIME

MDPs usually assign a time $t$ to states, actions, and rewards. This time $t \in \mathbb{N}$ is merely a discrete ordering of events. We model the origin of delays in the real world as elapsed wall clock time in the domain of $\mathbb{R}^+$. We use the following notation to distinguish between them:

$$t \in \mathbb{N} \qquad \text{(Order of events in MDP.)} \tag{30}$$

$$\tau \in \mathbb{R}^+ \qquad \text{(Wall clock time elapsed in the real world.)} \tag{31}$$

In the real world, there is an *interaction delay* in that it takes some time $\tau_{\text{observe}} \in \mathbb{R}^+$ to observe a state, some time $\tau_{\text{decide}} \in \mathbb{R}^+$ to generate the action, and some time $\tau_{\text{apply}} \in \mathbb{R}^+$ to apply the action to the environment. In this time, the state $s$ may evolve independently of an action being applied to the environment. Let this evolution process $\Delta$ be defined as

$$\Delta : S \times \mathbb{R}^+ \times S \to \mathbb{R} \tag{32}$$

$$\text{such that} \quad \widetilde{s} \sim \Delta(\cdot|s, \tau) \tag{33}$$

$$\text{subject to} \quad \forall s, \widetilde{s}, \tau_1, \tau_2 : \quad \Delta(\widetilde{s}|\widetilde{s}_{\text{i}}, \tau_2)\big|_{\widetilde{s}_{\text{i}} \sim \Delta(\cdot|s, \tau_1)} = \Delta(\widetilde{s}|s, \tau_1 + \tau_2) \tag{34}$$

where $s \in S$ is the state and $\tau, \tau_1, \tau_2 \in \mathbb{R}^+$ are real wall-clock times in which the state has had time to evolve. The evolved state is unknown to the agent and is thus referred to as $\widetilde{s} \in S$. The criterion in Equation 34 formally states that it should make no difference whether a state evolved for a single time period or if it is split into 2 time periods.

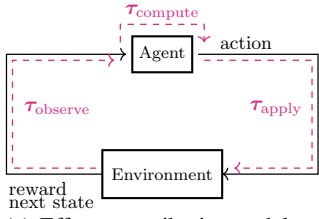

(a) Effects contributing to delay.

Standard (Assumed) RL Interaction:
$a \sim \pi(\cdot|s)$
$s' \sim p(\cdot|s, a)$

With Interaction Delay:
$a \sim \pi(\cdot|s)$
$\widetilde{s}_3 \sim \Delta(\cdot|s, \tau_{\text{observe}} + \tau_{\text{decide}} + \tau_{\text{apply}})$
$s' \sim p(\cdot|\widetilde{s}_3, a)$

(b) Violation of RL interaction assumption.

Figure 26: How delays violate the assumption used by state-of-the-art RL algorithms.

If the environment is sufficiently static, like a chess board, then this poses no issue because that $\Delta(\cdot|s, \tau)$ will always evolve to the same state. If the environment is more dynamic, such as balancing

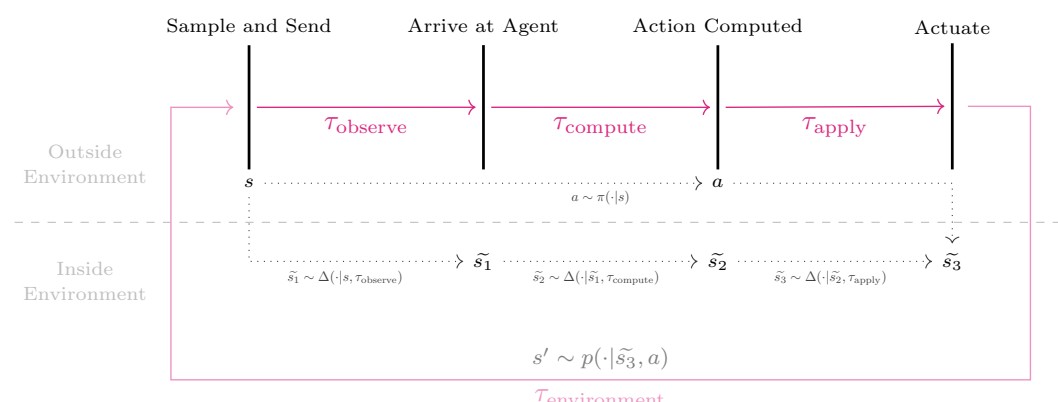

Figure 27: State evolution over the interaction process.

an inverted pendulum, then the interaction delay can result in the state we apply an action to has changed from the state that it was generated from. We illustrate the factors contributing to the interaction delay in Figure 26(a) and how they affect the evolving state in Figure 27. The violation of the equations is shown in Figure 26(b).

While there likely is some time passing within the environment in the real world (as illustrated by $\tau_{\text{environment}}$ in Figure 27), we consider that time $\tau_{\text{environment}}$ as part of the environment dynamics and not of the interaction delay.

These times may also be stochastic and unknown to the agent before generating the action. While they can be assumed identically and independently distributed (iid), effects such as clogging (over network or computation bandwidth) mean that a long delay is more likely to follow another long delay, resulting in a dependence in distributions. Delays can also be affected by how an agent interacts with the environment, for example, by controlling a system such that it moves to another access point on the network.

### G.2 INTERACTION LAYER TO ENFORCE DISCRETE DELAY

If we assume that the environment will be excited every $\tau_{\text{environment}}$ seconds, then we can express the delay as a discrete number of steps, rounded up to the nearest multiple of $\tau_{\text{environment}}$.

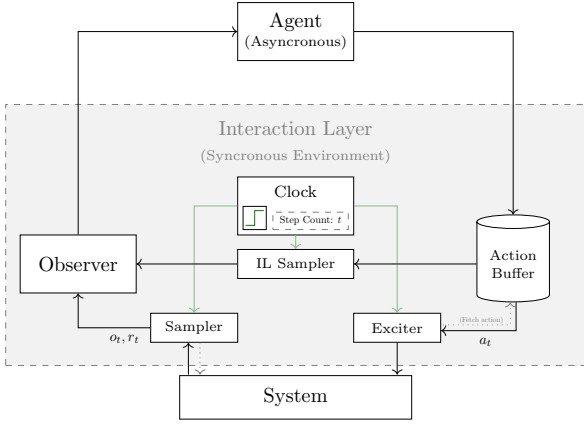

Figure 28: Illustration of the components that make up the interaction layer. A clock is used to ensure that the system is excited and sampled every $\tau_{\text{environment}}$.

Delay being expressed in real time makes it inconvenient to reason about with respect to the MDP describing the environment interaction. To resolve this, we introduce an *interaction layer* that sits

between the agent and the system we want to control. The primary role of the interaction layer is to discretize time and ensure that $\tau_{\text{environment}}$ is constant. It operates under the assumptions that the interaction layer can

1. observe the system at any time (read sensors),
2. excite the system at any time (apply actions), and
3. observe and excite with negligible real-world delay (assumed $\tau = 0$).

Under these assumptions, the role of the interaction layer is primarily to

1. maintain an *action buffer* of upcoming actions to apply to the system,
2. accept incoming actions from an agent and insert them into the *action buffer*,
3. ensure that interaction with the system occurs periodically on a fixed interval, and
4. transmit state information back to the agent.

The construction of an interaction layer is realistic for many real-world systems. Using the scenario illustrated in Figure 1 as an example, the interaction layer could be implemented as a microcontroller located on the vehicle itself. We illustrate the interaction layer in Figure 28.

From this perspective, the agent acts reactively. The interaction layer manages the interaction with the environment, and the agent generates new actions for the action buffer when triggered by emissions from the interaction layer. We denote emitted data from the interaction layer as an *observation packet* $o_t$, and the data sent to the interaction layer as an *action packet* $a_t$.

