# OpenReview forum: "Adaptive Reinforcement Learning for Unobservable Random Delays"
_ICLR.cc/2026/Conference — Submitted to ICLR 2026_

### Official Review · Reviewer_3YFw · 2025-10-17

**Soundness:** 3
**Presentation:** 3
**Contribution:** 3
**Rating:** 6
**Confidence:** 3

**Summary:**

This paper addresses the problem of reinforcement learning under random, unobservable interaction delays in cyber-physical systems. The authors propose the "interaction layer", a framework that enables agents to generate a matrix of future actions to handle time-varying delays and potential packet loss. Building on this framework, they develop ACDA (Actor-Critic with Delay Adaptation), a model-based RL algorithm that uses distribution embeddings and a heuristic for action selection under delay uncertainty. The method is evaluated on MuJoCo locomotion tasks and demonstrates superior performance compared to state-of-the-art delayed RL algorithms.

**Strengths:**

1. The paper tackles a practically important problem of unobservable random delays in real-world RL applications, moving beyond the restrictive constant-delay assumption used in prior work.

2. The interaction layer framework is well-motivated and provides a principled way to handle varying delays by generating action matrices that anticipate different delay scenarios.

3. The experimental results show consistent and substantial performance improvements over strong baselines (BPQL, VDPO, DCAC) across multiple environments and delay processes.

4. The paper provides comprehensive experimental details including rigorous ablation studies, hyperparameters, and justification for design choices such as action noise.

**Weaknesses:**

1. Limited Experimental Scope. The evaluation is restricted exclusively to MuJoCo locomotion tasks (Ant, Humanoid, HalfCheetah, Hopper, Walker2d), which represent a narrow class of continuous control problems. The generalizability to other important domains such as manipulation, navigation, discrete action spaces, or real-world robotic systems remains unclear.

2. Computational Complexity Analysis. While Appendix F.2 provides execution time measurements, the paper lacks a thorough complexity analysis comparing ACDA's computational requirements (both time and space) against baselines.

3. Heuristic Assumptions and Limitations. The heuristic in Section 4 assumes delays are temporally correlated (constant over short windows), which may not hold in many real-world scenarios such as wireless networks with bursty interference or cloud computing with variable load.

**Questions:**

1. Generalization beyond locomotion: Can you provide results or discussion on how ACDA would perform in other RL domains such as manipulation tasks, partially observable environments beyond delays, or discrete action spaces? What modifications would be necessary?

2. Action buffer horizon: How sensitive is performance to the choice of horizon h and prediction length L? Is there a principled way to set these parameters, or do they require manual tuning per environment?

---

> ### Author Response · Authors · 2025-11-24
>
> Thank you for your detailed review! You raise interesting questions and concerns, which we address below.
>
> > While Appendix F.2 provides execution time measurements, the paper lacks a thorough complexity analysis comparing ACDA's computational requirements (both time and space) against baselines.
>
> The computational complexity of ACDA is $O(h + L)$ (mentioned as "proportional to … h + L" in the text of Appendix F.2), assuming we have access to sufficiently parallel hardware. However, our focus has primarily been on demonstrating that the overhead should be manageable within our benchmarks. We have updated the paper (see new uploaded document) with Appendix F.3, which evaluates a solution for scenarios where the computational load exceeds the system's periodicity, as well as Appendix F.4, which analyzes the space requirements of the interaction layer.
>
> > The heuristic in Section 4 assumes delays are temporally correlated (constant over short windows), which may not hold in many real-world scenarios such as wireless networks with bursty interference or cloud computing with variable load.
>
> All evaluated delay processes include situations with delay bursts, and MM1 and $GE_{1,23}$ also include variable load. Despite these properties, we still observe that ACDA performs well. An interesting future direction of research is into algorithms that can adapt to independent and identically distributed (i.i.d.) delay processes.
>
> > Can you provide results or discussion on how ACDA would perform in other RL domains such as manipulation tasks, partially observable environments beyond delays, or discrete action spaces? What modifications would be necessary?
>
> The MuJoCo environments were chosen for reproducibility reasons and because they are affected by delays. ACDA is restricted to continuous action spaces because it is based on SAC. However, the overall algorithm structure of ACDA caters to many actor-critic structures, such as PPO, which could have been used instead of SAC to support discrete action spaces. We do not see ACDA being used in domains outside delayed RL; however, some of the ideas within ACDA can potentially be applied to different tasks that benefit from some form of adaptability.
>
> > Action buffer horizon: How sensitive is performance to the choice of horizon h and prediction length L? Is there a principled way to set these parameters, or do they require manual tuning per environment?
>
> The horizon $h$ specifies the capacity of the action buffer, which is part of the interaction layer and therefore fixed. A larger horizon gives the agent more control over what will happen during gaps when no new action packet arrives. A horizon that is too small makes the environment susceptible to delay spikes. The horizon $h$ should be set to the maximum expected gap in the interaction.
>
> The prediction length $L$ is, on the other hand, controlled by the agent when generating an action packet. Although ACDA keeps $L$ fixed for each delay process, it is possible for other agents to adjust $L$ dynamically during inference. We have updated the paper with additional information in Appendix B regarding how the prediction length $L$ is set for ACDA.
>
> We hope we have addressed your questions accordingly, and we are also willing to engage in further discussion.

---

> > ### Comment · Reviewer_3YFw · 2025-11-27
> >
> > Thanks for the effort in addressing my comments. I have no further questions at this point. I'll maintain my score.

---

### Official Review · Reviewer_Tjob · 2025-10-28

**Soundness:** 2
**Presentation:** 2
**Contribution:** 2
**Rating:** 4
**Confidence:** 2

**Summary:**

This work aims to solve the problem of unobservable random delays in reinforcement learning. This paper proposes a novel "interaction layer" framework, where the agent no longer generates a single action but instead produces an action matrix to dynamically adapt to different delay situations. The proposed algorithm, ACDA, utilizes this framework and has been shown in experiments to achieve significantly superior performance.

**Strengths:**

1. This paper tackles a practical problem in the field of RL with delays, that the delay is random and cannot be observed or predicted. This stands in sharp contrast to the unrealistic priori assumptions about delay found in existing works in this field. The reviewer affirms the contribution of this work.
2. The paper's experimental design is comprehensive. The authors introduce action noise to induce stochasticity into the environments. Furthermore, the results in Appendix E.2 and E.3 serve to reinforce the paper's methodological design and central claims.

**Weaknesses:**

1. The paper's most significant weakness lies in its comparative experimental design, as both the choice of baselines and their setup are unreasonable. The authors selected BPQL and VDPO, which are methods primarily designed for constant delay. However, in random delay environments, providing the worst-case delay as a priori knowledge to these algorithms is, in the reviewer's opinion, an unreasonable setup. The reviewer believes a more reasonable design would be to provide the mean of the random delay distribution as the priori delay knowledge for this class of methods; that is, the agent would be making decisions based on the average-case delay.
2. Given the paper's focus on the random delay problem, the authors have omitted a comparison with recent state-of-the-art methods that also specifically address random delays, such as DFBT [1] and State Augmentation-MLP [2]. Benchmarking against this relevant class of methods would be necessary to further substantiate the claimed effectiveness of the proposed method.
3. The paper's presentation is not good enough. The notation introduced is somewhat complex, and the process described in the text for Figures 2 and 3 is difficult to follow. For example, in Figure 2, the most recent observation received by the agent is labeled o_{t-2}, yet the description in lines 193-194 refers to o_t as the agent's observation, which is confusing. Furthermore, when Figure 3 is first introduced (lines 210-211), the definition for the counter c_t, which is essential to the figure caption, has not yet been provided, making it difficult to understand.

**Questions:**

1. In lines 418-419, the authors state, "we expect ACDA not to perform as well with the M/M/1 queue delays that fluctuate more." However, in Table 1, ACDA's performance under the MM1 Delay process is exceptionally strong, in most environments even exceeding its performance on the GE1,23 and GE4,32 processes. This result appears to contradict the authors' statement. Can the authors explain this phenomenon?
2. The parameter L does not appear to be explicitly defined in the main text. The reviewer found in Appendix F.2 that L represents the "prediction length," but how this parameter is determined is a significant concern for the reviewer. If L is set to the maximum value of the random delay, this implies an assumption that the maximum random delay is known, which would weaken the advantage of this work in handling unobservable random delays. If L is set as a fixed empirical value, the reviewer hopes the authors can provide the corresponding justification.
3. A very interesting result in Table 1 is why DCAC, a method explicitly designed for random delays, performs so poorly in this random delay environment—far worse, in fact, than the constant-delay methods BPQL and VDPO. Is this poor performance attributable to the stochastic transitions induced by the authors' use of action noise, which perhaps DCAC is not well-suited to handle?

Reference
[1] Directly Forecasting Belief for Reinforcement Learning with Delays. ICML 2025
[2] Addressing Signal Delay in Deep Reinforcement Learning. ICLR 2024

---

> ### Author Response · Authors · 2025-11-24
>
> Thank you for taking the time to review our paper. We address your concerns and questions below, and are willing to engage further should you have any additional questions.
>
> > The paper's most significant weakness lies in its comparative experimental design, as both the choice of baselines and their setup are unreasonable. The authors selected BPQL and VDPO, which are methods primarily designed for constant delay. However, in random delay environments, providing the worst-case delay as a priori knowledge to these algorithms is, in the reviewer's opinion, an unreasonable setup. The reviewer believes a more reasonable design would be to provide the mean of the random delay distribution as the priori delay knowledge for this class of methods; that is, the agent would be making decisions based on the average-case delay.
>
> In the original ICLR submission, Appendix E.3 contained evaluations that we believe would be even more favorable than the average-case delay for constant-delay baselines. However, we reran the state-of-the-art constant-delay baselines under the average-case delay for each delay process, which is located in the new Appendix E.4 (please see the new uploaded document). In most cases, we see that the constant-delay baselines perform worse under average-case delay than what was presented in Appendix E.3, and the average-case delay never yields the best average return for any benchmark.
>
> We decided to include the worst-case delay in the main body of the paper since constant-delay methodologies like BPQL and VDPO require the worst-case delay to operate on the MDPs they expect. Using the average-case delay does not result in an MDP since the delay can exceed the assumed upper bound. We also see examples where performance is worse under average-case delay, most consistently for BPQL under the M/M/1 queue delay process. Though we agree that the fairness in the evaluation is a valid concern, and we have therefore updated Section 5 in the paper to include how we motivated the methodology used to evaluate the constant-delay baselines, as well as referring to the cases in Appendix E.3 where constant-delay baselines are better than ACDA (please see the new uploaded document).
>
> > Given the paper's focus on the random delay problem, the authors have omitted a comparison with recent state-of-the-art methods that also specifically address random delays, such as DFBT [1] and State Augmentation-MLP [2]. Benchmarking against this relevant class of methods would be necessary to further substantiate the claimed effectiveness of the proposed method.
>
> As far as we are aware, from examining the papers and their code artifacts, neither methodology in [1] nor [2] considers the unobservable random delay scenario. A key aspect of handling unobservable random delays is determining how to act in the absence of perfect information. Other works, such as DEZ [3], explicitly operate under observable delays and describe real-world contexts where this assumption may be reasonable; however, we consider the problem statement where this assumption is not reasonable. We would appreciate clarification if we have overlooked anything in the works by [1] and [2].
>
> > The paper's presentation is not good enough. The notation introduced is somewhat complex, and the process described in the text for Figures 2 and 3 is difficult to follow. For example, in Figure 2, the most recent observation received by the agent is labeled $o_{t-2}$, yet the description in lines 193-194 refers to $o_t$ as the agent's observation, which is confusing. Furthermore, when Figure 3 is first introduced (lines 210-211), the definition for the counter $c_t$, which is essential to the figure caption, has not yet been provided, making it difficult to understand.
>
> Time is relative to the interaction layer, where the agent asynchronously acts on incoming observation packets (assuming that they arrive in order). Figure 2 illustrates an overview of the interaction, showing $o_t$ that was just generated by the interaction layer, and $o_{t-2}$ generated two steps earlier, just arriving at the agent. However, because the agent is oblivious to the actual time at the interaction layer when it receives an observation packet (due to our lack of requirement for synchronized clocks), the agent can assume that it received the observation instantaneously. Therefore, we also use $o_t$ from the agent's perspective, and also to avoid the use of notation such $o_{t-{\tau_o}}$. We have updated Section 3 to clarify this, and to the use of $c_t$ in Figure 3 (please see the new uploaded document).

---

> ### Author Response · Authors · 2025-11-24
>
> > In lines 418-419, the authors state, "we expect ACDA not to perform as well with the M/M/1 queue delays that fluctuate more." However, in Table 1, ACDA's performance under the MM1 Delay process is exceptionally strong, in most environments even exceeding its performance on the GE1,23 and GE4,32 processes. This result appears to contradict the authors' statement. Can the authors explain this phenomenon?
>
> We were referring to the relative performance of ACDA compared to other baselines under M/M/1 queue delays. We see that this was unclear in the original submission, and have updated this sentence in the second paragraph of Section 5 to be more clear (please see the new uploaded document).
>
> > The parameter L does not appear to be explicitly defined in the main text. The reviewer found in Appendix F.2 that L represents the "prediction length," but how this parameter is determined is a significant concern for the reviewer. If L is set to the maximum value of the random delay, this implies an assumption that the maximum random delay is known, which would weaken the advantage of this work in handling unobservable random delays. If L is set as a fixed empirical value, the reviewer hopes the authors can provide the corresponding justification.
>
> Thank you for your comment. We have updated Section 3 to include a paragraph that introduces the prediction length $L$. In the interaction layer framework, $L$ is determined by the agent for each action packet. Although this allows agents to adjust to maximum delays dynamically, ACDA assumes a maximum delay for each delay process, which is also used for $L$. We have updated Appendix B to include this information.
>
> If the delay for an action packet exceeds its prediction length L, that packet will be discarded. We have also updated Section 3 to clarify this further, as this behavior was previously only described in the formalism in Appendix C. In the case of M/M/1 delays, there is no maximum delay, and a sampled delay will likely exceed L.
>
> The values $\delta_t$ and $c_t$ in the observation packet convey information that agents could use to adjust $L$ dynamically. ACDA only uses these values to reconstruct trajectories for training purposes. Although it is not unfeasible to a priori estimate the worst-case delay, we see directions for future work where $L$ is adjusted dynamically.
>
> > A very interesting result in Table 1 is why DCAC, a method explicitly designed for random delays, performs so poorly in this random delay environment—far worse, in fact, than the constant-delay methods BPQL and VDPO. Is this poor performance attributable to the stochastic transitions induced by the authors' use of action noise, which perhaps DCAC is not well-suited to handle?
>
> No, we do not believe this to be due to the action noise. DCAC does not generate actions with a guarantee of when they will be applied, which is the case with BPQL and VDPO because of constant delay, and also the case with ACDA because of the action packet. The original DCAC results also show that its performance starts to degrade for random delays that are smaller than those used in our delay processes.
>
> Thank you again for your review. We hope you found our answers convincing, and we are happy to engage in further discussion if there are any additional questions.
>
> [1] Directly Forecasting Belief for Reinforcement Learning with Delays. ICML 2025
>
> [2] Addressing Signal Delay in Deep Reinforcement Learning. ICLR 2024
>
> [3] Valensi et al. "Tree search-based policy optimization under stochastic execution delay." 2024

---

### Official Review · Reviewer_5i36 · 2025-10-29

**Soundness:** 4
**Presentation:** 4
**Contribution:** 3
**Rating:** 6
**Confidence:** 3

**Summary:**

This paper proposes Actor-Critic with Delay Adaptation (ACDA), a model-based reinforcement learning algorithm designed to handle random, unobservable, and time-varying delays between the agent and the environment. The authors introduce an interaction layer that models delayed interactions as a POMDP, allowing the agent to generate a matrix of future actions and adapt dynamically when delays vary. ACDA employs a heuristic to estimate previously applied actions and a model-based distribution embedding to represent future state distributions.
Extensive experiments on multiple MuJoCo locomotion benchmarks with various stochastic delay processes show that ACDA consistently outperforms state-of-the-art baselines such as BPQL, VDPO, and DCAC. The method demonstrates strong robustness to delay variability.

**Strengths:**

1. The paper tackles an important and practical challenge in reinforcement learning, handling random, unobservable, and time-varying delays, which is highly relevant to real-world systems such as networked and robotic control.

2. The proposed interaction layer is conceptually elegant, bridging MDP and POMDP formulations for delayed environments and offering a general framework for asynchronous agent-environment interactions.

3. The ACDA algorithm integrates a heuristic delay adaptation mechanism with a model-based distributional embedding, enabling the agent to remain robust to stochastic delays without explicit delay estimation.

4. The experimental evaluation is extensive and well-executed, covering multiple MuJoCo tasks and diverse delay processes, with thorough ablations.

Overall, the paper is clearly written and easy to follow, with strong motivation, a clear problem setup, and a well-structured methodology. The experimental evaluation is extensive and thorough, providing solid empirical support for the proposed approach.

**Weaknesses:**

1. The method assumes that delays remain approximately constant within short time windows. This assumption may not hold under highly dynamic or non-stationary delay patterns, potentially reducing the method's robustness.

2. The paper does not provide formal convergence analysis, stability proofs, or error bounds. As a result, the robustness of ACDA is supported primarily by empirical evidence rather than theoretical justification.

**Questions:**

1. Is there a potential direction toward providing formal guarantee, such as performance bounds or stability proofs, under certain classes of stochastic delays?

2. How might the interaction-layer framework generalize to discrete-action, multi-agent, or large-scale RL settings?

3. Could future work explore learning the action-buffer management policy itself (rather than fixing its size or update rule), allowing the system to dynamically adjust to varying network conditions?

I find this paper to be creative and well-executed, with a clear motivation and a solid methodological contribution. While the lack of theoretical guarantees slightly limits the depth of the work, the paper presents a meaningful step forward.
Overall, I hold a positive opinion and would be happy to see the paper accepted.

---

> ### Author Response · Authors · 2025-11-24
>
> Thank you for the thorough review and for the encouraging words! You raise excellent questions, which we do our best to answer below.
>
> > Is there a potential direction toward providing formal guarantee, such as performance bounds or stability proofs, under certain classes of stochastic delays?
>
> We believe that the first step would be to formally analyze ACDA under ahead-of-time observable delays (agent observes delay of action packets before generating them), replacing the heuristic in Section 4.1 with an oracle. This is similar to the randomly delayed setting in DEZ [1]. As we touch upon in the beginning of Appendix B.1, there are also directions for formal guarantees in the observable delay setting with regards to the stochasticity of the environment. The potential direction we see going forward is to show or assume properties in the observable delay setting and then transfer those results to the unobservable delay setting. As deep RL domain is not well understood theoretically, we expect that any performance and stability guarantees for ACDA has to be done in a context where the learning aspect is mapped to a domain that is well understood.
>
> > How might the interaction-layer framework generalize to discrete-action, multi-agent, or large-scale RL settings?
>
> The interaction layer framework accepts discrete actions in the current formulation. However, the extended action space of the POMDP is always of infinite size due to the time tag and unbounded rows in the matrix. The continuous action spaces in the evaluation are due to ACDA being based on SAC; however, we do not see any obstacle to an alternative algorithm based on PPO, which can handle discrete actions.
>
> In the multi-agent setting, if all agents independently control different aspects of the environment, the natural generalization would be to have one interaction layer for each agent. It is not as clear what a good generalization would be in other scenarios, such as turn-based interaction on the same part of the environment. We believe that a generalization of the interaction layer would heavily depend on the multi-agent scenario.
>
> To apply the interaction layer in large-scale RL, we see a need for generalizations to the interaction layer framework to optimize the amount of information exchanged between the agent and the interaction layer, particularly concerning the action matrix when the underlying action space is large. We have added a new Appendix F.4 to the paper, outlining directions for minimizing the communication load by removing information from the action packet that is predicted to be unlikely to be used. To further generalize the interaction layer for large-scale RL, we see a need to relax the assumption that observation packets arrive in order and that all parts of an observation are contained within the same packet.
>
> > Could future work explore learning the action-buffer management policy itself (rather than fixing its size or update rule), allowing the system to dynamically adjust to varying network conditions?
>
> Yes, we see space for future frameworks where the agent has more control over what happens with the packets sent to it. While the current interaction layer framework allows for dynamically adjusting the prediction length $L$, we see an opportunity for more control over how a packet should be accepted or discarded, as well as sparse action matrix representations that can omit information for unlikely scenarios, among other improvements. We believe that these management policies are key to algorithms going beyond the heuristics of ACDA.
>
> We hope that we have answered all your questions, and we are happy to engage in further discussion.
>
> [1] Valensi et al. "Tree search-based policy optimization under stochastic execution delay." 2024

---

### Official Review · Reviewer_ShxM · 2025-10-30

**Soundness:** 3
**Presentation:** 2
**Contribution:** 2
**Rating:** 4
**Confidence:** 4

**Summary:**

This paper studies unobservable random delays in MDPs, where action execution may be randomly delayed, and the agent observes the delays only in hindsight. The authors introduce an interaction layer to handle these delays. Specifically, at each timestep, the agent generates an action matrix containing a sequence of actions to be executed until the next action packet, for every possible delay length. They then build a model to predict the future state from the current state and the future actions to be executed, and use it with a version of SAC combined with BPQL for training. Experiments conducted on five MuJoCo environments under three random delay processes demonstrate superiority over the baselines.

**Strengths:**

- The problem setting is important, as delays are often stochastic and action delays are not immediately observable.
- The paper is well-motivated and provides a good overview of existing work.
- The proposed interaction layer, which represents actions as a matrix, is novel.

**Weaknesses:**

- A key weakness of the approach is that representing the action packet as a matrix can be computationally and communicationally expensive. The work is motivated by improving efficiency under delays; however, computing the action matrix itself is significantly more time-consuming than generating a single action at each timestep. In particular, while the agent executes only $T$ actions, the framework requires computing and transmitting $T \times L^2$ actions over the network. As a result, many actions are never used, yet they add computational overhead, which could further increase delays in practice.


- The presentation of the method needs improvement. I had to read some parts multiple times and cross-reference earlier parts to understand the details. Some sections include irrelevant discussions, while key aspects are missing. For example, the POMDP formulation is disconnected from the rest of the paper, and the claim that a single delay $d_t$ is equivalent to other types of delays is neither proved nor sufficiently discussed.

- While the overall approach is novel, the two main components: generating sequences of actions and employing a model-based distributional agent, have been extensively explored in prior work. The idea of generating a sequence of actions is closely related to *action chunking*, an active research topic in the RL community. Similarly, model-based distribution agent idea has appeared in previous works on delayed RL[1, 2]. The authors could have focused more on explaining the unique contributions of their method rather than discussing previously established ideas in detail.

- I also have concerns regarding the experimental evaluation. First, at least one baseline result does not match the values reported in prior work (see under *Questions*). Second, the choice of delay distributions appears to favor the proposed method. The constant-delay baselines, adapted to handle random delays, assume the worst-case delay at every step. Although the authors partially address this concern in the appendix by modifying the assumed delay upper bound (showing improved results for baselines) this adjustment still favors their method. A more fair evaluation would include the uniform delay distribution, where baseline methods operate under the correct assumption.

[1] Wang, Wei, et al. "Addressing signal delay in deep reinforcement learning."

[2] Liotet, Pierre, et al. "Learning a belief representation for delayed reinforcement learning."

**Questions:**

- I assume L or h is the upper bound of the delay. What is the other variable?
- What happens if an outdated action packet arrives at the action buffer?
- It is not clear to me how $d_t$ replaces both observation and action delays. Could you elaborate? In particular, under both types of delays, there are timesteps where the agent receives no new observations. How is this situation equivalent in your formulation?

- The reported DCAC performance on HalfCheetah-v4 is 35.60 for the MM1 delay process. Based on the MM1 process shown, all delays are less than 21, with a mean around 5. However, previously reported DCAC results [3] on constant delays of 5 and 25 are approximately 2000 and 600, respectively, on the same environment without noise. From my experience, introducing 5% noise is unlikely to cause such a large performance gap. What is the source of this discrepancy?

[3] Wu, Qingyuan, et al. "Variational delayed policy optimization."

---

> ### Author Response · Authors · 2025-11-24
>
> Thank you for taking the time to write such a thorough review of our paper. We address your concerns and answer your questions below, and are also willing to engage further.
>
> > A key weakness of the approach is that representing the action packet as a matrix can be computationally and communicationally expensive. The work is motivated by improving efficiency under delays; however, computing the action matrix itself is significantly more time-consuming than generating a single action at each timestep. In particular, while the agent executes only $T$ actions, the framework requires computing and transmitting $T \times L^2$ actions over the network. As a result, many actions are never used, yet they add computational overhead, which could further increase delays in practice.
>
> In the original ICLR submission, we believe we have already addressed your concerns in Appendix F.2 by estimating the computational delay for the evaluated environments. We found that the estimated computational delay of 6.1 ms (measured on the same hardware used to run benchmarks) was below even the lowest actuation period (8 ms) in the evaluated environments. The computational delay should not be a problem for the evaluated environments. However, since we are aware that other environments may exist where this delay can be a problem, we developed and evaluated an action skipping method to mitigate excessive computational delays should they exceed the actuation period of the underlying system. This method, compatible with the existing interaction layer framework, only computes action packets based on the last received observation packet. If multiple observation packets reach the agent during the period when it is computing a single action packet, it discards all observation packets except the last one. This skipping works because ACDA computes horizons of actions that fill the gaps if a new action packet does not arrive at a specific time step. Our evaluation shows that ACDA can still achieve good performance, even if only computing an action packet based on every 8th observation packet. Amortized, this results in a computational delay that is ⅛ of the original. We have detailed this approach and present the preliminary results in the new Appendix F.3 (please see the new uploaded document).
>
> The newly uploaded document also contains an analysis of the communication overhead in Appendix F.4. This analysis shows that the worst-case bandwidth requirement for our benchmarks is 4.5 MiB/s, which is no problem for modern communication channels, such as 5G and WiFi. Should the bandwidth become a problem in a lower bandwidth channel, such as Bluetooth (1-2 Mbps), we propose suggestions in Appendix F.4 for reducing the communication overhead, including sparse action matrices and action skipping.
>
> > The presentation of the method needs improvement. I had to read some parts multiple times and cross-reference earlier parts to understand the details. Some sections include irrelevant discussions, while key aspects are missing. For example, the POMDP formulation is disconnected from the rest of the paper, and the claim that a single delay $d_t$ is equivalent to other types of delays is neither proved nor sufficiently discussed.
>
> The POMDP is only partially introduced in Section 3, with the purpose to establish notation and highlight the problem of ahead-of-time unobservable delays. This is used within the main body of the paper. The full formal definition of the POMDP is instead located in Appendix C.
>
> The delay $d_t$ is not equivalent to any other specific delay, but rather represents the round-trip delay consisting of observation, computation, and application. We have updated Section 3 with further clarification regarding the single delay $d_t$ (please see the new uploaded document). We hope that this clarifies potential misunderstandings.

---

> > ### Author Response · Authors · 2025-11-24
> >
> > > While the overall approach is novel, the two main components: generating sequences of actions and employing a model-based distributional agent, have been extensively explored in prior work. The idea of generating a sequence of actions is closely related to action chunking, an active research topic in the RL community. Similarly, model-based distribution agent idea has appeared in previous works on delayed RL[1, 2]. The authors could have focused more on explaining the unique contributions of their method rather than discussing previously established ideas in detail.
> >
> > Action chunking concerns situations where multiple actions are intended to be executed as a unit, whereas ACDA instead generates a sequence of actions to cover gaps in the interaction, without any assumption that all actions in the sequence will be applied. Although we do not see these as equivalent approaches, this is an interesting parallel, and we have added references to action chunking in the new uploaded document.
> >
> > While Section 4.2 introduces the model-based agent, its primary purpose is to describe how ACDA populates the action packet matrix using this model. The referenced papers present similar models to ours, but we do not see that our model structure (explicitly embedded state distribution updated recurrently) is exactly described by the previous works, where [2] uses a transformer architecture and [1] describes state-action pairs as inputs to the recurrence. We would appreciate clarification if we have overlooked anything in these papers.
> >
> > > Second, the choice of delay distributions appears to favor the proposed method. The constant-delay baselines, adapted to handle random delays, assume the worst-case delay at every step. Although the authors partially address this concern in the appendix by modifying the assumed delay upper bound (showing improved results for baselines) this adjustment still favors their method. A more fair evaluation would include the uniform delay distribution, where baseline methods operate under the correct assumption.
> >
> > We evaluate our approach on realistic delay processes, such as M/M/1, which models delays in a network queue. It is unclear what real-world condition a uniform random delay process would correspond to. The evaluated constant-delay baselines do operate under the correct assumption, given the constant-delay augmentation described in Appendix A. Note also that we can enforce the constant delay on ACDA, which we evaluate in Appendix E.2 (denoted as BPQL w/ MDA, since ACDA acts as BPQL under constant delay, but with a model-based policy instead of an MLP).
> >
> > > I assume L or h is the upper bound of the delay. What is the other variable?
> >
> > We recognize that these variables were implicitly defined in the equations and text, and have therefore updated the paper to clearly introduce them in Section 3 (please see the new uploaded document). $L$ represents the largest possible delay handled by an action packet, whereas $h$ represents the capacity of the action buffer. $L$ could be dynamically adjusted by a different kind of agent (ACDA keeps $L$ constant), where $h$ is part of the interaction layer configuration and remains fixed during the interaction.
> >
> > > What happens if an outdated action packet arrives at the action buffer?
> >
> > An action packet is discarded if it is outdated or if the delay of the action packet exceeds $L$. This behavior is part of the POMDP formalism described in Appendix C; however, as these considerations are also of practical importance, we have updated the main text to include this information as well (please see Section 3.2 in the new uploaded document).
> >
> > > It is not clear to me how  $d_t$ replaces both observation and action delays. Could you elaborate? In particular, under both types of delays, there are timesteps where the agent receives no new observations. How is this situation equivalent in your formulation?
> >
> > In the interaction layer formalism, time is viewed from the perspective of the interaction layer, rather than the agent, which acts reactively to incoming observation packets. As discussed in Section 3.3, $d_t$ represents the delay from the perspective of the interaction layer, which only considers the round-trip delay of receiving an action based on a generated observation packet. We updated the paper with further clarifications about this (please see the new uploaded document).

---

> > > ### Author Response · Authors · 2025-11-24
> > >
> > > > The reported DCAC performance on HalfCheetah-v4 is 35.60 for the MM1 delay process. Based on the MM1 process shown, all delays are less than 21, with a mean around 5. However, previously reported DCAC results [3] on constant delays of 5 and 25 are approximately 2000 and 600, respectively, on the same environment without noise. From my experience, introducing 5% noise is unlikely to cause such a large performance gap. What is the source of this discrepancy?
> > >
> > > We believe that this is because DCAC does not adapt to random delays; it sends a single action without any guarantees of when it will be applied. In the constant delay case, DCAC always generates actions for the same delay and therefore has guarantees on when they will be applied. The original results [4] show that DCAC also struggles in some of their random WiFi delay benchmarks, with delays much lower than for our delay processes. The interaction layer, utilized by ACDA, gives agents control over when actions are applied under random delays, which is not the case for DCAC.
> > >
> > > Thank you again for your detailed review. We hope we addressed your concerns, and are happy to engage in further discussion.
> > >
> > > [1] Wang, Wei, et al. "Addressing signal delay in deep reinforcement learning."
> > >
> > > [2] Liotet, Pierre, et al. "Learning a belief representation for delayed reinforcement learning."
> > >
> > > [3] Wu, Qingyuan, et al. "Variational delayed policy optimization."
> > >
> > > [4] Yann Bouteiller, et al. “Reinforcement learning with random delays.”, 2021

---

> > > > ### Comment · Reviewer_ShxM · 2025-11-27
> > > >
> > > > Thanks to the authors for their rebuttal.
> > > >
> > > > I understand that the authors considered realistic delay distributions for their experiments. However, introducing a method capable of addressing random delays requires stronger evidence in my view. The random distributions considered in this paper have a special property: they are very small most of the time and only rarely generate large values. This is clearly visible in the GE variants and, although relatively less obvious in MM1, the gap between your method and the baselines is also much smaller there. This raises a serious question about whether the observed gains hold for a more general family of random delays or whether they are tied to this special class of distributions.
> > > >
> > > > Therefore, I cannot recommend this paper for acceptance as applicability to more general random distributions remained unclear.

---

> > > > > ### Author Response · Authors · 2025-12-02
> > > > >
> > > > > Thank you for your response to our rebuttal. As stated in the introduction of our ICLR submission, ACDA is specifically designed for temporally correlated delays, and we do not claim generalizability to arbitrary random delay distributions. Our experiments using delay distributions representative of temporally correlated delays, such as the standard M/M/1 model, demonstrate that ACDA outperforms existing state-of-the-art approaches, even when adjusting the assumed delay to favor constant-delay methods.

---

### Meta-Review · Area_Chair_zrW1 · 2026-01-05

**Summary:**

In this work, the authors introduce a framework that enables agents to adaptively handle unobservable and time-varying delays in online reinforcement learning. Building on this framework, model-based algorithm, Actor-Critic with Delay Adaptation (ACDA) has been developed, which significantly outperforms alternative approaches across diverse locomotion tasks.

The reviewers appreciate the paper for its technical novelty, importance of the problem, elegant formulation, extensive experiments and promising empirical results. After the rebuttal, some of the reviewers' concerns have been resolved, such as the computation overhead, baseline (DCAC) performance, presentation and notation issues. However, important concerns remain. The ACDA method, specifically designed for temporally correlated delays, remained unclear whether it can apply to more general random distributions. Also, lack of theoretical guarantee slightly limits the depth of the work. In summary, I believe the current study could benefit from establishing itself on more general delay distributions and performing more theoretical or qualitative analysis on it strong performance.

**Reviewer Concerns:**

### Addressed Concerns

- Computational overhead
- Missing baselines (DFBT, state-augmentated MLP)
- DCAC baseline performance
- Clarity & notation
- Baseline comparison fairness

### Remained Concerns

- Generality of delay distributions
- Theoretical guarantees

**Reviewer Scores:**

Reviewer ShxM and 3YFw have stated that they will keep the scores. Reviewer 5i36 may not change the score since the authors did not response to his/her concerns (weakness). Reviewer Tjob is likely to increase the score.

---

### Decision · Program_Chairs · 2026-01-26

Reject